# Characteristics of ocean mesoscale eddies in the Canadian Basin from a high resolution pan-Arctic model

Noémie Planat[1], Carolina O. Dufour[1,2], Camille Lique[2], Jan K. Rieck[1], Claude Talandier[2], and L. Bruno Tremblay[1]

[1]McGill University, Departement of Atmospheric and Oceanic Sciences, Montréal, Québec, Canada
[2]University of Brest, CNRS, Ifremer, IRD, Laboratoire d'Océanographie Physique et Spatiale (LOPS), IUEM, F29280, Plouzané, France

**Correspondence:** Noémie Planat (noemie.planat@mail.mcgill.ca)

**Abstract.** Mesoscale eddies are ubiquitous in the Arctic Ocean and are expected to become more numerous and energetic as sea ice continues to decline. Yet, the spatio-temporal characteristics of these eddies are poorly documented. Here, we apply an eddy detection and tracking method to the output of a high-resolution ($1/12°$) regional model of the Arctic - North Atlantic in order to investigate mesoscale eddies in the Canadian Basin over the period $1995 - 2020$. Over that period, about $6,000$ eddies per year are detected in the surface layer, while about $9,000$ eddies per year are detected in the pycnocline layer, and about $5,500$ eddies per year are detected in the Atlantic Water layer. The eddy population is generally distributed about equally between cyclones and anticyclones. Yet, within the pycnocline and surface layers, a clear dominance of anticyclones over cyclones is found at the centre of the Beaufort Gyre, in line with observations from Ice Tethered Profilers (ITPs). The observed dominance of anticyclonic eddies reported by ITPs thus likely partially arises from the regional focus of the ITPs. On average, eddies travel 11 km, have a radius of 12.1 km, and last 10 days, although the majority of eddies are short-lived ($50\%$ of eddies last less than 4 days). These statistics hide strong regional and temporal disparities within the eddy population. In the surface layer, the seasonal, interannual, and decadal variability in the number of eddies and in their mean characteristics follow that of the sea ice cover. In contrast, within the pycnocline layer and below, the number and properties of eddies show a weakened seasonality. At all depths, the characteristics and density of the eddy population show a strong asymmetry between the slope and the centre of the Canadian Basin. While the upper 85 m show a greater number of eddies over the slope than over the centre of the basin, this pattern is reversed in the pycnocline layer, where a muted eddy activity is observed along the slope and up to 300 km offshore. Within the Atlantic Water layer, a relatively large number of eddies is generated in the vicinity of the cyclonic boundary current along the slope. The vast majority of eddies have a weak temperature and salinity signature with respect to their environment, although a significant portion of the long-lived eddies, located along the Chukchi shelf break, have a relatively large temperature anomaly and penetrate into the Beaufort Gyre, thus suggesting a mechanism for the penetration of heat into the gyre. Over the 26 years analysed here, the number of eddies generated within the upper 85 m increases by $34\%$, with the largest increase occurring in the open ocean and marginal ice zone. Within the pycnocline layer, the number of eddies increases by $45\%$, with a strong year-long increase in 2008, presumably in response to the Beaufort Gyre spin-up in $2007 - 2008$ associated with the record low in sea ice extent. The number of eddies in the Atlantic Water layer shows an overall increase of $41\%$, but little interannual variability. We suggest that this model-based eddy census can thus

help investigate recent changes in the dynamical equilibrium of the Beaufort Gyre by providing a consistent spatio-temporal characterization of mesoscale eddies in the Canadian Basin over the past two decades.

## 1 Introduction

Observations and numerical models reveal that mesoscale eddies are ubiquitous in the Arctic Ocean, including under sea ice (e.g. Manley and Hunkins, 1985; Cassianides et al., 2023; Liu et al., 2024). These eddies are thought to play an important role in the transport of heat, salt, and nutrients from the shelves to the deep basins (Watanabe, 2011; Watanabe et al., 2014; Spall et al., 2008; Pickart et al., 2005), and possibly in the modulation of the Marginal Ice Zone (MIZ; Gupta et al., 2024; Martínez-Moreno et al., 2025; Manucharyan and Thompson, 2022). In the Canadian Basin, mesoscale eddies are also hypothesized to be a key component of the dynamical equilibrium of the large-scale circulation through the dissipation of potential energy that accumulates within the anticyclonic Beaufort Gyre (BG, Manucharyan et al., 2016; Manucharyan and Spall, 2016; Meneghello et al., 2020; Armitage et al., 2020). Additionally, eddies possibly play a role in the build-up of the subsurface heat reservoir by driving the penetration of relatively warm summer Pacific Waters (sPW) into the gyre (MacKinnon et al., 2021; Spall et al., 2018; Planat et al., 2025). However, despite their possible role in the thermo-dynamical equilibrium of the BG, characteristics of the mesoscale eddy field and their evolution through time remain largely unknown in the Arctic, for one part due to the sparsity of observations, in particular under ice, and for another part due to the high resolution needed for models to represent the mesoscale at high latitude.

In the literature, the term *eddy* encompasses a broad range of definitions. Observations of eddies in the Arctic Ocean have, however, mostly reported on coherent structures identified as anomalies with respect to their environment. Thus, from now onwards, we will focus on these coherent structures. Within the Canadian Basin, observations from Ice-Tethered Profilers (ITPs; Toole et al., 2011) and moorings, deployed as part of the Beaufort Gyre Exploration Project (https://www.whoi.edu/beaufortgyre), have enabled the detection of $\mathcal{O}(400)$ eddies between 2004 and 2019 (Zhao et al., 2014; Cassianides et al., 2023). The majority of these eddies were found within the halocline $(50-300 \text{ m})$, with a few detected at greater depth. Analyses of synthetic-aperture radar data in 2007, 2011, and 2016 in the Western Arctic identified more than $7,500$ surface eddies within the seasonally ice-free and MIZ regions (Kozlov et al., 2019). Similarly, altimetry-based detection within the seasonally ice-free region reported $2,000$ eddies between 1993 and 2018 (Kubryakov et al., 2021), an order of magnitude difference from *in situ* observations, likely due to the better spatial coverage of the ice-free region and MIZ. Analyses of rotating ice floes with optical satellite images were also used to provide information about the eddy population within the MIZ of the BG, revealing thousands of eddy-like signatures over the last two decades (Manucharyan et al., 2022). Even though no consensus was found in the eddy count across the different observational datasets, all satellite observations have shown regions densely populated with eddies over the continental shelf and slope, and in the open ocean and MIZ (Kozlov et al., 2019; Kubryakov et al., 2021), while *in situ* observations have demonstrated the presence of numerous eddies in the central basin below sea ice and at depth

(Carpenter and Timmermans, 2012; Zhao et al., 2014; Cassianides et al., 2023).

The density of eddy population, as well as their spatial extents (both lateral and vertical) and polarity, have provided hints at the processes driving eddy generation. In particular, the polarity has been scrutinized to understand the dynamics of eddies better, but reconciling the picture provided by the different observation datasets has proven difficult. While *in situ* observations show a predominantly anticyclonic eddy field ($> 95\%$, Zhao et al., 2014; Cassianides et al., 2023), in line with optical satellite imagery which finds twice as many anticyclonic as cyclonic ice floes (Manucharyan et al., 2022), altimetry shows an

equally distributed polarity (Kubryakov et al., 2021), and synthetic-aperture radar imagery shows twice as numerous cyclones as anticyclones (Kozlov et al., 2019). Some mechanisms of eddy generation were proposed to support the strong asymmetry documented from *in situ* observations, such as subduction processes at outcropping fronts (Manucharyan and Timmermans, 2013; Brannigan et al., 2017) and baroclinically unstable coastal boundary currents (D'Asaro, 1988; Hunkins, 1974; von Appen and Pickart, 2012). Furthermore, part of the anticyclone to cyclone asymmetry may be attributed to the stronger coherency and

a slower decay of anticyclones, a characteristic reported for eddies at lower latitudes (Chelton et al., 2011), probably leading to an over-representation of anticyclones in eddy censuses (Stegner et al., 2021; Giulivi and Gordon, 2006). Contrasts between surface-intensified and at-depth eddies may also impact the statistics of eddy polarity, but this is yet to be shown.

Investigating eddy sizes, *in situ* observations have reported eddies across both the submeso- and meso-scales, with radii ranging from 3 to 15 km when detected from ITP profiles and from 3 to 80 km when detected from mooring profiles (Cassianides

et al., 2023). Satellite observations have typically observed eddies with diameters ranging from $\mathcal{O}(10)$ km up to $\mathcal{O}(100)$ km (Kubryakov et al., 2021; Manucharyan et al., 2022), with synthetic-aperture radar images capturing features down to $\mathcal{O}(1)$ km (Kozlov et al., 2019). Only recently has the resolution reached by realistic models become fine enough to resolve at least part of the mesoscale spectrum in the Arctic, where the first Rossby radius of deformation varies between $\approx 15$ km in the Canadian Basin and $\approx 8$ km in the Eurasian Basin, down to $1 - 2$ km on the shelves (Nurser and Bacon, 2013; Hu et al., 2019; Wang

et al., 2020). An analysis of the Eddy Kinetic Energy (EKE) in the entire Arctic Ocean in a 1 km resolution model shows peaks of EKE at $400$ m depth at spatial scales of around 60 km (Liu et al., 2024). Further, Liu et al. (2024) show that about half of the EKE is contained in scales smaller than 30 km.

Within the water column, eddies are found to form in three distinct regions: at the surface, within the halocline, and at depth. In the upper surface layer, shallow eddies are confined by the strong stratification and have a vertical extent of typically 100 m,

while below, eddies can span up to $O(1)$ km and are located around $1,200$ m (Carpenter and Timmermans, 2012). In between, double core eddies have been detected with a shallow core at the base of the pycnocline and a deep core within the Atlantic Water (AW) (Zhao and Timmermans, 2015). Idealized model configurations of the BG have shown vertical modes of baroclinic instabilities with similar vertical structure (Meneghello et al., 2021). Overall, while observations have revealed different types and origins of eddies based on their dimensions and repartitions, the number of detected features has remained relatively low,

hence preventing a systematic documentation of their spatial characteristics and geographical distribution, and more robust statistics of the eddy population. The recent advent of fine-resolution ocean-sea ice models enables such an investigation. Yet,

it remains to be done.

In the Canadian Basin, mesoscale eddy activity displays a strong seasonal cycle at the surface that is directly linked to that of sea ice (Hunkins, 1974; Meneghello et al., 2021; Manucharyan and Thompson, 2022; Rieck et al., 2025b). In ice-free regions, thus mostly during summer, a vigorous mesoscale eddy activity is reported in both observations and models. In contrast, below sea ice, or more generally in winter, a quiescent surface layer is observed with eddies that last as short as a few days (Meneghello et al., 2021). The short lifetime of under-ice eddies highlights the role of sea ice in dissipating eddy energy through friction. Eddies may persist beyond months at subsurface since they are shielded from the effect of sea ice by the strong stratification (D'Asaro, 1988; Hunkins, 1974). However, subsurface eddy lifetime cannot be precisely estimated from observations, though, as both ITPs and moorings only capture a portion of the eddy trajectory.

As sea ice has shrunk and the gyre intensified in the Canadian Basin along the last decades, the number of mesoscale eddies is thought to have increased. Satellite observations of spinning ice floes hinting at the eddy field have suggested such a trend over the past two decades (Manucharyan et al., 2022). Likewise, the number of eddies has been found to vary on interannual time scales with the intensity and freshwater content in the BG (Kubryakov et al., 2021; Manucharyan et al., 2022; Zhang et al., 2016). These observations tend to confirm the suggested role of mesoscale eddies in the gyre equilibration through the conversion of potential energy, which accumulates within the freshwater reservoir at the centre of the anticyclonic BG, into EKE. Along the same line, modelling shows an enhancement of the EKE concurrent with the intensification of the gyre following increased wind forcing and sea ice retreat in 2007 (Regan et al., 2020). However, EKE was shown to only increase for a couple of years in the model of Regan et al. (2020). To fulfil their role in the dynamical equilibrium of the gyre, Manucharyan and Stewart (2022) argue that eddies should be generated from baroclinic instabilities within the gyre. Such a generation mechanism cannot, however, lead to strong polarity asymmetry, as documented from ITP measurements. To reconcile this dynamical constraint with observations, Manucharyan and Stewart (2022) further suggest that both types of eddies exist in the BG. On the one hand, small and cold anticyclones travelling freely from the shelfbreak, where they are generated through boundary current instabilities or outcropping fronts (Manucharyan and Timmermans, 2013; D'Asaro, 1988; Zhao et al., 2014), to the centre of the Gyre. On the other hand, larger and weaker eddies formed from baroclinic instability in the interior of the gyre that are yet to be observed from *in situ* measurements, and that actively participate in the dynamical equilibration of the gyre, as evidenced by Armitage et al. (2020) from observation-based energy budgets.

Finally, the shrinking and thinning of sea ice that has been observed over the past decades and is projected to continue into the future (Meredith et al., 2001; Meier and Stroeve, 2022) inevitably reduces the frictional dissipation of eddies, thus allowing more eddies to survive in the surface layer. The projections of the future Arctic with eddy-rich models show an increasingly energetic ocean with enhanced eddy activity in ice-free regions but also under sea ice (Rieck et al., 2025b; Li et al., 2024). The enhanced eddy activity at the surface may be driving more lateral mixing of heat with potential feedback on the ice. Likewise, changes in stratification and sea ice cover may have affected the eddy activity, characteristics, generation, and dissipation mechanisms. Notably, stronger baroclinic instabilities result from a less concentrated ice cover (Meneghello et al., 2021). In addition, if the upper layer stratification weakens as the sea ice reduces, subsurface eddies that persist all year long,

shielded from sea ice by the strong vertical stratification, may extend across reaching the surface (Meneghello et al., 2021). The evolution of the eddy characteristics over the Arctic in transition is yet to be investigated to understand the changes that have occurred over the last decades and foresee the upcoming changes in the eddy field and possible feedbacks on the ice cover.

Overall, no systematic and quantitative characterization of eddies has been performed in the Canadian Basin, leaving many questions open on the dynamics of that basin. In this paper, we propose a census of mesoscale eddies that develop in the Canadian Basin using a high-resolution regional model of the Arctic. To do so, we detect and track eddies to extract key properties such as size, lifetime, polarity, and thermohaline anomalies. The resulting eddy dataset comprises $O(10^3)$ eddies/year, which allows us to derive robust statistics on eddy properties. The dataset, which is fully coherent in space and time, is used to document changes of eddy characteristics between 1995 and 2020, hence covering a period of changes in ocean dynamical and sea ice state in the Canadian Basin. The rest of the paper is organized as follows. The model and the eddy detection and tracking algorithm are described in Section 2. The spatio-temporal eddy census is presented in Section 3. A discussion of key differences with observations is offered in Section 4 together with the main findings of this study and future perspectives.

## 2 Methods

### 2.1 The pan-Arctic high-resolution model CREG12

#### 2.1.1 Model and simulation

We use an updated version of the $1/12°$ regional Arctic-North Atlantic configuration CREG12 (Canadian Regional; Dupont et al., 2015). CREG12 is based on the ocean modelling platform Nucleus for European Modelling of the Ocean (NEMO) version 4.2.2 (Madec et al., 2023) and the Sea Ice modelling Integrated Initiative 3 (SI3) sea ice model, with levitating sea ice, five categories of ice, and two layers of snow (Vancoppenolle et al., 2023). The model is run on an ORCA12 seamless regional grid with horizontal resolution $\approx 3 - 4$ km in the central Arctic (Barnier et al., 2014). It uses a $z^*$ vertical coordinate with 75 levels spaced by $1$ m at the surface and $150$ m at $1,500$ m. This relatively fine horizontal grid size allows for an explicit resolution of most of the mesoscale spectrum within the deep basins where the first Rossby radius of deformation $R_0$ is $\approx 10 - 15$ km, but not over the continental slope and shelf where $R_0 < 7$ km (Nurser and Bacon, 2013, see also Fig. S1 in the supplementary material). Higher resolution simulations of the Arctic Ocean ($\approx 1$ km) have shown that the EKE spectrum peaks around $50$ km (Li et al., 2024) and that more than $80\%$ (resp. $65\%$) of the EKE is contained in scales larger than $10$ km (resp. $20$ km; Liu et al., 2024). Therefore, we argue that $1/12°$ is a resolution fine enough to represent most of the mesoscale features in the BG and along its margins (but not over the shelves), with an associated computational cost that allows for decadal integrations. The configuration includes a third-order momentum flux formulation, a second-order scheme for tracer advection, with an additional bi-Laplacian viscosity and diffusivity formulation depending on the local velocity, and a turbulence closure scheme for vertical mixing. The representation of tidal mixing effects is included in the comprehensive parameterization of mixing by breaking internal tides and lee waves (de Lavergne et al., 2016).

The simulation is initialized in 1979 from the World Ocean Atlas 2009 for temperature (Levitus et al., 2010) and salinity (Antonov et al., 2010) with the ocean at rest and is run until 2020. Sea ice conditions are initialized from the Pan-Arctic Ice Ocean Modeling and Assimilation System (PIOMAS; Zhang and Rothrock, 2003). The ocean and sea ice are forced with hourly atmospheric fields from the European Centre for Medium-Range Weather Forecasts Reanalysis version 5 (ERA5, Hersbach et al., 2020). To compensate for the known warm biases of the sea ice surface temperature of ERA5 (e.g. Batrak and Müller, 2019), the snow conductivity is set to $0.5$ $\text{Wm}^{-1}\text{K}^{-1}$, the ice-ocean drag coefficient to $7 \, 10^{-3}$, the atmosphere-ocean drag coefficient to $1.2 \, 10^{-3}$, and the ice strength to $2 \, 10^{-4}$ N m$^{-2}$. The open boundary conditions at the Bering Strait and along $27°$N in the Atlantic are specified daily from the output of GLORYS12V1, a global reanalysis at $1/12°$ resolution run from 1993 to 2020 (Lellouche et al., 2018). Prior to 1993, outputs of GLORYS12V1 between 1993 and 2021 are used to build a daily climatology and force the open boundaries of CREG12. At the Bering Strait, meridional velocities are adjusted to constrain the inflow to about 1.1 Sv, matching observation estimates (Woodgate, 2018). The river run-off and Greenland melting are specified following Weiss-Gibbons et al. (2024). An additional sea surface salinity restoring with piston velocity of 167 mm day$^{-1}$ is implemented in ice-free regions at monthly frequency using the World Ocean Atlas 2009 (Antonov et al., 2010). For additional details on the run, the reader is referred to Talandier and Lique (2024).

### 2.1.2   Evaluation of the simulation

In the rest of this paper, the Canadian Basin is defined as the region between $69 - 85°$N and $108 - 180°$W, thus fully encompassing the BG and its surrounding area. For analysis purposes, we define the Beaufort Box (BB), a region in the BG between $73 - 77°$N and $135 - 152°$W (see Fig. 1).

We present here a brief evaluation of the model's representation of the hydrography, circulation, and sea ice conditions in the Canadian Basin. For a more in-depth assessment of the model's performance, the reader is referred to Regan et al. (2020) and Barton et al. (2022) who use similar configurations. In this study, we focus on the period $1995-2020$ to let the model equilibrate between 1979 and 1994. Over the period of analysis, the mean September sea ice concentration is comparable to that derived from satellite observations, with small differences on the Eurasian shelf and a low bias in the western Canadian Basin (Fig. 1a,b). Across the Arctic and along the simulation, the sea ice extent deviates from that derived from satellite observations by around $-7\%$ in September and $-16\%$ in March on average. When compared to the PIOMAS reanalysis (Zhang and Rothrock, 2003), the sea ice thickness is 35 cm thinner in September and 20 cm thinner in March (Fig. S2b, S3b in the supplementary material). The interannual variability of sea ice extent is well captured by the model across the 26 years of simulation. A strong decline in sea ice concentration starting around 2000 and persisting in time appears in the model, in agreement with observations (Fig. S2c, S3c in the supplementary material). The corresponding location of ice loss is generally well represented despite some biases in the ice concentration along the Eurasian shelf in summer, and high biases along the Yermack plateau and the Greenland eastern shelf in winter (Fig. S2a,b, S3a,b in the supplementary material).

The Sea Surface Height (SSH) patterns are comparable in the model and observations (Fig. 1c,d), indicating that the surface geostrophic circulation of the model correctly reproduces the circulation of the BG (anticyclonic) and of the Nansen Basin

(cyclonic). Within the BG, CREG12 successfully represents the vertical distribution of temperature extrema associated with the three main water masses present in this region (Fig. 1e,f), namely the sPW (temperature maximum at 100 m), the winter
Pacific Waters (wPW; temperature minimum at 200 m), and the AW (temperature maximum at 550 m). Small biases in the magnitude of the temperature extrema themselves (warm bias for the wPW, and cold bias for the sPW and AW) are noted. Despite a high salinity bias at the surface in CREG12, the modelled stratification displays the so-called "bowl shape" of the BG, visible through the tilted isopycnals along the edges of the gyre, although slightly weaker in the northernmost side of the BG in CREG12. The overall fresh water content, referenced to $34.8$ g kg$^{-1}$, shows a strong increase between $2003-2009$
in the Canadian Basin as documented from the Beaufort Gyre Exploration Project (Proshutinsky et al., 2009) followed by a plateau (Fig. S4 in the supplementary material).

Overall, the model offers a realistic representation of the main circulation features, with the anticyclonic BG extending down to $\approx 250$ m and intensifying along the Chukchi shelf break (see the map of Mean Kinetic Energy (MKE) in Fig. S5 in the supplementary material). The cyclonic boundary current within the AW layer is found around the BG at 500 m with a returning
branch of weaker intensity along the Canadian Archipelago (Fig. S5c). Upper outflows through the Canadian Archipelago are similar to observation-based derived circulation (Fig. S5, see also Planat et al., 2025). Climatologies of EKE computed relative to monthly means show large values along the shelf break and along topographic features such as the Northwind ridge, both at the surface (not shown) and within the pycnocline (Fig. 2a). In contrast, the deep basin is more quiescent, with EKE one to two orders of magnitude lower than on the shelves (Fig. 2a,b). The shelf-deep basin contrast in EKE magnitude is a typical feature
of the mooring-based estimates (von Appen et al., 2022). Yet, the intensity of EKE is about one order of magnitude smaller in our model than that derived from observations (von Appen et al., 2022), as documented previously in Regan et al. (2020). The MKE, which captures the location of the main currents, is of similar order of magnitude as in observations (von Appen et al., 2022), with discrepancies being partly attributed to the difference in the exact location of the main currents between models and observations (Fig. S5 in the supplementary material). Finally, the vertical structure of the total kinetic energy is similar
to that derived from the Beaufort Gyre Exploration Project Moorings (compare Fig. S6 in the supplementary material with for instance Fig. A1 from Meneghello et al., 2020) with sub-surface intensified structures between $30-200$ m, and deeper (although weaker) structures between $400-2,000$ m, as evidenced in observations by Carpenter and Timmermans (2012).

## 2.2 Detection and tracking of mesoscale eddies

### 2.2.1 Detection

We perform an offline detection and tracking of mesoscale eddies within the Canadian Basin over $1995-2020$. The mesoscale eddies we capture span a broad range of mesoscale rotating features, from the evanescent vortices quickly dissipated by sea ice to the more persistent features that may eventually evolve into materially coherent vortices. The eddy population detected thus includes parts of the "turbulent soup" that is expected to develop at the surface in response to the atmospheric and ice forcings, and should be captured by the model. Though short-lived, these features, which are characteristic of the surface
ocean, deserve an investigation as they allow one to examine the energy dissipation exerted by sea ice and contribute to the

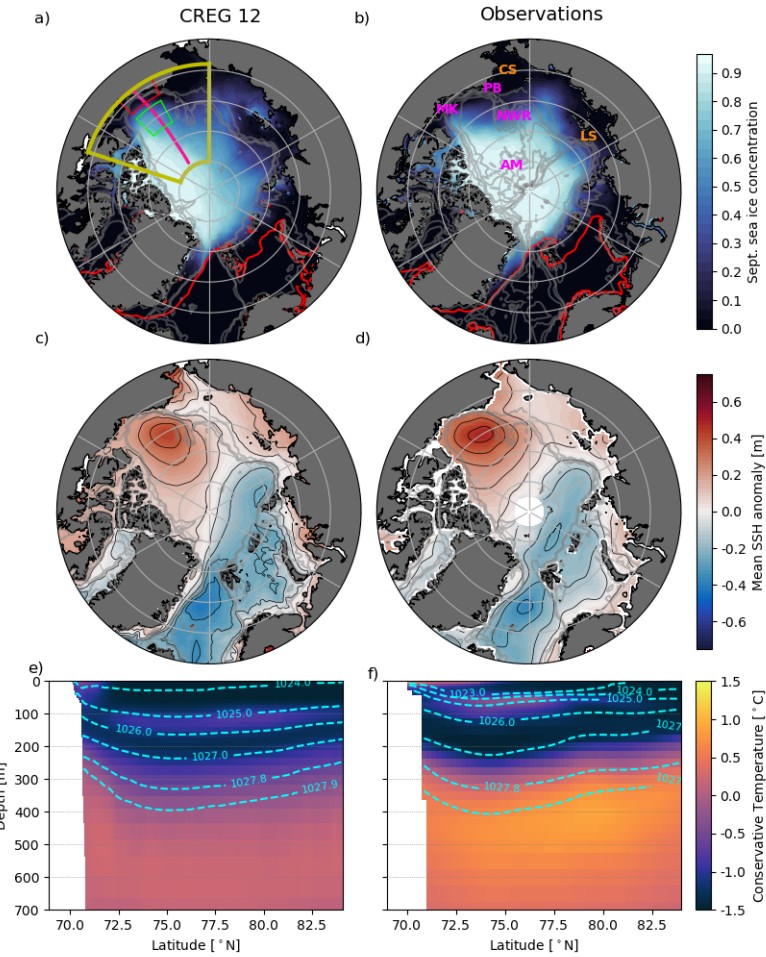

**Figure 1.** Mean sea ice concentration in September (background color) and in March (80 % contour in red) over $1995-2020$ from (a) CREG12 and (b) National Snow and Ice Data Center (NSIDC) Climate Data Record (DiGirolamo et al., 2022), a blend between the NASA-Team algorithms (Cavalieri et al., 1984) and the NASA-Bootstrap algorithm (Comiso, 1986). Mean Sea Surface Height (SSH) anomaly with respect to the mean over $2011-2020$ and above $65°$N from (c) CREG12 and (d) the updated altimetry-based product of Armitage et al. (2016). Black contours are evenly spaced every 0.1 m between $-0.75$ m and 0.75 m. Mean conservative temperature (background) and potential density referenced to surface (dashed contours) along a transect at $145°$W over $2005-2014$ from (e) CREG12 and (f) the World Ocean Atlas 2023 climatology (Locarnini et al., 2024; Reagan et al., 2024). Note the different periods displayed for each variable to match those of the observation datasets. Boxes on panel a) represent the regions used for our analyses corresponding to the Alaskan shelf area (red) and the Beaufort Box (BB, green). The thick yellow box indicates the Canadian Basin i.e., the entire domain analysed in this study. The pink line is the section used for panel (e) and (f), and for Fig. S9 in the supplementary material. In panel b), CS and LS stand for Chukchi and Laptev sea, respectively, and NWR, PB, AM, and MK for Northwind ridge, Pt. Barrow, Alpha-Mendeleev ridge, and McKenzie river, respectively. Thin gray lines show the bathymetry, respectively the 100, 500, and $1,000$ m depth isobaths on (a), (c), (d), and the 100, 500, $1,000$, $2,000$, and $3,000$ m depth isobaths on (b).

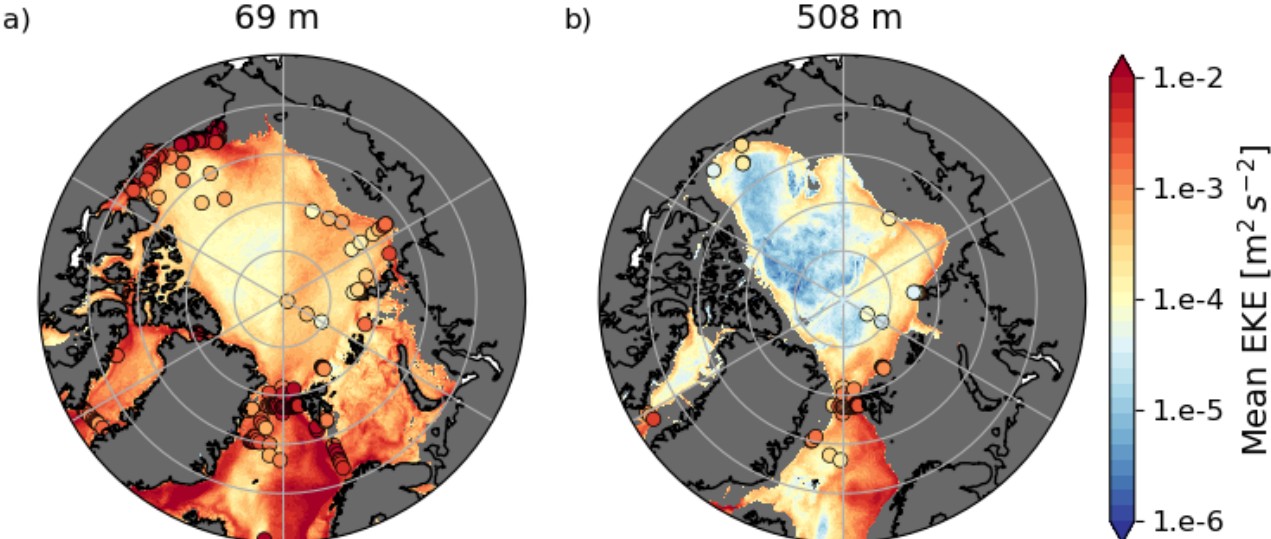

**Figure 2.** Eddy Kinetic Energy (EKE) computed from velocity anomalies with respect to the monthly means in CREG12 and averaged over the 26 years of simulation (a) at 69 m (within the halocline) and (b) at 508 m (within the Atlantic Waters (AW) layer). Superimposed are mooring-based estimates of EKE from von Appen et al. (2022), computed with a fourth-order Butterworth filter with 2-day to 30-day cutoffs. The reader is referred to von Appen et al. (2022) for the exact calculation method.

dynamical equilibrium of the basin. In the following, we focus on features with characteristic sizes from $R_0 \approx 10$ km to $2\pi R_0 \approx 60$ km (defining the mesoscale, e.g. Tulloch et al., 2011).

To identify eddies, we use the *eddytools* python package documented in Rieck et al. (2025a). Eddies are detected using the Okubo-Weiss parameter (OW; Okubo, 1970; Weiss, 1991), which measures the relative importance of shear and strain to vorticity in the velocity field (Fig. 3a) :

$$\mathrm{OW} = (\partial_x \mathrm{u} - \partial_y \mathrm{v})^2 + (\partial_x \mathrm{v} + \partial_y \mathrm{u})^2 - (\partial_x \mathrm{v} - \partial_y \mathrm{u})^2 \tag{1}$$

where $\mathrm{u}, \mathrm{v}$ denote the velocities along the *x* and *y* directions of the grid, locally orthogonal. The resulting OW field (Fig. 3b) is compared to the local OW standard deviation ($\sigma_{\mathrm{OW}}(x,y)$) averaged over the full time period (Fig. 3c). $\sigma_{\mathrm{OW}}$ is computed over a $L_\sigma \times L_\sigma$ box, $L_\sigma$ being chosen small enough to capture the regional differences between e.g. the centre of the gyre and the boundary currents, but large enough so that $\sigma_{\mathrm{OW}}$ is not impacted by individual eddies. Eddies have to meet the following condition to be retained in the census:

$$\mathrm{OW}(x,y,t) < -\alpha \sigma_{\mathrm{OW}}(x,y), \tag{2}$$

where $\alpha$ is a threshold value typically chosen between 0.2 and 0.5 (Isern-Fontanet et al., 2003; Chelton et al., 2007; Pasquero et al., 2001). As we aim to detect any vortex-like feature that may develop in the Canadian Basin, including those that are not

materially coherent, we choose an Eulerian over a Lagrangian approach for detection. The OW-method is based on velocities (u, v) and thus preferable over SSH-based methods for detection in sea ice-covered areas where SSH-based detections are known to miss objects that do not have a surface expression. Additionally, the OW-method has the advantage of being computationally efficient and thus seems well-suited for a detection run for several decades. A comparison of the OW-based detection with those from Nencioli et al. (2010, u, v-based) and Chelton et al. (2011, SSH-based) was performed by Rieck et al. (2025a, see their

Fig. S3). They show that the OW-based method detects higher numbers of eddies compared to the other methods, mostly due to its capability to detect weak eddies, i.e. eddies with small rotational velocities and SSH anomaly. This detection bias towards weak eddies is commented in Section 4.

The detection is implemented using daily averaged model output in the Canadian Basin and is run for each vertical level of the model independently above $1,200$ m (which represents a lower bound of the AW layer), totalling $49$ levels. No 3D

representation of eddies is attempted here, as connecting the results between the vertical layers is not trivial and would require a substantial development of the detection and tracking algorithm. A brief evaluation of the vertical structure of eddies is however proposed in Section 3.1.1. Note that because $\sigma_{OW}$ is computed independently for each depth level, the minimum OW used to identify an eddy also varies with depth. In other words, at depths of intense mesoscale activity, the OW an individual eddy needs to overpass to be identified as such is higher. For each eddy, we estimate its radius with $R = \sqrt{area/\pi}$ even

though the eddies might be elliptical in shape. We set the smallest eddies that the algorithm detects to occupy 20 grid points, which correspond to equivalent circular eddies with a minimum of 5 grid points across the diameter. A 5-grid point diameter circular structure corresponds to an eddy of about 7.5 to 10 km radius, depending locally on the grid size of the model, which is the lower bound of $R_0$ over the deep part of the Canadian Basin (see Nurser and Bacon, 2013). Over the shelf, where $R_0$ is smaller ($\approx 2-5$ km), we only detect the largest of the mesoscale rotating features. Statistics presented here include all detected

features, but remain valid when filtering out eddies on the shelf, as the vast majority of eddies are detected over the continental slope and within the basins (not shown).

Sensitivity tests for $\alpha$ show that the vertical distribution of the metrics investigated (the mean eddy radius, duration, polarity $r_{C/T}$ i.e. the ratio of cyclones to total number of eddies, and a proxy for the vorticity $|\Omega|$, see Sect. 2.2.3), is robust to changes of $\alpha$ from 0.1 to 0.5. Yet, we note changes in the total number of detected features with slightly larger, weaker, and shorter

eddies for smaller $\alpha$ (Fig. S7 in the supplementary material). For our analyses, we choose $\alpha = 0.3$ as an intermediate value. The box length $L_\sigma$ over which to compute $\sigma_{OW}$ needs to be tuned to the main spatial scales of dynamical regimes in the basin. In other words, $L_\sigma$ should be small enough to capture the jet-like circulation along the Alaskan and Chukchi slopes that are about $200-300$ km large, and large enough to allow statistically relevant values of $\sigma_{OW}$. We found that $L_\sigma = 200$ km is a reasonable value to resolve the different dynamical regimes within the Canadian Basin. Similar to $\alpha$, sensitivity tests indicate

that changing $L_\sigma$ within the range $[50, 400]$ km does not modify the vertical distribution of the mean eddy radius, duration, vorticity, and polarity, although the total number of detected features varies (not shown). Overall, modifications of $\alpha$ and $L_\sigma$ impact the precise definition of particular eddies, but not the statistical properties of the eddy field.

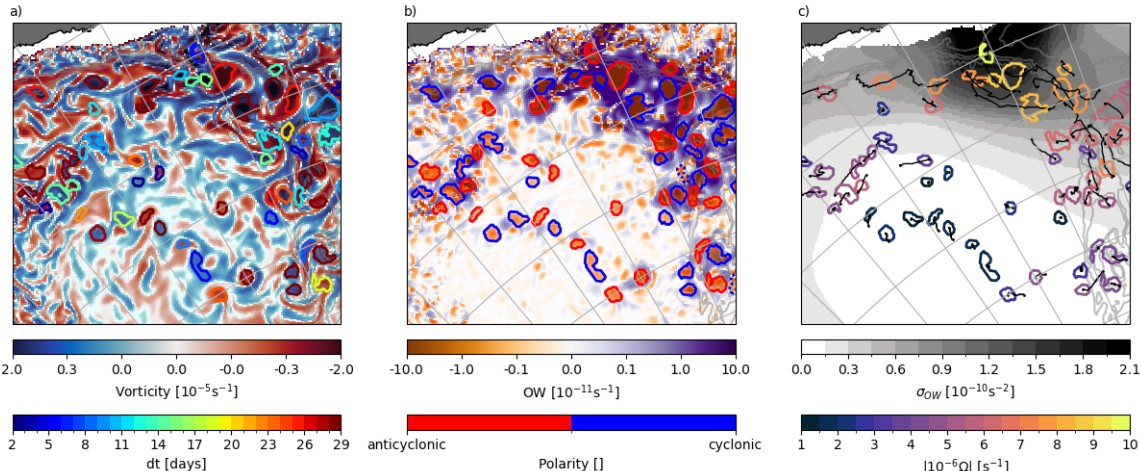

**Figure 3.** Example of detection and tracking of mesoscale eddies along the south-eastern edge of the Beaufort Gyre (BG) at 30 m on September 16th, 1996. Shown are: (a) the vorticity ($= \partial_x v - \partial_y u$), (b) the Okubo-Weiss (OW) parameter, and (c) the OW standard deviation $\sigma_{OW}$. Superimposed on each panel are contours indicating the eddies detected by the algorithm, coloured according to (a) their duration, (b) their polarity (red indicating anticyclones and blue cyclones), and (c) their intensity i.e. the absolute value of the difference between the vorticity in the centre of the eddy and the average vorticity along the edge of the eddy. Plain contours on (a), (b), and (c) indicate eddies lasting more than 2 days, dotted red and blue contours on (b) indicate eddies with a duration of one day. Black thin lines on (c) indicate the eddy trajectories. Thin gray lines on (a), (b), and (c) indicate the 100, 500, 1,000, and 1,500 m isobaths.

### 2.2.2 Tracking

Eddies are tracked over consecutive days using three main criteria: (i) their speed of propagation, (ii) their polarity, and (iii) their radius $R$ (Fig. 3c). For each eddy detected on day *t*, if an eddy with a similar radius (within $[0.5R, 1.5R]$) and the same polarity lies within a search radius $R_s$ on day *t+1*, it is chosen as a continuation of the track. In case there is more than one eddy matching these criteria, the one with the centre located the closest to the original eddy's centre is chosen. Results do not appear sensitive to the choice of a search radius $R_s \in [15, 53]$ km, and we choose a search radius $R_s = 22$ km corresponding to a propagation speed of $25$ cm s$^{-1}$, which is approximately the speed of the fastest simulated current within the domain, located along the Chukchi slope in summer. Location of eddies is obtained from their centre of mass - thus, the distance travelled by a given eddy between two consecutive days is possibly smaller than the grid resolution if an eddy is very slow. The radius, location, and grid points occupied by each individual eddy are detected every day. We assume eddies to be born (generated) the first time they are detected and to die (dissipated) the last time they are detected. However, the algorithm may occasionally

lose track of eddies, leading to a given eddy being counted as two successive eddies of similar properties. This interruption in the tracking generally occurs with weak features that are not well developed and thus not detected as eddies over consecutive days. We remove these eddies by filtering out any eddy that does not persist over at least two consecutive days. While this filtering does not fully overcome the interruption in the eddy detection – in particular if one eddy splits into two different ones, or equivalently if two eddies merge – it removes most of the issue and enables us to focus more on well-developed eddies. Still, the majority of eddies we detect are relatively weak and have a duration shorter than their turnaround time scale, defined as the time it takes for a water parcel to do a full revolution, $\tau = 2\pi/|\Omega|$ (i.e. an approximation of the expression suggested by Smith and Vallis, 2001, that is $\tau = 2\pi/\zeta_{rms}$, where $\zeta_{rms}$ is the root-mean-square vorticity). A discussion of the characteristics of the more vigorous and persistent eddies is proposed in Section 4.

### 2.2.3 Properties

For each detected and tracked eddy, we extract its intensity, which we define as the absolute vorticity amplitude of the eddy, i.e., the absolute value of the difference between the vorticity in the centre of the eddy and the average vorticity along the edge of the eddy ($|\Omega|$). We equivalently report its relative intensity $|\Omega|/f(\lambda)$ where $f$ is the Coriolis parameter computed as a function of latitude $\lambda$. By spatially averaging over the eddy area, we also extract its absolute salinity ($S$), conservative temperature ($T$), and the mean sea ice concentration $A$ and thickness $h$ above each eddy for each day.

Eddy properties are computed and tracked along the eddy pathway. Except where mentioned, properties are extracted at the eddy generation time. The properties of the eddy environment are defined by spatially averaging over a box that we take to be $n = 3$ times larger than the eddy dimensions in $x$ and $y$ directions (thus not of identical size along both directions) and from which we remove the eddy area. We note $\Delta X = X_i^{eddy} - X_i^{env}$ the anomaly of property $X$ at the time of eddy generation $i$. If two eddies develop next to each other, they will become each other's environment as we do not use a 2D eddy mask. In the interior of the basin, the gradients of density that may generate eddies are generally small, and so is the density anomaly of each eddy. To increase the robustness of the quantification of these anomalies, we choose to report only on the strongest anomalies. We define such *significant* anomalies in the following way. We first define the deviation $\delta X$ from the noise of the environment using the standard deviation of $X$ across the environment (i.e., excluding the eddy area), $\sigma_X^{env}$:

$$\delta X = \Delta X + \begin{cases} +\sigma_X^{env} \text{ if } \Delta X < 0 \\ -\sigma_X^{env} \text{ if } \Delta X > 0. \end{cases} \tag{3}$$

Then, the anomaly $\Delta X$ is said to be *significant* if it is of the same sign as $\delta X$, that is, if the anomaly is larger in absolute value than the standard deviation over the area.

Finally, the normalized amplitude of the seasonal cycle is defined for each property $X$ as:

$$SC_X = \frac{X_{max} - X_{min}}{\overline{X}}, \tag{4}$$

where the maximum, minimum, and mean are taken along the seasonal cycle.

| | Radius $R$ | Lifetime $dt$ | Distance travelled $D$ | Intensity $|\Omega|$ | Temperature anomaly $\Delta T$ | Salinity anomaly $\Delta S$ |
|---|---|---|---|---|---|---|
| $10^{\text{th}}$ perc. | 9.8 km | 2 day | 0.6 km | $5.7\ 10^{-7}$ s$^{-1}$ | $-0.10°$C | $-0.25$ g kg$^{-1}$ |
| $90^{\text{th}}$ perc. | 15 km | 21 days | 26 km | $1.1\ 10-5$ s$^{-1}$ | $0.20°$C | $0.24$ g kg$^{-1}$ |

**Table 1.** Eddy characteristics defined from the $10^{\text{th}}$ and $90^{\text{th}}$ percentiles of the distribution for all eddies at all depths above $1,200$ m. Percentiles of the temperature and salinity anomalies are computed on the subset of eddies that have significant anomalies, that is, on $\approx 15\%$ of the total eddy population.

## 3 Results

### 3.1 Characteristics of mesoscale eddies at annual and seasonal scales

#### 3.1.1 Across the Canadian Basin

Over $1995 - 2020$, and on average along the vertical, we detect and track about $6,000$ eddies per year in the Canadian Basin. This large number contrasts with the very few vortices detected from *in situ* observations below the ice ($\mathcal{O}(10)$ eddies per year, Cassianides et al., 2023; Zhao et al., 2014). However, it is closer to the numbers reported from satellite observations in the MIZ or the open ocean (up to $\mathcal{O}(1,000)$ eddies per year, Kubryakov et al., 2021; Kozlov et al., 2019; Manucharyan et al., 2022). Most of the eddies detected in the model have a radius similar to the Rossby radius of deformation ($\bar{R} = 12.1$ km), are short lived with an average duration $\bar{dt} = 10$ days and do not travel far with an average distance travelled $\bar{D} = 11.1$ km (Fig. 4a-c). Of all eddies detected, $49\%$ are cyclones. Cyclones and anticyclones have a similar intensity ($\overline{|\Omega|} = 4.6\ 10^{-6}$ s$^{-1}$ corresponding to a relative intensity of $\overline{|\Omega|/f} = 0.03$; Fig 4b). The eddy intensity, lifetime, and distance travelled have a standard deviation of the same order of magnitude as the mean. In particular, the distribution of the distance travelled shows three peaks in the distribution (Fig. 4b): a first one corresponding to quasi-stationary eddies, and two secondary ones centred around $4$ km and $8$ km. Only $15\%$ of the eddies show significant temperature and salinity anomalies with respect to their environment (Fig. 4e,f). The narrow and short tail of the statistical distribution of $\Delta S$ indicates that the overwhelming majority of detected eddies have properties close to the mean, while the wider distribution of $\Delta T$ indicate relatively large temperature anomalies for a significant portion of the eddy population (see box whiskers on Fig. 4 and the $10^{\text{th}}$ and $90^{\text{th}}$ percentiles in Table 1). Interestingly, eddies with properties at the tail of the distributions do not represent a distinct population of eddies. For instance, larger eddies do not live systematically longer (see Fig. S8 in the supplementary material).

So far, we have presented the statistics of the eddy properties aggregated over the whole $1995 - 2020$ period and over all depth levels above $1,200$ m, hence accounting for the same eddy several times if that eddy spans several depth levels. We observe some vertical coherency when looking at a few structures individually, in particular for structures spanning the pycnocline between $85 - 225$ m, surface intensified eddies, or eddies spanning the whole water column below $200$ m (Fig. 5).

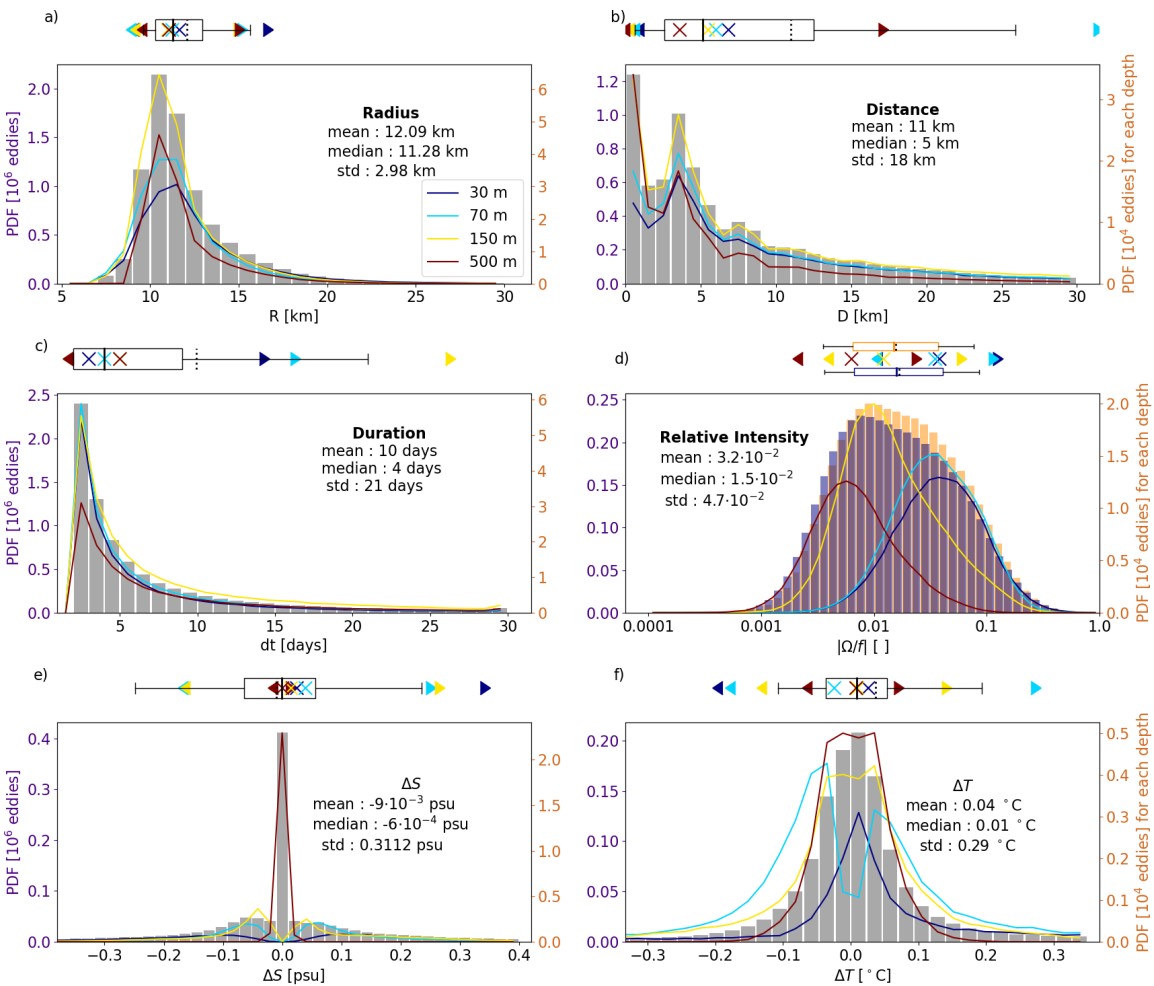

**Figure 4.** Histogram of the properties of eddies generated at all depths in the model: (a) radius, (b) distance travelled, (c) duration, (d) relative intensity for cyclones (blue) and anticyclones (orange), and anomalies in (e) salinity and (f) temperature with respect to the surrounding environment (see Sect. 2). All variables are estimated at the time of eddy generation, that is, the first time an eddy is detected. The number of eddies is reported in millions along the left axis (indigo). Anomalies are only accounted for when significant (see Sect. 2), that is, only $\approx 15\%$ of the eddy population is considered for panels (e) and (f). Box plots indicate the quartiles Q1 and Q3, the median (plain line) and mean (dotted line), and the $10^{-th}$ and $90^{-th}$ percentiles in the whiskers. Plain lines correspond to the histogram of properties at specific depths (11 m, 30 m, 69 m, 147 m, and 508 m), reported along the right axis in tens of thousands of eddies (orange). On panel d), plain lines report the histogram of absolute relative intensity, that is, for both cyclones and anticyclones together. Coloured ◄, ► and × respectively indicate the $10^{-th}$, $90^{-th}$ and median at the corresponding depth.

This vertical structure is similar to the vertical structure obtained from observations (Carpenter and Timmermans, 2012; Zhao and Timmermans, 2015) or predicted from baroclinic instability estimate in the BG (Meneghello et al., 2021). This coherency reflects in the statistics of the characteristics of eddies with significant differences across depth (see the coloured plain lines and ◄, ► and + in Fig. 4). Within the top $1,200$ m, the total number of eddies generated at each model depth level remains roughly constant between the surface and 85 m, and below 225 m, but increases by two-thirds between 85 m and 225 m (Fig. 6a). We also note important transitions in the ratio of anticyclones versus cyclones, radius, and eddy durations around these depths, suggesting different mechanisms of formation and dissipation. On average across the basin and over the 26 years, these transition depths correspond to the depth at which the sPW ($\sim$85 m), and the wPW ($\sim$225 m, see Fig. 6g,h) are found, forming together the pycnocline layer.

Based on the evolution of the statistical properties with depth, together with the observation of the coherent structures with finite depth extent, we thus define three layers: the upper layer ($0 - 85$ m), the pycnocline layer ($85 - 225$ m), and the AW layer ($225 - 1,200$ m). Next, we describe the eddy properties and discuss their formation and dissipation within each of these three layers. The results presented in the following are robust to the exact definition of the layer boundaries ($\pm 2$ model depth levels).

### 3.1.2 Within the upper layer

Within the top 85 m, about $6,000$ eddies are detected every year. The properties of these eddies show a marked seasonal cycle (Fig. 6), mainly following the seasonal cycle of the sea ice cover. From winter to summer, the number of eddies increases by a factor of 10 with a minimum in April, just before the onset of sea ice melting, and a maximum in September when sea ice is at its minimum (Fig. 6a). The Mixed Layer (ML) depth – computed as the depth at which the potential density has increased by $0.01\,\mathrm{kg\,m^{-3}}$ compared to the potential density at 1 m – decreases from 35 m in January-May to 3 m in July, and increases from August to December, when it recovers 30 m (see also the stratification on Fig. 6h). The change in stratification associated with the ML depth delimits different regimes of variations for the mean radius, polarity, and intensity. Within the ML, the averaged radius of eddies increases from 12 km in early summer to 14 km in fall, while below the ML, the averaged radius of eddies barely changes (increases from 12 km to 12.5 km; Fig. 6b). In winter, a dominance of anticyclones is found at the surface ($0 - 10$ m) and of cyclones just below ($10 - 40$ m, i.e., to the base of the ML), while at the very surface, eddies are essentially anticyclonic year-long (Fig. 6d). Below the base of the ML, the proportion between cyclones and anticyclones remains more equally distributed throughout the year, with about $55\%$ anticyclones. Within the top 85 m, the most intense eddies are found during the stratifying and de-stratifying periods, corresponding to the onsets of sea ice growth (October and November) and to the melt season (May-July), respectively. In winter, intense eddies are also found at the base of the ML (Fig. 6c). Eddies persist longer in summer ($7 - 8$ days) than in winter when their lifetime is reduced by about half (Fig. 6e). Lifetimes likely influence the distance travelled, with eddies propagating over $15 - 16$ km in summer and $7 - 8$ km in winter on average (Fig. 6f). Lifetime for the vast majority of eddies ($85\%$) is significantly shorter than the theoretical mean turnaround time. Thus, part of the detected eddies are likely not fully developed and belong to the "turbulent soup" that is generated in response to the surface density gradients and gets quickly dissipated by sea ice in winter. These eddy lifetimes are similar to the characteristic times of spin-down through sea ice dissipation when sea ice is taken as the main drag (e.g. $\lesssim 4$ days, Meneghello et al., 2021;

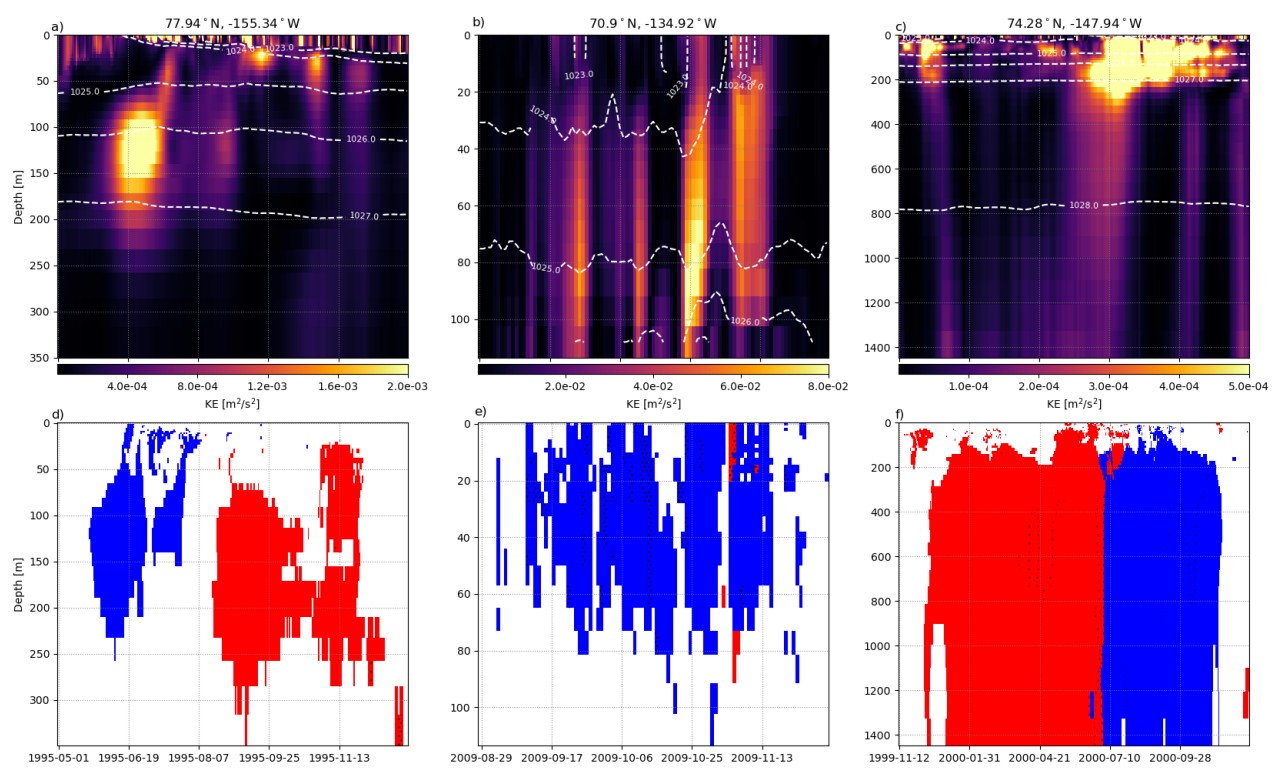

**Figure 5.** Daily averaged total kinetic energy (upper row) of a *virtual* mooring located at $77.94°$N, $155.34°$W (first column), at $70.9°$N, $134.92°$W (second column), and $74.28°$N, $147.94°$W (last column). Dashed white lines correspond to surface-referenced isopycnals ($\sigma_0$). Polarity of eddies passing by the virtual mooring (second row) and identified by our algorithm with cyclones in blue and anticyclones in red. White means no eddy detected. Dots (.) indicate an interruption in the tracking, meaning that the algorithm identifies a newly born eddy.

Pedlosky, 1982). We come back to this point in Sect. 4.

Eddy properties present large spatial variations in the surface layer of the Canadian Basin (Fig. 7). In particular, there is a
strong contrast between the continental slope and the deep basin, with up to 10 times larger density of eddy population over the
slope ($\approx 150$ km wide, Fig. 7a). The greater generation of eddies over the slope peaks in October, when sea ice extent is close
to its minimum and winds start to increase (see Fig. S9 in the supplementary material). While the production of eddies in the
deep basin remains low on average, it becomes similar to the production over the slope when sea ice concentration drops below
$\approx 80\%$, that is, between July and November, depending on the latitude (see Fig. S9a). Simultaneously, eddy lifetime increases
from evanescent ($dt = 1 - 3$ days) below the pack in winter to about 15 days in summer on average (Fig. S9b). Thus, over the
domain, eddy lifetime is mainly enhanced where the ice concentration is lower than $15\%$ (not shown).

Over the slope, eddies have, on average, a positive temperature anomaly ($\approx 0.3°$C) with the exception of some anomalously
cold eddies forming over the Chukchi shelf. Where the mean September sea ice concentration is higher than $15\%$, eddies do
not have a temperature anomaly, aligning with Cassianides et al. (2023)'s detection of a majority of vortices with no significant
temperature anomaly. We also find a contrast in radius between the eastern and western sides of the gyre, especially off-shore
the Chukchi shelf break and above Northwind ridge, where eddies are found to be about $60\%$ larger (Fig. 7b). These eddies
are carried within the intense anticyclonic BG circulation (Fig. S5 in the supplementary material) and therefore travel up to 40
km throughout their lifetime, a distance much larger than the averaged distance travelled of 8 km at the centre of the basin.

Anticyclones are predominant over the centre of the gyre, while over the slope, a greater proportion of cyclones are found
(Fig. 7d). Vorticity anomalies within the BG do not indicate any preference for the generation of anticyclones (not shown).
One hypothesis that explains this cyclone/anticyclone asymmetry, which has been formulated for the Mediterranean Sea and
more generally in other contexts of turbulent flows, is that anticyclones are more persistent than cyclones that tend to split into
smaller objects, leading to anticyclones being more systematically identified in eddy censuses (Stegner et al., 2021; Giulivi and
Gordon, 2006). Beech et al. (2025) further suggests the role of sea ice in preferentially dissipating small cyclones. Whether
this applies to the BG is worth future investigation. We speculate that such a filtering mechanism might mainly apply in the
centre of the gyre, where mean currents are negligible, and the turbulent field can freely develop, while strong mean currents
that generate and dissipate eddies are likely to be the dominant factor in determining eddy polarity near the gyre's edges.

Up to 300 km off the shelf, high density in the eddy population is accompanied by intensities in the eddy field up to one
order of magnitude higher than in the deep basin (Fig. 7c). This is visible all along the shelf break of the domain, from the
Chukchi shelf break to the Canadian Arctic Archipelago shelf break. The most intense eddies are found at the mouth of the
McKenzie River and at Pt. Barrow (not shown), being respectively fresher and saltier than their environment (Fig. 7e,f). Along
the Alaskan and Chukchi slopes, on both sides of Pt. Barrow, eddies with positive salinity anomalies are detected in the inner
part of the slope, while eddies with negative anomalies are detected in the outer part. This pattern illustrates the penetration of
the Pacific Waters from Pt. Barrow along the baroclinically unstable Alaskan coastal and Chukchi Slope currents (Corlett and
Pickart, 2017; Spall et al., 2008), and supports observations of the penetration of eddies associated with a salty anomaly into the
BG at Pt. Barrow (MacKinnon et al., 2021, in the submesoscale range). Additionally, eddies associated with a fresh anomaly

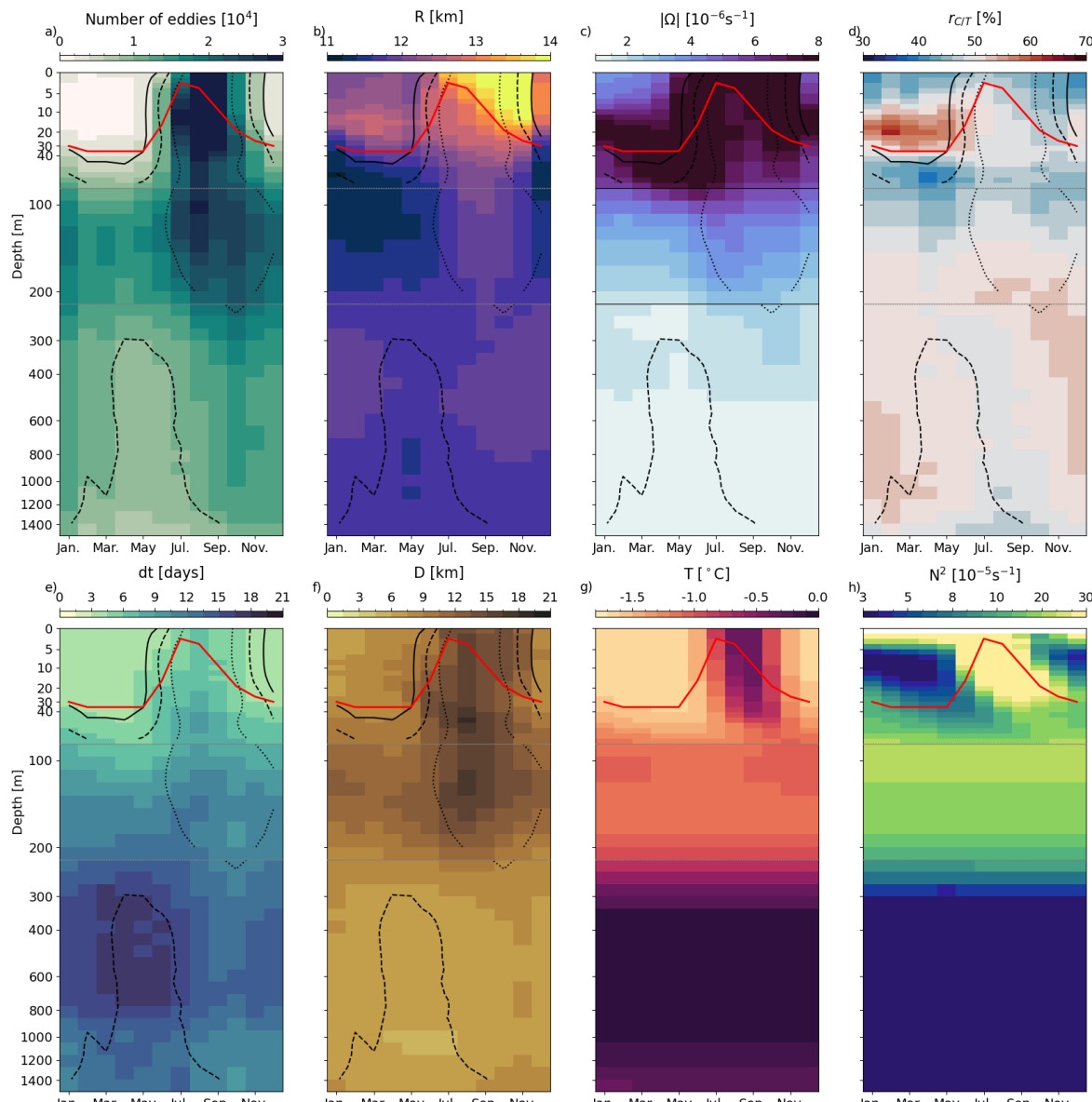

**Figure 6.** Seasonal cycle of eddy and basin properties with depth. (a) The number of eddies generated at each model depth level and month summed over the 26 years of simulation, and associated averaged properties: (b) radius, (c) intensity, (d) polarity, (e) lifetime, (f) distance travelled. Basin-averaged (g) potential temperature and (h) stratification ($N^2$). For panels b-h, properties are averaged at each model depth level and month over the 26 years of simulation. For panels a-f, dotted, dashed, and plain lines indicate the isocontours corresponding to $5,000$, $10,000$, and $20,000$ eddies as calculated in (a), thus indicating where the statistics might be less robust due to the lower number of eddies. Note the use of a non-linear vertical axis to highlight the variability in the upper layer. Dotted horizontal grey lines delineate the three layers introduced in Section 3: upper layer, pycnocline layer, and Atlantic Waters (AW) layer. The red line indicates the Mixed Layer (ML) depth computed from a potential density threshold referenced to 1 m of $0.01 \, \text{kg m}^{-3}$.

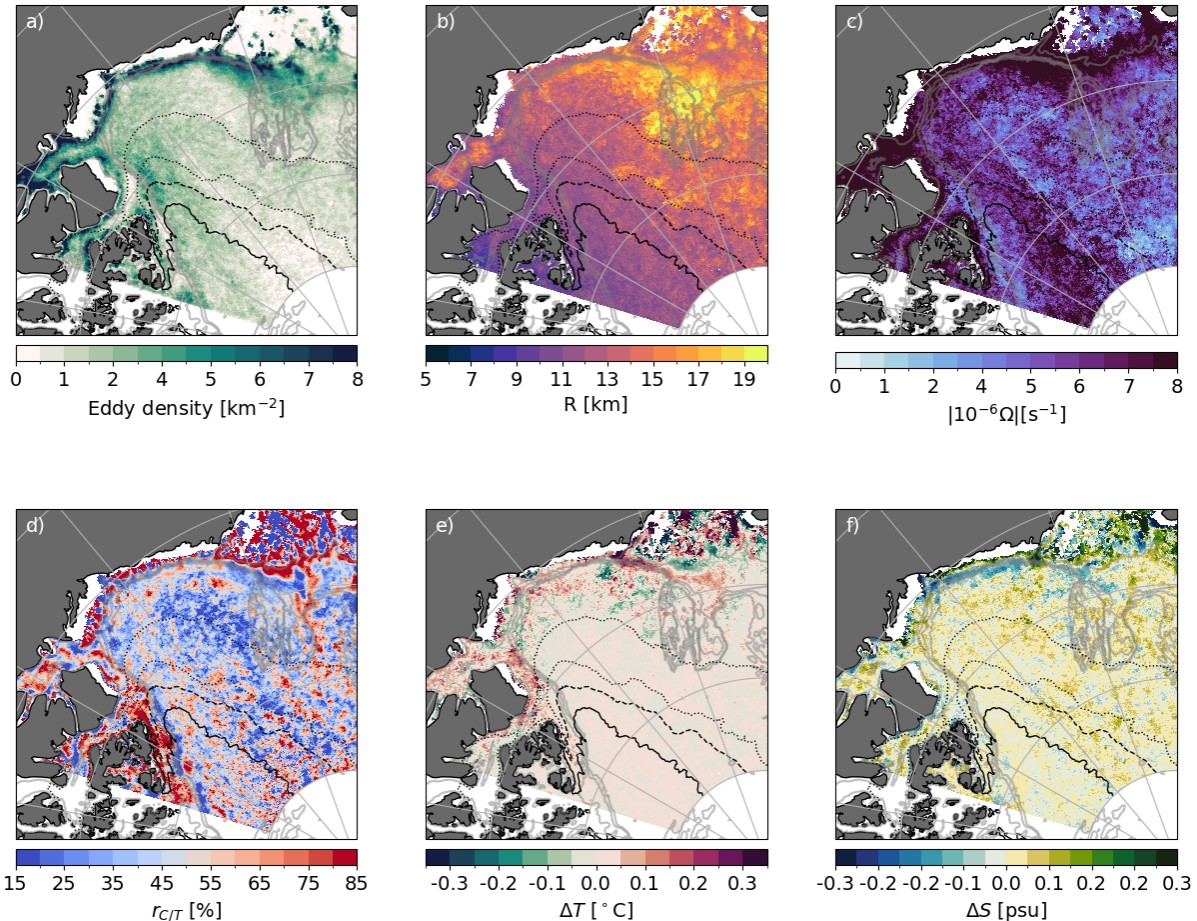

**Figure 7.** Eddy properties at 30 m (i.e., within the upper layer) over the 26 years of simulation. (a) Density of the eddy population (i.e., number of individual eddies detected per km$^2$) and associated properties: (b) averaged radius, (c) intensity, (d) polarity, and anomalies of (e) temperature and (f) salinity with respect to the environment of eddies. Temperature anomalies are calculated with respect to the local freezing temperature. Temperature and salinity anomalies are only accounted for when anomalies are significant (which represents about 15% of all eddies, see Sect. 2). All variables are extracted at nominal depth 30 m, and summed (panel a) or averaged (panels b-f) over the 26 years of simulation. Note that all fields show similar structures at all depths between 0 and 85 m except for the radii that are significantly larger within the Mixed Layer (ML, see Fig. 6c). Plain, dashed, dotted, and loosely dotted black lines show respectively the 90%, 80%, 50%, and 15% contours of the climatological September sea ice concentration. Gray lines show the 100, 500, 1,000, 1,500 m isobaths.

found along the outer part of the slope confirm the role of fresh water input from McKenzie River in generating instabilities that develop into eddies propagating downstream along the anticyclonic circulation (Kubryakov et al., 2021).

### 3.1.3 Within the pycnocline layer

Over the 26 years of simulation, there are about $30\%$ more eddies detected in the pycnocline layer ($\sim 85 - 225$ m) than in the upper layer ($9{,}000$ on average in the pycnocline layer vs $6{,}000$ eddies on average in the upper layer per year; Fig. 6a). Eddies detected within the pycnocline layer are evenly distributed between cyclones and anticyclones (Fig. 6d) and are smaller and weaker than in the upper layer on average (mean radius decreases from 12.4 km to 11.6 km and intensity decreases from $7.4\ 10^{-6}$ s$^{-1}$ in the upper layer to $3.3\ 10^{-6}$ s$^{-1}$, Fig. 6b,c). Although weaker, eddies in the pycnocline layer last about 6 days longer than in the upper layer, likely due to the absence of ice or air drag to dissipate eddies through friction (Fig. 6e). Despite this increased longevity, the mean distance travelled by eddies in the pycnocline layer only increases by about 1.1 km compared to the upper layer (Fig. 6f), presumably because of the weaker background flow advecting these eddies. Therefore, of the eddies generated over the slope, only the strongest and longest-lived eddies may be able to travel far enough to reach the gyre and could thus participate in the transport of heat, salt, and nutrients from the continental shelf to the deep basins (see Sect. 4).

In the pycnocline layer, eddy characteristics show a weaker seasonality compared to the upper layer. Quantitatively, the normalized amplitude of the seasonal cycle in eddy number diminishes with depth, from $SC_N = 2.3$ at 30 m to $SC_N = 0.5$ at 150 m (see Sect. 2 for a definition of $SC_X$). The other properties also show a decreased normalized amplitude of their seasonal cycle compared to the upper layer, by $\approx 60\%$ for the radius, $50\%$ for the intensity and distance travelled, and by $40\%$ for the lifetime (refer to Table 3 for detailed modifications of the seasonal cycles). This damped seasonality is expected as the pycnocline shields eddies from dissipation by sea ice.

For most eddy characteristics (radius, intensity, duration, distance, and polarity), the spatial distribution within the pycnocline layer is generally similar to that of the upper layer (compare Fig. 7 with Fig. 8) but persists throughout the year due to the absence of seasonal variability. The similarity in spatial distribution with the upper layer is expected as the anticyclonic circulation that dominates most of the region investigated extends down to the pycnocline associated with the wPW (see Fig. S5 in the supplementary material, see also Planat et al., 2025). However, the spatial distribution of the density in eddy population is notably different between the pycnocline and the upper layers along the southern edge of the BG (Fig. 8a). There, a strong reduction in the density of the eddy population is found compared to the shelf and deep basin (Fig. 8a). This reduced density compared to the upper layer occurs despite the eddies being relatively intense, long-lived, and travelling relatively far along the anticyclonic flow (Fig. 8c,e,f). We suggest that the inner part of this local reduction in eddy generation is linked to a stabilizing effect of the continental slope. The growth of instabilities is known to be hampered over regions where the ratio of the continental slope to the isopycnal slopes is greater than 1 (Manucharyan and Isachsen, 2019), as is the case for the slope of the BG in the model (not shown, see Regan et al., 2020). However, this reduction occurs up to 250 km away from the shelfbreak. There, we observe diminished background potential vorticity gradients (not shown, see Fig. 9 from Meneghello et al., 2021) associated with diminished baroclinic instabilities, which offer an alternative explanation for the extended area with muted eddy activity.

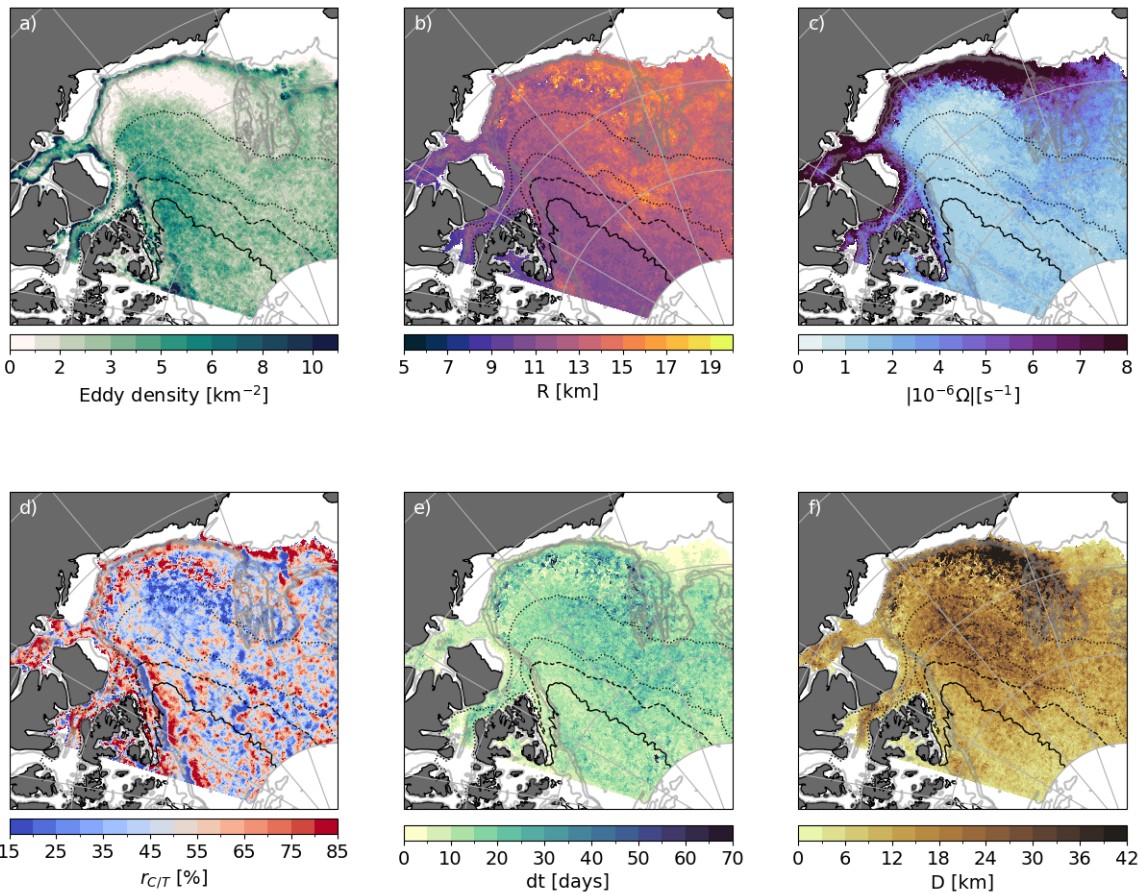

**Figure 8.** Eddy properties at 150 m (i.e., within the pycnocline layer) over the 26 years of simulation. (a) Density of the eddy population (i.e., number of individual eddies detected per km$^2$) and associated properties: (b) averaged radius, (c) intensity, (d) polarity, (e) lifetime, and (f) distance travelled. All variables are extracted at nominal depth 150 m, and summed (panel a) or averaged (panels b-f) over the 26 years of simulation. Note that all fields show similar structures at all depths between 85 and 225 m. Plain, dashed, dotted, and loosely dotted black lines show respectively the 90%, 80%, 50% and 15% contours of the climatological September sea ice concentration. Gray lines show the 100, 500, 1,000, 1,500 m isobaths.

### 3.1.4 Within the AW layer

Below the pycnocline layer and down to $1,200$ m, within the AW layer, the total number of eddies over the $26$ years of simulation decreases by $37\%$ compared to the pycnocline layer (from on average $9,000$ per year in the pycnocline layer to $5,500$ eddies per year in the AW layer; Fig. 6a). This decrease is reduced to $20\%$ if one compares eddy density, as the area where eddies developed is reduced due to the bathymetry. Because the layer is located below the pycnocline, the seasonal variability in eddy properties is almost completely shut down (Fig. 6). In that layer, we find eddies similar in radius ($\approx 11.8$ km, Fig. 6b) but weaker in intensity than in the pycnocline layer ($1.5\,10^{-6}$ s$^{-1}$ compared to $3.3\,10^{-6}$ s$^{-1}$, Fig. 6c). The distance travelled by eddies decreases from $12.2$ km to $7$ km (Fig. 6f), and the polarity remains equally shared between cyclones and anticyclones (Fig. 6d). We note that the averaged lifetime of eddies is longer than in the pycnocline layer ($14$ days compared to $11$ days) but remains small despite the few processes that could dissipate eddies at this depth. This relatively short lifetime may point to the fact that most of the eddies detected in this layer do not fully develop (if we consider their turnaround time). Of all the eddies detected in that layer, only $6\%$ persist longer than their turnaround time. These long-lasting eddies may live up to $150$ days ($99^{\text{th}}$ percentile), surpassing all the maximum durations detected in the other layers and matching estimates of weeks to years made from observations (Hunkins, 1974; Timmermans et al., 2008, although these estimates are largely uncertain as ITPs and moorings only capture a portion of the eddy trajectory). We refer the reader to Sect. 4 for a discussion on long-lasting eddies.

Within the AW layer, eddy properties show different patterns compared to the layers above. Significant differences are expected given that the mean circulation of that layer departs strongly from that above (Fig. S5 in the supplementary material). In particular, eddies are predominantly generated over the continental slope along the path of the AW cyclonic boundary current (Fig. 9a). A smaller density of eddies is generated in the rest of the domain, with some hot spots of high density in eddy population located close to the northern boundary of our domain. The latter corresponds to short-lived eddies, and we discuss more extensively the "turbulent soup" formed by short-lived eddies in Sect. 4. Along the shelfbreak and boundary currents, the eddy intensity is larger by up to one order of magnitude compared to the rest of the domain, where eddies are notably weaker (Fig. 9c). No clear spatial pattern in polarity arises at the scale of the basin, except along the shelf breaks of the Chukchi Sea and Canadian Archipelago, where anticyclones tend to dominate in the inshore part of the current while cyclones dominate in the offshore part of the current (Fig. 9d). Off the western flank of Northwind ridge are found the largest (up to $20$ km), farthest-reaching (up to $40$ km) and longest-lived (up to $60$ days) eddies (Fig. 9b,e,f). The EKE is one order of magnitude larger in that area than within the deep basin and displays hotspots in the form of large structures detaching from the cyclonic boundary current that hugs Northwind ridge (not shown). In this region, there are large uncertainties in the literature on the exact path of the AW, as AW are thought to intermittently detach from the slope-intensified cyclonic boundary current, or alternatively flow along double boundary currents (McLaughlin et al., 2009; Li et al., 2020; Planat et al., 2025; Karcher et al., 2012; Lique et al., 2015). We suggest here some instabilities in the cyclonic boundary currents associated with the generation of large eddies.

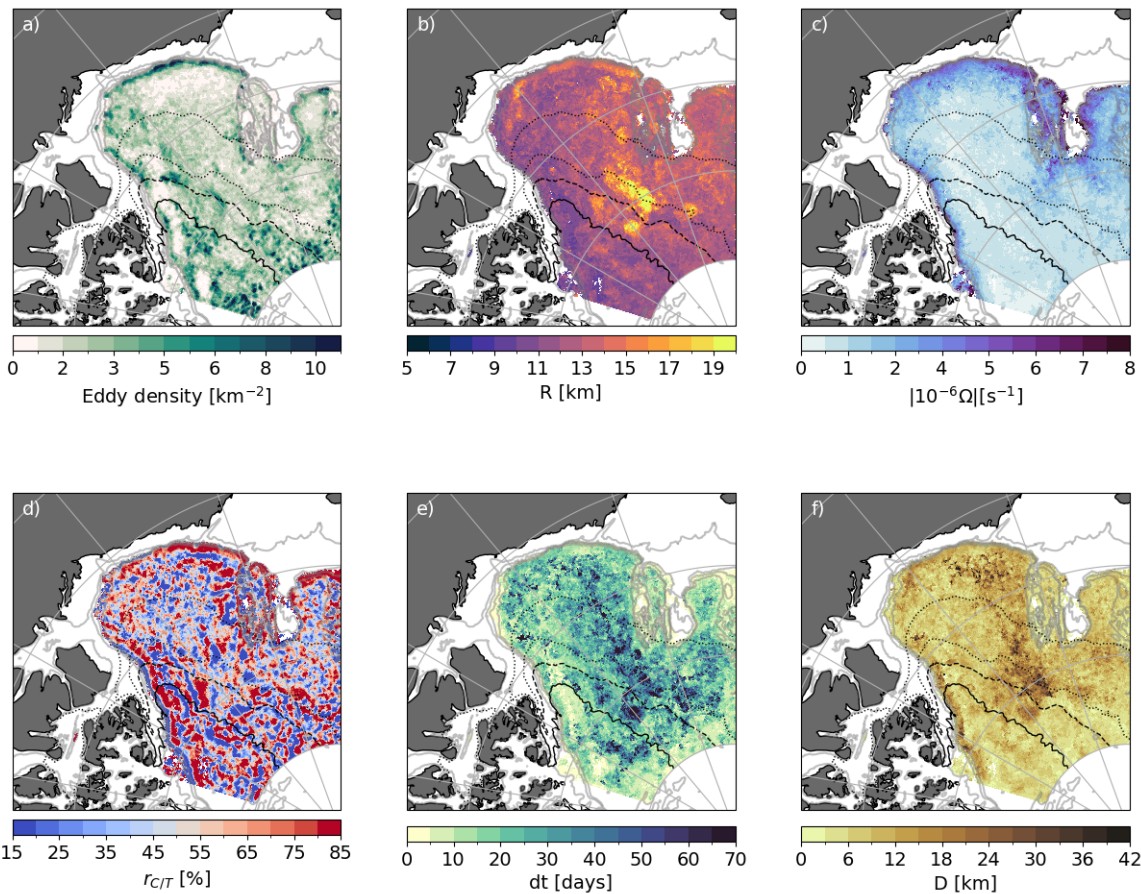

**Figure 9.** Eddy properties at 500 m (ie., within the AW layer) over the 26 years of simulation. (a) Density of eddy population (i.e., number of individual eddies detected per km$^2$) and associated properties: (b) averaged radius, (c) intensity, (d) polarity, (e) lifetime, and (f) distance travelled. All variables are extracted at nominal depth 500 m, and summed (panel a) or averaged (panels b-f) over the 26 years of simulation. Note that all fields show similar structures at all depths between 225 and 1,200 m. Plain, dashed, dotted, and loosely dotted black lines show respectively the 90%, 80%, 50%, and 15% contours of the climatological September sea ice concentration. Gray lines show the 100, 500, 1,000, 1,500 m isobaths.

## 3.2   Evolution of the population of eddies over $1995 - 2020$

Over the last two decades, both the sea ice cover and the mean circulation in all layers in the Canadian Basin have changed drastically. Indeed, the sea ice extent has shrunk with a trend of $-12.7\%$ (Meier and Stroeve, 2022), while an acceleration of the anticyclonic circulation of the BG has been found to occur around 2007 in both observations (Giles, 2012; Regan et al., 2019) and models (Regan et al., 2020) associated with a decrease in the ice cover and and increased Ekman pumping, with intensified winds (Meneghello et al., 2018). Our model represent the decadal change in sea ice cover and the fast-accelerating

period of the BG (Fig. 10a). Over the 26 years of simulation, the sea ice extent of the Canadian Basin decreases by $55\%$ over the 26 years of simulation, while the mean gradient of SSH over the Canadian Basin, which is a proxy for the intensity of the mean circulation at the surface, show a step increase of $0.15$mm km$^{-1}$ between 2006 and 2008, for an overall increase of $16\%$ ($\%$ changes are computed using averages over the first and last 5 years of the simulation). Here, we report on the changes in the population of eddies between 1995 and 2020 in light of the modifications of these two forcing fields.

Over $1995 - 2020$, the number of eddies generated in the Canadian Basin increases in all layers, by $34\%$ in the upper layer, $45\%$ in the pycnocline layer, and $40\%$ in the AW layer (Fig. 10). Increases in eddy number are also found when looking regionally at the Beaufort Box (BB, see Sect. 2 for the definition) and the Alaskan shelf area at all depths; with the exception of the AW layer, which shows an overall decrease in the eddy number in the BB of $26\%$. However, a rebound in the eddy

number is seen around 2017, suggesting a possible lagged increase in the eddy number in that layer. A key difference between the Alaskan shelf and BB is that the latter is primarily energized above the ML ($+94\%$ above 30 m *vs* $-15\%$ between 30 and 85 m) while the former presents an increase in the number of eddies that is roughly constant with depth throughout the upper layer. This difference may be explained by the greater energy input in the Alaskan area that becomes seasonally ice-free earlier than the interior of the basin, where little additional energy linked to the sea ice decline can therefore penetrate the water

column.

Along with the increase in the eddy population, between the first and last 5 years of the simulation, eddies become bigger ($+0.7$ km), travel further ($+2.2$ km) and carry relatively warmer waters ($+0.0027°$C; Table 2). These changes are in line with an increased stratification, which increases the Rossby radius. We estimate a change of $+0.5$ km for the Rossby radius in the BB between the first and last decade of the simulation. This increase in the Rossby radius enhances the effective resolution of

the model, potentially leading to a higher number of eddies detected. Yet, the change of effective resolution, defined as $R_0/ds$ (where $ds$ is the maximal grid spacing), is only significant in the northeastern side of the domain (not shown), a region where we detect overall very few vortices (for instance, see Fig. 7).

In the upper layer, eddies last longer ($+0.6$ days), most probably in relation to the reduced impact of sea ice. In both the AW layer and the pycnocline layer, the eddy intensity is increased. This is presumably due to the faster mean circulation, with in-

creased MKE and EKE in all layers (not shown). Polarity remains unchanged through the 3 decades. A detailed comparison of the histograms for each property between the first and last 5 years of the simulation is shown in Fig. S10 of the supplementary

material.

An increase in the number of eddies is expected as sea ice shrinks, and is in line with recent literature reporting on an in-
crease in energy over the Arctic (Regan et al., 2020; Li et al., 2024; Manucharyan et al., 2022; Armitage et al., 2020). Yet, it
remains unclear whether this increase is due only to the expansion of the open ocean and the MIZ, or also to increased levels
of energy within the MIZ and pack ice linked to the sea ice becoming less concentrated, thinner, and more mobile (Kinnard
et al., 2011; Kwok, 2018; Rampal et al., 2009) and to changes in atmospheric forcing. If the former applies, then the density of
the eddy population within the open ocean should remain constant. To investigate this question, we look at the density of the
eddy population within three sea ice regions in the upper layer of our domain: the pack ice (where ice concentration is $\geq 80\%$),
the open ocean (where ice concentration is $\leq 15\%$), and the MIZ, which lies in between. For a given year and region, the eddy
population density is calculated as the total number of individual eddies detected for that year over the mean sea ice area for
that year in that region. Over the 26 years, we find an increase of $10\%$ and $20\%$ of the density in eddy population in the open
ocean and MIZ, respectively, and no change on average in the pack ice (Fig. S11 in the supplementary material). In the MIZ and
open ocean, the increase in eddy generation is mainly a step increase in 2008, with reduced (shut down) interannual variability
in the MIZ (open ocean) in the following years. This enhancement in the density of the eddy population presumably results
from the additional energy penetrating into the ocean in the recent state of the BG. This is expected to continue in the future as
suggested by (Li et al., 2024). In the pack ice, the constant density results from opposing changes with an increase above the
ML of $10\%$ and a decrease of similar amplitude below, with strong year to year variability (see Fig. S11 in the supplementary
material).

Overall, our results suggest that, in the upper layer, the number of eddies do not only increase because of an expansion of the
open ocean area at the expense of sea ice, but also because of an energy surplus in the Canadian Basin in the MIZ and open
ocean. The increase of energy below the pack ice, suggested in future projections of the Arctic (Rieck et al., 2025b), is only
seen in the upper ML in the current climate.

On interannual time scales, the BB, the Alaskan shelf area, and the whole Canadian Basin show important variability in
the number of eddies detected in the upper layer. Significant correlations with the ice cover, either with the September sea ice
extent, or with the yearly cumulated area with ice less concentrated than $80\%$, are only visible at the surface (down to $\approx 20$ m,
that is approximately down to the ML depth), at the exception of the Alaskan area where they are found significant deeper (not
shown). In both the upper and pycnocline layers, the number of eddies starts rising around 2000 before culminating in 2008
($+54\%$ between 2006 and 2008 in the pycnocline layer, Fig. 10b,c). The increase in the number of eddies that peaks in 2008
in our model bears similarity to the increase in EKE that was reported by Regan et al. (2020) to occur over one year following
the gyre acceleration in 2007 and the low sea ice record of that year. The authors suggest that beyond $2007 - 2008$, the BG
is able to expand spatially over Northwind ridge and thus the need for eddies to release the accumulated potential energy is
reduced. The transient increase in the number of eddies reported in our analysis thus tends to confirm this hypothesis, with a
slightly longer equilibration time. In the pycnocline layer of the BB, the increase in the number of eddies in 2008 persists for

| | R [km] | D [km] | dt [days] | $|\Omega|$ $[10^{-7}\mathrm{s}^{-1}]$ | $\Delta S$ | $\Delta T$ | $r_{C/T}$ [%] |
|---|---|---|---|---|---|---|---|
| Upper layer (0 − 85 m) | 1 ; 8% | 2.4 ; 21% | 0.6 ; 10% | -2.1 ; -3% | -0.008 | 0.004 | -0.7 |
| Pycnocline layer (85 − 225 m) | 0.5 ; 4% | 2.0 ; 18% | -1.3 ; -11% | 7.9 ; 26% | 0.003 ; | 0.004 ; | -0.06 |
| AW layer (225 − 1,200 m) | 0.4 ; 4% | 2.0 ; 30% | -1.4 ; -9% | 7.4 ; 57% | -2.5e-5 | 0.0009 | 3 |
| All (0 − 1,200 m) | 0.7 ; 6% | 2.2 ; 22 % | -0.5 ; -5% | 3 ; 7 % | 0.0003 | 0.0027 | 0.8 |

**Table 2.** Changes in mean eddy properties for each layer when comparing the last 5 years of the simulation and first 5 years. Red indicate an increase, blue a decrease. Also reported are relative increases in % for the mean radius, distance, duration and intensity.

an additional couple of years, in contrast to the Alaskan shelf area and basin average, where the increase remains punctual. Further down, the AW layer also displays a weaker increase in the number of eddies in the BB (Fig. 10d). This increase at depth suggests a direct top-down coupling between the BG and the AW layers, despite the insulation of the pycnocline, as
documented by Lique and Johnson (2015) and Lique et al. (2015) for the mean circulation of the AW layer. The processes sustaining the increase in the number of eddies within both the pycnocline and AW layer, for 5 years, following the increase of the BG remain to be explored. Additionally, we note that the number of eddies generally keeps increasing after 2016, while the gyre intensity declines. Our simulation lacks a few more years to interpret the observed changes.

Within the AW layer, little interannual variability in the number of eddies is found in contrast to the layers above (Fig. 10d).
However, when looking at the long-lasting eddies (6% of the total population), we do see important year-to-year variability (see Fig. S12 in the supplementary material). These more persistent features show a large relative increase (300%) starting around 2011, which is mainly due to an increase in the number of eddies located east of Northwind ridge. These structures, which are large (30 − 50 km), develop in particular in the late years of the simulation (2012 − 2020) when the cyclonic boundary current on the Northwest flank of Northwind ridge reverses to an anticyclonic flow with increased horizontal shear (not shown). What
drives these changes in the AW mean pathways, and how the latter influence the generation of large eddies is beyond the scope of this paper and is left for future analysis.

## 4    Discussion and conclusion

In this study, we apply an eddy detection and tracking algorithm to the output of a high-resolution regional model of the Arctic to document the characteristics of mesoscale eddies in the Canadian Basin and examine the evolution in the number and
characteristics of eddies over 1995 − 2020. Over that period, we report an average of about ,000 eddies generated per year in the surface layer, about 9,000 eddies in the pycnocline layer, and about ,500 in the AW layer of the Canadian Basin. Most of these eddies are found to be the size of the Rossby radius of deformation (mean eddy radius of about 12.1 km), stationary (distance travelled of about 11.1 km), and short-lasting (lifetime of about 10 days; Fig. 4). The distribution between cyclones and anticyclones is about equal in the investigated domain. In addition, the majority of eddies do not have a *significant* temperature
nor salinity anomaly relative to their environment, where *significant* only accounts for the strong anomalies. Nevertheless, some non-negligible anomalies are visible along the shelf in the surface layer (Fig. 7e,f). All the documented properties (radius,

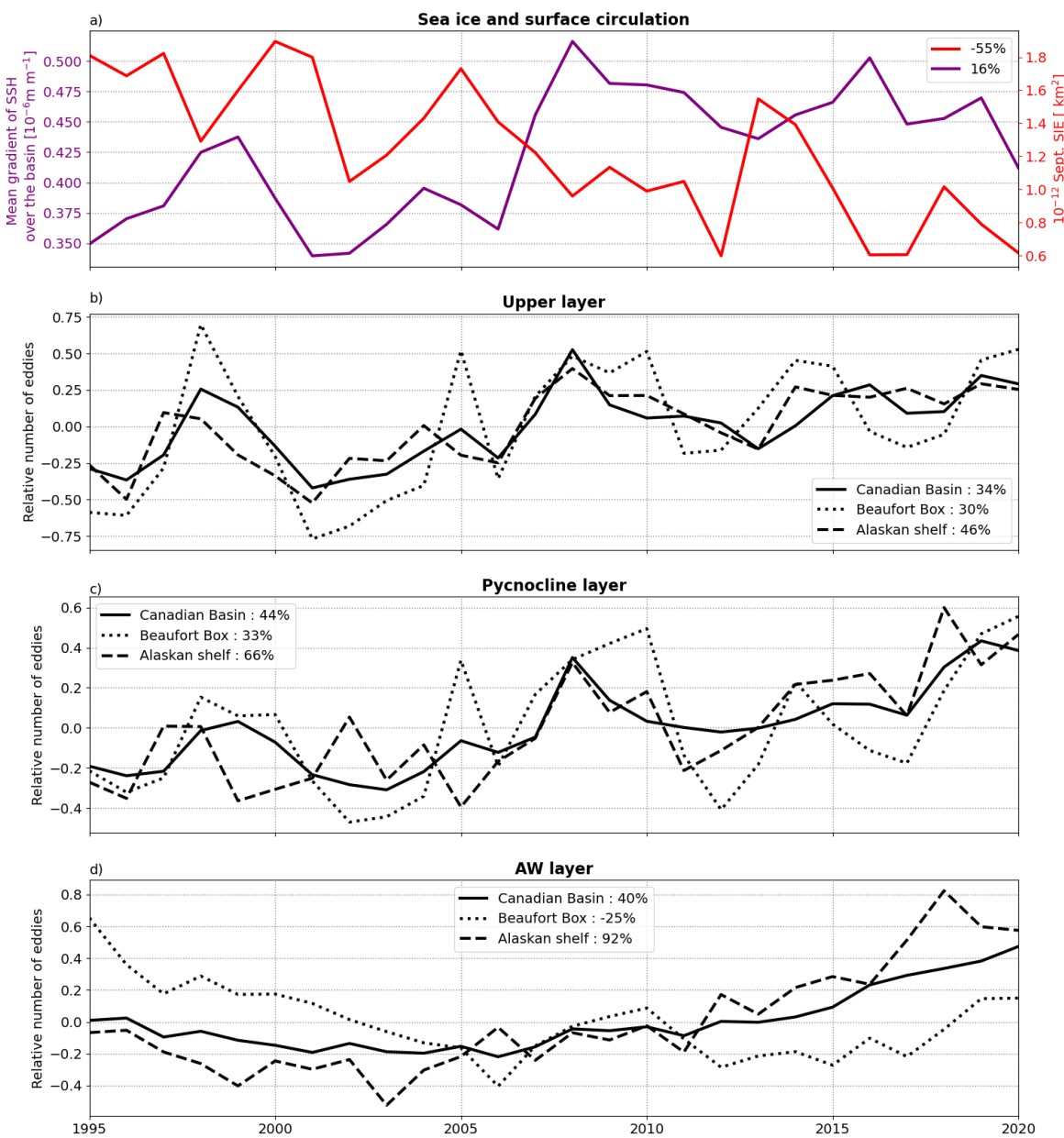

**Figure 10.** Time series of (a) the September sea ice extent over the Canadian Basin (red, total area of the domain is $2.7 \, 10^{12} \, \text{km}^2$) and norm of the gradient of SSH calculated at every location of the domain and averaged over the Canadian Basin. (b), (c) and (d) show the layered-averaged number of eddies relative to the 26-year average for (b) the upper layer, (c) the pycnocline layer, and (d) the AW layer, further divided into the whole Canadian Basin (plain black line), the Alaskan shelf area (dashed black line) and the Beaufort Box (dashed dotted line). Reporting numbers of eddies relative to the $1995 - 2020$ average for each depth level permits a comparison of the temporal evolution of depth levels with different absolute numbers of eddies.

polarity, intensity, lifetime, distance travelled and temperature and salinity anomaly) vary significantly across space and time (Table 3), pointing to the role of the environment, such as the stratification, sea ice or main currents, in setting different processes important for the generation and dissipation of eddies in the Canadian Basin.

Our analysis highlights three layers that arise from the stratification of the Canadian Basin and show consistent characteristics for eddies across the vertical. It is important to note that the definition of the three layers relies on the statistical properties of the whole eddy field over 26 years and hence does not account for the temporal variability of the mean circulation, the location of the gyre, and the mean isopycnal depths and slopes. Thus, the depths used as delimiters of the three layers do not necessarily correspond to the actual, instantaneous isopycnal layers that they are assumed to represent at all times. This is particularly

true where the isopycnal surfaces are strongly tilted or even outcrop over the slope and the shelf break. There, fixed depth layers ensure that there is no averaging between regions in contact with sea ice and those away from the sea ice influence that would otherwise complicate the interpretation of the results. The results presented in this study, and in particular the spatial structure of the eddy properties and their temporal variability, remain consistent when slightly varying the layers' upper and lower boundaries, indicating the robustness of the key features reported for the three layers, especially in the centre of the BG

where the isopycnals are relatively flat.

In the upper layer (top 85 m), which lies on average above the pycnocline, eddy properties display a significant seasonal cycle generally in phase with that of sea ice. Eddy population is the densest when sea ice concentration decreases below 80%, a threshold in line with the results of Manucharyan and Thompson (2022), and over the continental slope. At about ∼85 m, the upper boundary of the pycnocline insulates the eddy field from dissipation by sea ice. Therefore, weaker seasonality is

detected in the eddy characteristics below this depth. Between 85 m and 225 m depth, a reduced density in the eddy population is found within a 250 km wide area along the Alaskan shelf break that is attributed to the smaller gradients of background potential vorticity. Deeper down, the AW layer shows a muted seasonal cycle in eddy properties and little to no similarities to the layers above, due to the efficient insulation of this region by the pycnocline. In particular, while the pycnocline layer shows anticyclones forming preferably at the centre of the BG, in the AW layer, a symmetry is found along the slope, with

anticyclones forming in-shore and cyclones off-shore.

The 1995 − 2020 period is marked by an overall increase in eddy density at all depths (35 − 45%), in line with Meneghello et al. (2021) findings of enhanced baroclinic instabilities with reduced ice cover, leading to enhanced eddy generation. This increase of eddy density also matches the predictions of an increasingly energetic Arctic Ocean (Rieck et al., 2025b; Li et al., 2024). A smaller increase is visible in the upper open ocean and upper MIZ (10%), and is limited to the ML of the pack ice

area. We argue that the higher density in the eddy population is the result of an increasing penetration of energy in the upper layers of the basin, where the ice concentration is small enough. Large interannual variability in the number of eddies is visible in the pycnocline layer, with in particular a peak in the eddy population between 2007 − 2009, when the BG is known to have accelerated. Post 2010, the eddy population is similar in the pycnocline layer to that prior to 2007, which tends to confirm the hypothesis of Regan et al. (2020) that the gyre is able to expand above Northwind ridge and leading to a diminution of the

baroclinic instabilities. However, we report longer equilibration times for the number of eddies, especially in the BB. The upper layer also shows the imprint of the ice cover on yearly time scale with significant correlations between the ice area and the

| | N. of Eddies | $R$ [km] | $dt$ [days] | $D$ [km] | $r_{C/T}$ [%] | $|\Omega|$ [s$^{-1}$] | $\Delta T$ [6°] | $\Delta S$ [k kg$^{-1}$] |
|---|---|---|---|---|---|---|---|---|
| Upper layer 0 − 85 m | 6,000/2.4 | 12.4/0.1 | 5.4/0.6 | 11.3/0.7 | 48.1/0.09 | $7.4\,10^{-6}$/0.4 | 0.08/1.7 | −0.01/ − 7.1 |
| Pycnocline layer 85 − 225 m | 9,000/0.6 | 11.6/0.04 | 10.8/0.1 | 12.2/0.3 | 49.5/0.05 | $3.3\,10^{-6}$/0.3 | 0.01/5.1 | 0.008/6.3 |
| AW layer 225 − 1,200 m | 5,500/0.5 | 11.8/0.02 | 14.2/0.3 | 7.1/0.1 | 50.8/0.06 | $1.5\,10^{-6}$/0.2 | 0.003/1.1 | −0.003/ − 2.5 |
| **Total** | 6,00/2.6 | 12.1/0.06 | 9.9/0.2 | 11.1/0.4 | 49/0.05 | $4.6\,10^{-6}$/0.04 | 0.04/1.6 | −0.006/7.6 |

**Table 3.** Mean/seasonal cycle amplitudes of the eddy properties reported along this manuscript for each layer and for the Canadian Basin as a whole.

number of eddies along the Alaskan shelf break and at the surface of the BB. Overall, the interannual and decadal variability of the pycnocline and surface layers result from both local changes in sea ice dissipation and large-scale changes in energy input.

One of the striking and most intriguing characteristics of the eddy field reported so far in the literature is the eddy polarity, which ranges from $r_{C/A} = 5\%$ to $r_{C/A} = 70\%$ (see Sect. 1, see also Cassianides et al., 2023; Kozlov et al., 2019). At the scale of the Canadian Basin, as many anticyclones as cyclones are found in the upper layer and pycnocline layer of our model. This partition is maintained for small eddies ($< 15$ km), or long-lived eddies (not shown). Yet, the polarity of eddies shows a marked spatial pattern over the Canadian Basin with a larger proportion of anticyclones than cyclones ($\approx 70\%$ anticyclones) in the

centre of the BG (see Figures 7d, and 8d). This predominance of anticyclones aligns with that estimated using a dataset based on ITPs (Cassianides et al., 2023; Zhao et al., 2014), though observations suggest a much higher proportion of anticyclones in that region ($95\%$ versus $65\%$ in our model when applying our eddy detection following the temporal and spatial sampling of the ITPs, see Fig. S13 in the supplementary material). Although the comparison between the ITP dataset and our model is somewhat limited by the fact that the observed eddy field may be dominated by sub-mesoscale features, which our model does

not resolve, our results suggest that part of the anticyclonic dominance documented by the ITP dataset is simply linked to the ITP sampling location, as was already suggested by Beech et al. (2025).
       Our algorithm identifies as an "eddy" a broad range of features, from the ephemeral ones that last a couple of days to the more persistent ones that are likely more coherent. When separating between short- and long-lasting eddies based on their duration being, respectively, shorter or longer than their turnaround time, we find that the bulk of the eddy dataset consists of

short-lasting eddies, which we refer to as a "turbulent soup". Within this ephemeral eddy population, it is likely that some short features are artifacts of the tracking algorithm, which may lose track of the weakest eddies, or of eddies splitting/merging. Indeed, the OW detection algorithm is known to be biased towards weak eddies (Rieck et al., 2025a). Yet, most of the features are likely actual, short-lasting eddies that are evanescent by nature. Within the upper layer, the very short lifetime of these eddies could be attributed to the presence of sea ice (spindown time scale of eddies due to ice friction is estimated around 4

days, Meneghello et al., 2021). At depth, weak vortices have been suggested to form from the stirring of interior potential vorticity gradients (Manucharyan and Stewart, 2022). Alternatively, the weak eddies and the relatively low EKE in our model arise from its resolution (Sect. 2). Overall, these relatively weak and short-lasting eddies that form a turbulent soup are not captured by the observational dataset, which may explain some of the important differences found between our census and the observation-based literature. Nonetheless, these eddies may play an important role in the transfer of energy, and we leave it to future analysis to quantify their integrated role in the penetration of heat, salt, and nutrients into the deep basin.

Long-lasting eddies may more closely resemble the eddy population captured by observations. We find that these eddies represent $15\%$ of the population in the upper layer, $10\%$ in the pycnocline layer and $6\%$ in the AW layer (see also the mean statistical properties of long-lasting eddies on Fig. S14 of the supplementary material). A fraction of these eddies display temperature anomalies, in particular along the shelf break, while only weak salinity anomalies are observed within this eddy subset. These temperature anomalies are mostly positive at the surface, where eddies are formed either within the ML or the warm Near Surface Temperature Maximum; and in the upper part of the pycnocline layer, where they form within the warm sPW (not shown). In contrast, these anomalies are mostly negative in the lower part of the pycnocline layer, where they form in the cold wPW (not shown). Because these long-lasting eddies travel for a few tens of kilometres, we hypothesize that the most coherent eddies actively play a role in transporting heat, in line with previous observations of warm eddies directly penetrating into the BG from Pt. Barrow (MacKinnon et al., 2021). An example of such an eddy is given in Fig. 11a, with an anticyclone carrying warm water off-shore when leaving Pt. Barrow. This eddy is seen to subduct at depth from mid-September, with a colder and fresher layer developing above (Fig. 11b). However, a robust quantification of the heat transport associated with these eddies is not trivial as it requires computing the temperature anomaly, which is highly dependent on the definition of the environment. This analysis is thus left for future work.

This study is a first attempt to perform a systematic and quantitative characterization of mesoscale eddy properties in the Canadian Basin. Yet, an evaluation of the modelled eddy characteristics against observations is hampered by the incomplete resolution of the mesoscale spectrum in our $1/12°$ model at these high latitudes, given that most features identified with moorings, ITPs or satellite fall between the meso- and the submeso-scales (Cassianides et al., 2023; Kozlov et al., 2019). Therefore, these results should be reproduced by a model with higher resolution to be confirmed and compared with observations. The increase in horizontal resolution should be accompanied by an increase in vertical resolution that would not only improve the representation of the (sub)meso-scale features, but also improve the representation of some of the processes sourcing these eddies, especially in the ML, and therefore more accurately represent the variety of features found in the Canadian Basin (e.g., Manucharyan and Timmermans, 2013). A comparison of eddy characteristics between model and observations should account for the typical sampling biases in observations, such as the spatial distribution of the ITPs and the seasonality of the satellite acquisition of ocean surface properties due to the sea ice cover (Kozlov et al., 2019). This comparison could be undertaken with an Observing System Simulation Experiment and would help further our current understanding of eddies from observations and quantify the biases and uncertainties associated with the available observations. Finally, the other important limitation of our study lies in the approach used for the detection of eddies, which lacks a vertical dimension. We believe implementing a

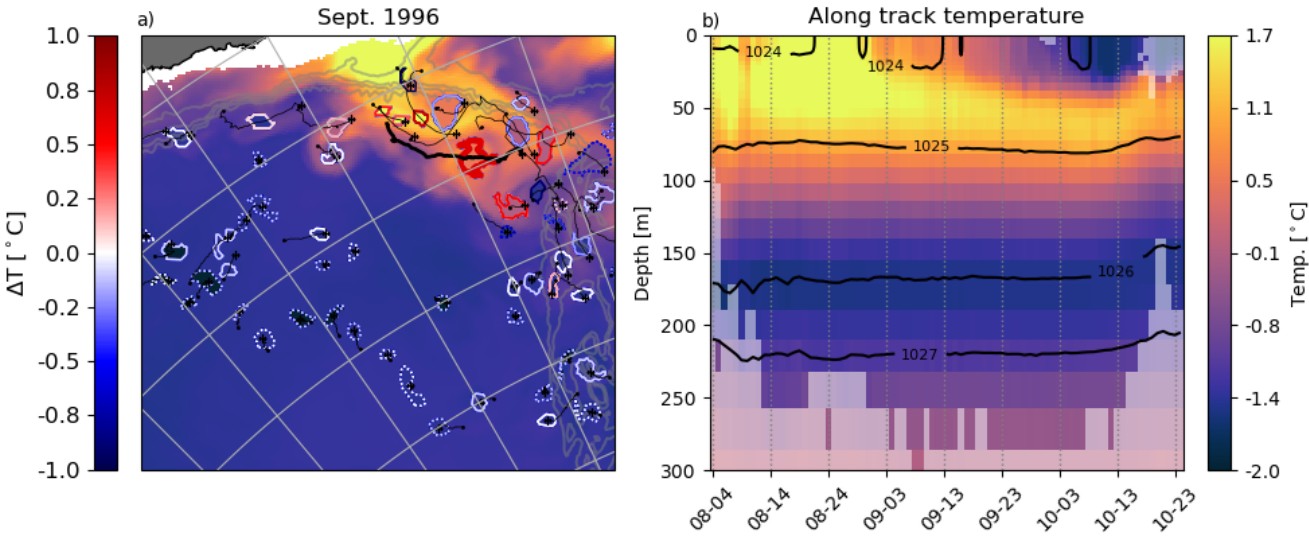

**Figure 11.** (a) Conservative Temperature at 30 m in September 1996 overlaid with contours corresponding to all long-lasting eddies detected (not necessarily for the first time) on September 16th 1996. Colours of contours indicate the temperature anomaly of eddies, and thin black lines their trajectory. Eddies are coloured (filled) by their mean temperature. (b) Conservative temperature along the eddy track is identified on (a) with a thicker black line. At a given depth, intense color shading indicates that the algorithm detects an eddy at that depth, location and time while pale color shading indicates that no eddy is detected. Thick black lines show isopycnals along the eddy track.

3D reconstruction of eddies could form a substantial improvement of the present work, and would allow us to tackle additional questions regarding the formation mechanisms of the mesoscale eddies. In particular, one would be able to investigate the transport of the mesoscale eddies along isopycnals and their subduction at depth, as suggested in Fig. 11 and in the literature (Manucharyan and Timmermans, 2013), or the links between isopycnal displacements and eddy generation (Cassianides et al., 2023).

To conclude, we present a first characterization of the spatio-temporal properties of mesoscale eddies in the Canadian Basin in a high-resolution regional model of the Arctic-North Atlantic. By doing so, we reveal strong differences in eddy properties

across space and time, as well as important variability in the number of eddies generated over $1995 - 2020$ in relation to the loss of sea ice and acceleration of the BG. Our eddy census can thus provide a benchmark against which censuses from other models could be compared, and could form a starting point to explore questions that remain on the BG dynamical and thermodynamical equilibrium, as well on the transport and mixing of nutrients, salt, or other tracers.

*Code and data availability.* The *eddytools* python package used to perform the eddy detection and tracking, along with its documentation is available at https://github.com/jk-rieck/eddytools@N-tracking-properties. The scripts used in this study to detect, track and analyse the mesoscale eddies are available at https://github.com/noemieplanat/Eddies_CB/releases/tag/submission. The documentation of the CREG12 experiment can be found in Talandier and Lique (2024). The detection and tracking for 30, 69, 147 and 508 m depth are available here 10.5281/zenodo.17713586. Fresh water content estimates were obtained from https://www2.whoi.edu/site/beaufortgyre/data/freshwater-content-gridded-data/, accessed on October 1st, 2025. The World Ocean Atlas (WOA) 2023 climatology was downloaded from https://www.ncei.noaa.gov/access/world-ocean-atlas-2023/, accessed in January 2025 (Locarnini et al., 2024; Reagan et al., 2024). Arctic dynamic topography/geostrophic currents data were provided by the Centre for Polar Observation and Modelling, University College London (Armitage et al., 2016, 2017). Sea ice concentration was obtained from the National Snow and Ice Data Center (NSIDC, Di-Girolamo et al., 2022). PIOMAS Reanalysis (Zhang and Rothrock, 2003) was downloaded from https://psc.apl.uw.edu/research/projects/arctic-sea-ice-volume-anomaly/

*Author contributions.* NP designed and conducted the study with input from COD, CL and LBT. CT and CL ran CREG12 model. JKR built the eddytools python package, and NP applied it to the output of CREG12 with appropriate modifications. NP processed and analyzed the eddy database. NP, JKR, COD, CL and LBT contributed to the interpretation. NP wrote the manuscript, with contributions from COD, CL, LBT, JKR, and CT for editing.

*Competing interests.* The authors declare no conflict of interest.

*Acknowledgements.* NP was supported by the Fonds de recherche du Québec - Nature et Technologie (FRQNT) through a Doctoral Training Scholarship, a Natural Sciences and Engineering Research Council of Canada (NSERC) Accelerator Supplements awarded to COD (grant no. RGPAS/2018-522502), and a NSERC Discovery Grant awarded to LBT (grant no. RGPIN/2018-04838). NP also received a scholarship from ISblue (Interdisciplinary graduate school for the blue planet - ANR-17-EURE-0015) and financial support from Québec-Océan for this work. COD and JKR acknowledge funding from a Canada Research Chair (grant no. 252794) awarded to COD. NP, COD, CL, JKR and LBT acknowledge the financial support from the Fonds de recherche du Québec – Nature et technologies (FRQNT) and the French Ministry of Europe and Foreign Affairs through the Samuel-de-Champlain grant (https://doi.org/10.69777/329860). CL and CT were supported by funding from the CLIMArcTIC project funded by the "PPR Océan et Climat—France 2030" (contract ANR-22-POCE-0005). The pan-Arctic simulations were performed using HPC resources from the French GENCI-CINES center (Grant 2023-A0130107420). The authors

also acknowledge the technical and scientific contributions of Benjamin Valette and Sacha Coez to the preparatory phase of the study. We finally thank two anonymous reviewers and the editor J. Mak for their constructive and insightful input that we believe greatly improved the paper.

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
