# Peer review of "Characteristics of ocean mesoscale eddies in the Canadian Basin from a high resolution pan-Arctic model"

_EGUsphere, 2025_

## Author Comment (AC1)

**Authors responses to editor's comments**

**Characteristics of ocean mesoscale eddies in the Canadian Basin from a high resolution pan-Arctic model**

Noémie Planat, Carolina O. Dufour, Camille Lique, Jan K. Rieck, Claude Talandier, L. Bruno Tremblay

**Manuscript Reference Number: 2025-3527**

Date: November 25th, 2025

In the following, all page numbers refer to the revised manuscript.

**General comments**

**Editor:** I was waiting for the second referee to post (and an informal referee to comment) before I posted, and thankfully they corroborate what I was going to say already. I also think the paper can be published eventually, but quite a bit of elaboration and appropriate literature review will be required, and would recommend the authors to do replies and make revisions in due course.

**Authors:** We appreciate the meaningful effort at summarizing the reviewers' points and adding constructive suggestions for improvement of the paper. We have substantially modified the manuscript, in particular by rewriting the introduction and restructuring the results section. We have also provided a detailed justification of the ability of our model to resolve a large portion of the mesoscale eddy spectrum and added more discussion on the limitations of the method. Below, we provide point by point responses to the general comments provided by the editor.

**Editor:** To be blunt about this, my big issue with a lot of the eddy detection papers is "so what?", and to me this paper falls into that category. The polarity, eccentricities, depth extent, numbers etc by themselves are to me rather meaningless if they are not linked up explicitly to the appropriate physical mechanisms and/or possible consequences. A few of these might be:

- biases in polarity may be related to deviation from geostrophic theory

- vortices have a tendency to circularise, so highly eccentric vortices may indicate forcing mechanisms at play disrupting their shape

- deep/intermediate/shallow extent of the eddies may be related to the generation mechanism of the eddies

- numbers may indicate changes in the generation or dissipation mechanism (e.g. ice-ocean drag)

At the moment some of the eddy metrics are presented and it is as if the reader is supposed to just know why these are important (I am not personally convinced they are to be honest). Anyhow, the fix is probably reasonably "easy" to do in principle: motivate the science and relationship with the metrics to be presented in a more comprehensive fashion in the introduction section (and may be recap accordingly in the conclusion), adding (probably quite a few) references of the literature as appropriate.

Set up the scene on why the metrics to be discussed are going to be relevant, then go into detail as much as the authors would like.

**Authors:** To address your "so what?" concern, we have worked on reframing and restructuring the paper. Mainly:

- The introduction has been rewritten to link the metrics with the physical mechanisms. In particular, we examine the eddy location, number and radius in relation to their generation mechanisms and the eddy duration in relation to the dissipation mechanisms. We use these metrics in the paper to (1) document the spatial and seasonal characteristics of mesoscale eddies in the Canada Basin and (2) investigate the evolution of the number of eddies over the past decades in link with the sea ice decline and circulation intensification.

- The results section has been restructured to align with the two main goals stated above.

- The discussion and conclusion section has been modified to better discuss the use of these metrics and the interpretation of some results in link to the limitation of the model. Biases in eddy polarity are discussed in this section, as well as eddy temperature anomalies and possible impact on heat transport.

We believe that these major changes provide a more compelling story in better adequation with our approach and goals.

**Editor:** Note that the referees point out that there are some details that are not entirely sufficient at present (e.g. statistical significance), so do have a look at those and address those accordingly.

**Authors:** We now provide more details about the implemented methods following the recommendations by the reviewers. For example, we added an explanation on how the reported % increases and decreases are calculated on lines 473-474 (addressing the reviewer's concerns about statistical significance) and modified the wording associated with the SSH-gradient calculation in the caption of Fig. 10 so that the methodology is clearer.

**Editor:** Eddy detection per vertical level, as noted by the referees also. This overblows the amount of eddies there actually are, and the vertical coherence and structure of the eddies are one of the most important aspects associated with geostrophic eddies partly because these probably relate to the generation mechanism and/or background turbulence characteristics (e.g. Eady-modes are deep while Charney-Green modes are more vertically confined; QG theory suggests inverse cascades and barotropisation but maybe there are things present in the system that does not allow that to proceed fully). This really should be fixed, or failing it being fixed completely in this article, some samples of cases showing vertical coherency should probably be shown somewhere to at least demonstrate it is doable. I am making the

assumption that there are cases with vertical coherency because that's what theory would suggest; it would be more surprising if there wasn't (although I would then be inclined to think the choice of method has deficiencies instead).

**Authors:** We acknowledge that the detection of eddies per depth level prevents a precise count of the total number of eddies in the basin and an in-depth investigation of the mechanisms linked to the generation and evolution of these eddies, especially processes at the time scale of one individual eddy's life (e.g. barotropisation). That being said, when reporting the number of eddies in the basin, we have been careful to account for this limitation. Yet, reading the reviews has made us realize that our wording may have been misleading so we now have adapted it following the suggestion of the reviewers by reporting the number of eddies per layer instead of averaged per model depth level.

Our 2D approach of detecting eddies is due to the limitation in computational resources (tracking eddies in 3D is extremely expensive; see e.g. Crews et al. [2018], Le Vu et al. [2018], Nencioli et al. [2010]) and by the fact that reconstructing the vertical extent of detected eddies is far from being trivial, so that there is a higher confidence in 2D detection and tracking than in a similar exercise in full 3D. While the incomplete detection keeps us away from an in-depth investigation of the vertical structure of eddies and of its underlying mechanisms, it allows us to encompass a large domain where robust statistics can be provided at every depth level.

Within this framework, we can still obtain an idea of the vertical structure by looking at a few specific objects. This does not form any statistical characterisation of the vertical extent, but as mentionned by the editor, should give confidence in the performance of the detection and tracking. This can be done, for instance, by looking at the vertical structure of kinetic energy at virtual moorings, in relation to the eddies passing by these moorings that our algorithm has detected (Fig. 5). These moorings show surface intensified eddies, eddies spanning the pycnocline and eddies spanning the whole AW layer, which agree well with previous observations and idealized model-based vertical structures [Carpenter and Timmermans, 2012, Zhao and Timmermans, 2015, Meneghello et al., 2021].

The definition of the three layers used in the manuscript thus overall arises from *(i)* the mean stratification of the basin, *(ii)* the examination of the variation of eddy metrics across the vertical (Fig. 4) and *(iii)* examining a few eddies vertical extent (Fig. 5). The latter has now also been included at the beginning of the result section of the revised manuscript so as to provide some insights into the vertical structures of detected eddies; and to give confidence in the layer-averaged description of the eddy statistics :

*Lines 329-332 : We observe some vertical coherency when looking at a few structures individually, in particular for structures spanning the pycnocline between 70-250 m, or surface*

*intensified eddies, or eddies spanning the whole water column below 200 m (Fig. 5). This vertical structure is similar to vertical structure obtained from observations [Carpenter and Timmermans, 2012, Zhao and Timmermans, 2015] or predicted from baroclinic instability estimate [Meneghello et al., 2021] in the CB.*

**Editor:** 1/12 degree is probably not completely sufficient to resolve eddies in the Arctic region and the eddies that are actually detected here might have been mis-represented / mangled already. If the authors go with "high res models are expensive and we have this model handy so we will do it on this as a first try" then sure, but the authors go with "this is a precursor to compare with observations", then a natural question that arises would be whether a comparison is even meaningful at all. The latter stance to me is a problematic one to take with the present dataset; the former stance is at least slightly more defensible.

**Authors:** We are well aware of the limitations of the 1/12 degree resolution. The choice of the model was motivated by the goal of the paper to investigate the evolution of the number of eddies over the last decades that have seen the most drastic changes in the sea ice state and circulation. The relatively long period of integration of the model not only allows for this investigation but also for a longer period of equilibration of the solution than that performed with fully eddy-resolving models. While the model only resolves the larger scales of the mesoscale spectrum, this part has been shown to contain most of the kinetic energy [65% of the EKE is contained in scales larger than 20km, Liu et al., 2024].

The revised manuscript contains the following additions and modifications to clearly state the limitations of the model in terms of resolution of mesoscale eddies and to motivate the choice of resolution :

- *Lines 147-153 (Method) : This relatively fine horizontal grid size allows for an explicit resolution of most of the mesoscale spectrum within the deep basins where the first Rossby radius of deformation $R_o$ is $\approx 10 - 15$ km, but not over the continental slope and shelf where $R_o < 7$ km [Nurser and Bacon, 2013, see also Fig. S1]. Higher resolution simulations of the Arctic Ocean ($\approx 1$ km) have shown that the EKE spectrum peaks around 50 km [Li et al., 2024] and that more than 80% (resp. 65%) of the EKE is contained in scales larger than 10 km [resp. 20 km; Liu et al., 2024]. Therefore, we argue that $1/12°$ is a resolution fine enough to represent most of the mesoscale features in the Beaufort Gyre and along its margins (but not over the shelves), while it runs at a cost that allows for decadal integration.*

- Supplementary materials : A figure (S1) showing the ratio of the first Rossby radius of deformation $Ro$ to the model grid spacing $dx$ is shown (as required by one of the reviewers). It shows that the BG area is resolved with $Ro/dx = 3$ and the Makarov

Basin with $Ro/dx = 2.5$. It also shows that the shelves are resolved with $Ro/dx \leq 1$.

- Discussion : It is now clearly acknowledged that any direct and quantitative comparison with observations is limited by the model resolution that is not high enough to resolve the very small eddies detected by e.g. ITPs (down to 3 km).
  *Lines 608-611 : Yet, a proper and detailed comparison with observations would require using a model at higher-resolution given that most features identified with moorings, ITPs or satellite fall between the meso- and the submeso-scales. Such a comparison would also benefit from an adequate model subsampling, through an Observing System Simulation Experiments, to take into account the observations sampling biases.*

**Editor:** Detection methods based on Okubo-Weiss. I have my opinions on Okubo-Weiss as well as Eulerian methodologies but the thing that is sticking out the most is a lack of discussion with other eddy detection methodologies (as raised also by the referees). By all means make a choice, but that choice needs to be evaluated against other known methodologies, and that's not really done here at the moment.

**Authors:** The justification of why OW was chosen now appears in the method section.
*Lines 232-241 : As we aim to detect any vortex-like features that may develop in the Canadian Basin, including those which are not materially coherent, we choose a Eulerian over a Lagrangian approach for detection. The OW-method is based on velocities (u, v) and thus preferable over SSH-based methods for detection in sea ice-covered areas where SSH-based detections are known to miss objects that do not have a surface expression. Additionally, the OW-method has the advantage to be computationally efficient and thus seems well-suited for a detection run for 26 years at each model level between the surface and 1200 m. A comparison of our OW-based detection with those from Nencioli et al. [2010, u, v - based] and Chelton et al. [2011, SSH-based] was performed by Rieck et al. [2025, see their Fig. S3]. They show that the OW-based method detects higher numbers of eddies compared to the other methods, mostly due to its capability to detect weak eddies, i.e. eddies with small rotational velocities and SSH anomaly. This detection bias towards weak eddies is commented in the discussion section.*

**Editor:** Comparison with observations: There was some suggestion the work forms a precursor to comparison with observations, but then do the current results tell us more about where we should observe, what kind of temporal / horizontal / vertical resolution would be need, and whether that provides a logistical constraint on what is observable and what is not?

**Authors:** Our study provides a continuous and complete eddy census of the Canada Basin over 26 years. Doing so, it confirms at the scale of the basin some of the results based on punctual observations, such as the abundance of eddies all along the continental slope and the

presence of large relatively long-lived eddies within the halocline and at depth in the central basin. Yet, a proper and detailed comparison with observations would require using a model at higher-resolution given that most features identified with moorings and ITPs fall between the meso- and the submeso-scales. Such a comparison would also benefit from an adequate model subsampling, through for e.g. an OSSE, to take into account the observations sampling biases. As mentioned in response to previous comments, we have rewritten the introduction and part of the conclusion and discussion to clarify the goals and implications of the paper. In particular we have removed the following sentences:

- *Introduction : This census aims to facilitate the comparison with observations [...]*

- *Conclusion and discussion: By providing a thorough characterization of eddy properties in the Canada Basin, this model-based eddy census can inform the analyses and interpretation of eddy censuses based on observations and help quantify some of the biases associated to in-situ deployments.*

**Editor:** I would suggest the authors consider justifying this work in another way, because I think that stance is hindering rather than helping the paper by raising more questions that lead to not very convincing answers...

**Authors:** The introduction now emphasizes the documentation of the spatio-temporal characteristics of eddies in the Canada Basin and their evolution over the past decades.
* * *
**Minor comments**

**Editor:** We generally don't like or accept phrases like "The eddy dataset is available upon request" on line 526. If the raw dataset is large then please say so, but minimally the expectation is that a subset of the raw data or the processed data be made available. The review process can continue but is subject to that aforementioned point about data availability be addressed accordingly.

**Authors:** Unfortunately, the raw dataset is too large (over 50GB) to be stored on public data storage. We now provide on Zenodo a subset of the raw dataset with eddy detection and tracking output for 4 depths (30 m, 70 m, 150 m and 500 m ) discussed in the paper.

**References**

L. Crews, A. Sundfjord, J. Albretsen, and T. Hattermann. Mesoscale Eddy Activity and Transport in the Atlantic Water Inflow Region North of Svalbard. *Journal of Geophysical Research: Oceans*, 123(1):201–215, 2018. ISSN 2169-9291. doi: 10.1002/2017JC013198. URL `https://onlinelibrary.wiley.com/doi/abs/10.1002/2017JC013198`. _eprint: https://agupubs.onlinelibrary.wiley.com/doi/pdf/10.1002/2017JC013198.

Briac Le Vu, Alexandre Stegner, and Thomas Arsouze. Angular Momentum Eddy Detection and Tracking Algorithm (AMEDA) and Its Application to Coastal Eddy Formation. *Journal of Atmospheric and Oceanic Technology*, 35(4):739–762, April 2018. ISSN 0739-0572, 1520-0426. doi: 10.1175/JTECH-D-17-0010.1. URL `https://journals.ametsoc.org/view/journals/atot/35/4/jtech-d-17-0010.1.xml`.

Francesco Nencioli, Changming Dong, Tommy Dickey, Libe Washburn, and James C. McWilliams. A Vector Geometry–Based Eddy Detection Algorithm and Its Application to a High-Resolution Numerical Model Product and High-Frequency Radar Surface Velocities in the Southern California Bight. *Journal of Atmospheric and Oceanic Technology*, 27 (3):564–579, March 2010. ISSN 1520-0426, 0739-0572. doi: 10.1175/2009JTECHO725.1. URL `http://journals.ametsoc.org/doi/10.1175/2009JTECHO725.1`.

J. R. Carpenter and M.-L. Timmermans. Deep mesoscale eddies in the Canada Basin, Arctic Ocean. *Geophysical Research Letters*, 39(20):2012GL053025, October 2012. ISSN 0094-8276, 1944-8007. doi: 10.1029/2012GL053025. URL `https://agupubs.onlinelibrary.wiley.com/doi/10.1029/2012GL053025`.

Mengnan Zhao and Mary Louise Timmermans. Vertical scales and dynamics of eddies in the Arctic Ocean's Canada Basin. *Journal of Geophysical Research: Oceans*, 120(12):8195–8209, December 2015. ISSN 21699291. doi: 10.1002/2015JC011251. Publisher: Blackwell Publishing Ltd.

Gianluca Meneghello, John Marshall, Camille Lique, L Erik Isachsen, Edward Doddridge, Jean-Michel Campin, Heather Regan, and Claude Talandier. Genesis and Decay of Mesoscale Baroclinic Eddies in the Seasonally Ice-Covered Interior Arctic Ocean. *JOURNAL OF PHYSICAL OCEANOGRAPHY*, 51, 2021.

Caili Liu, Qiang Wang, Sergey Danilov, Nikolay Koldunov, Vasco Müller, Xinyue Li, Dmitry Sidorenko, and Shaoqing Zhang. Spatial Scales of Kinetic Energy in the Arctic Ocean. *Journal of Geophysical Research: Oceans*, 129(3):e2023JC020013,

March 2024. ISSN 2169-9275, 2169-9291. doi: 10.1029/2023JC020013. URL `https://agupubs.onlinelibrary.wiley.com/doi/10.1029/2023JC020013`.

A. J. G. Nurser and S. Bacon. Arctic Ocean Rossby radius Eddy length scales and the Rossby radius in the Arctic Ocean Arctic Ocean Rossby radius. *Ocean Sci. Discuss*, 10:1807–1831, 2013. ISSN 1807-1831. doi: 10.5194/osd-10-1807-2013. URL `www.ocean-sci-discuss.net/10/1807/2013/`.

Xinyue Li, Qiang Wang, Sergey Danilov, Nikolay Koldunov, Caili Liu, Vasco Müller, Dmitry Sidorenko, and Thomas Jung. Eddy activity in the Arctic Ocean projected to surge in a warming world. *Nature Climate Change*, 14(2):156–162, February 2024. ISSN 1758-6798. doi: 10.1038/s41558-023-01908-w. URL `https://www.nature.com/articles/s41558-023-01908-w`. Publisher: Nature Publishing Group.

Dudley B. Chelton, Michael G. Schlax, and Roger M. Samelson. Global observations of nonlinear mesoscale eddies. *Progress in Oceanography*, 91(2):167–216, October 2011. ISSN 00796611. doi: 10.1016/j.pocean.2011.01.002. URL `https://linkinghub.elsevier.com/retrieve/pii/S0079661111000036`.

Jan K. Rieck, Carolina O. Dufour, Louis-Philippe Nadeau, and Andrew F. Thompson. Heat Transport toward Sea Ice by Transient Processes and Coherent Mesoscale Eddies in an Idealized Southern Ocean. *Journal of Physical Oceanography*, 55(4):377–396, April 2025. ISSN 0022-3670, 1520-0485. doi: 10.1175/JPO-D-24-0073.1. URL `https://journals.ametsoc.org/view/journals/phoc/55/4/JPO-D-24-0073.1.xml`.

---

## Author Comment (AC2)

**Authors responses to reviewer #2 comments**

**Characteristics of ocean mesoscale eddies in the Canadian Basin from a high resolution pan-Arctic model**

Noémie Planat, Carolina O. Dufour, Camille Lique, Jan K. Rieck, Claude Talandier, L. Bruno Tremblay

**Manuscript Reference Number: 2025-3527**

Date: November 25th, 2025

In the following, all page numbers refer to the revised manuscript.

**General comments**

**Reviewer:** This manuscript documents the characteristics of mesoscale eddies in the Canadian Basin simulated by a 1/12th deg regional model of the Arctic. While the work has the potential to be a useful contribution, the manuscript in its current form is largely descriptive, and the results presented are not particularly compelling. It is not entirely clear what new physical insights are gained from this study and also what the important implications of the results are.

*Authors:* We thank the reviewer for their detailed review of the manuscript and insightful suggestions. To address the major concerns of the reviewer on the presentation of the results and on the insights and implications of the study, we have largely modified the structure of the introduction and that of part of the results to better clarify the focus and outcome of the study. We have also rewritten part of the discussion. We believe that these modifications have significantly improved the manuscript. Below, we address all the concerns raised by the reviewer and indicate the changes made to the manuscript.

**Reviewer:** Importantly, since the study is entirely model based, it is essential to demonstrate that the model simulates the eddy field in the Canadian Basin well by comparing with available observations. At present, the evaluation is very limited: only one figure (Figure 2) shows model-simulated EKE compared with mooring-based estimates at two depths, and the model-data comparison is addressed in a single sentence, which simply states that the agreement is good. This level of validation is insufficient. A more extensive and quantitative comparison with available observations is needed for the simulated eddy field. Without this, readers cannot be confident in the reliability of the subsequent analysis of eddy characteristics from the model output. In fact, Figure 2 shows that there are considerable discrepancies between the simulated and observed EKE.

*Authors:* We agree with the reviewer that validating the eddy field is key to give confidence in the results of the study. In the revised manuscript, the discussion on the comparison between the mooring-based estimates of EKE (von Appen et al. 2022 dataset) and our model EKE has been extended to better highlight the differences and similarities (see updated text below). We also have extended the evaluation of the mesoscale eddy field of the model by comparing our kinetic energy to Fig. A1 from Meneghello et al. 2021 using a virtual mooring. This figure shows that the vertical levels of energy are similar to the one derived from mooring observations, with sub-surface intensified structures between 30-200 m, and deeper (although weaker) structures between 400-2000 m.

*Lines 201-212 : Climatologies of EKE computed relative to monthly means show larger values along the shelf break and along topographic features such as Northwind Ridge, both at*

*the surface (not shown) and within the pycnocline (Fig. 2a). In contrast, the deep basin is more quiescent, with EKE one to two orders of magnitude lower than on the shelves (Fig. 2a,b). The shelf-deep basin contrast in EKE magnitude is a typical feature of the mooring-based estimates [von Appen et al., 2022]. Yet, the intensity of EKE is about one order of magnitude smaller in our model than that derived from observations [von Appen et al., 2022], as documented previously in Regan et al. [2020]. The MKE, which captures the location of the main currents, is of similar order of magnitude as in observations [von Appen et al., 2022] , with discrepancies being partly attributed to the difference in the exact locations of the main currents between models and observations (Fig. S5). Finally, the vertical structure of the total kinetic energy is similar to that derived from the Beaufort Gyre Exploration Project Moorings [compare Fig. S6 with for instance Fig. A1 from Meneghello et al., 2020] with sub-surface intensified structures between 30-200 m, and deeper (although weaker) structures between 400-2000 m, as evidenced in observations by Carpenter and Timmermans [2012].*

**Reviewer:** The model stratification differs from climatology (Figure 1). What is the corresponding radius of deformation in the model? Can your model properly resolve it?

**Authors:** In the revised manuscript, we have added a figure in the appendix (Fig. S1), showing the ratio of the first Rossby radius of deformation $Ro$ to the model grid spacing $dx$ (as also requested by the other reviewer). This figure shows that the BG area is resolved with $Ro/dx = 3$ and the Makarov Basin with $Ro/dx = 2.5$. It also shows that the shelves are resolved with $Ro/dx \leq 1$. We have added a discussion on the ability of the model to resolve the eddy field in the method section :

*Lines 147-153 : This relatively fine horizontal grid size allows for an explicit resolution of most of the mesoscale spectrum within the deep basins where the first Rossby radius of deformation $R_o$ is $\approx 10 - 15$ km, but not over the continental slope and shelf where $R_o < 7$ km [Nurser and Bacon, 2013, see also Fig. S1]. Higher resolution simulations of the Arctic Ocean ($\approx 1$ km) have shown that the EKE spectrum peaks around 50 km [Li et al., 2024] and that more than 80% (resp. 65%) of the EKE is contained in scales larger than 10 km [resp. 20 km; Liu et al., 2024]. Therefore, we argue that $1/12°$ is a resolution fine enough to represent most of the mesoscale features in the Beaufort Gyre and along its margins (but not over the shelves), while it runs at a cost that allows for decadal integration.*

**Reviewer:** There are several other eddy detection methods, e.g., contours of SSH anomaly. Could briefly discuss why you choose a method based on the OW parameter and what the advantage of this method is over other existing eddy detection methods.

**Authors:** We have followed the reviewer's suggestion and have added some discussion on

the eddy detection method to the method section :

*Lines 232-241 : As we aim to detect any vortex-like features that may develop in the Canadian Basin, including those which are not materially coherent, we choose a Eulerian over a Lagrangian approach for detection. The OW-method is based on velocities (u, v) and thus preferable over SSH-based methods for detection in sea ice-covered areas where SSH-based detections are known to miss objects that do not have a surface expression. Additionally, the OW-method has the advantage to be computationally efficient and thus seems well-suited for a detection run for 26 years at each model level between the surface and 1200 m. A comparison of our OW-based detection with those from Nencioli et al. [2010, u, v - based] and Chelton et al. [2011, SSH-based] was performed by Rieck et al. [2025, see their Fig. S3]. They show that the OW-based method detects higher numbers of eddies compared to the other methods, mostly due to its capability to detect weak eddies, i.e. eddies with small rotational velocities and SSH anomaly. This detection bias towards weak eddies is commented in the discussion section.*

**Reviewer:** It would be of interest to many readers to see the spatial distribution of EKE in the three layers, as this would bring together information on eddy number and intensity.

**Authors:** We show in the main manuscript the EKE at 70 and 500 m because these are depths with available observation-based estimates of EKE (Fig. 2). The EKE at 30 m, 70 m, and 150 m is visible on Fig. I, where strong similarities are distinguishable between all layers.

**Reviewer:** All the trends reported need to be tested for statistical significance.

**Authors:** We report the decadal change in the number of eddies by comparing the first 5 years with the last 5 years of the simulation, hence no trend is computed. It now appears more clearly at the beginning of Sect. 3.2 :

*Lines 473-474 : % changes are computed using averages over the first and last 5 years of the simulation)*

**Reviewer:** The strong reduction in eddy number along the southern edge of the BG is attributed to the stabilizing effect of the continental slope. However, the slope (as indicated by the grey contours) appears to lie south of this region of low eddy number? In addition, why would the stabilizing effect of the slope result in stronger eddy intensity there?

**Authors:** As suggested by the reviewer, the stabilisation effect of the slope likely occurs close to the slope itself. Offshore, we compute gradients of background potential vorticity ($\partial_{lon}PV$, see Fig. II, or see also Fig. 9 from Meneghello et al. [2021]). The area with decreased eddy density is located where $\partial_{lon}PV$ is minimum, associated to the isopycnal tilting of the gyre.

[Figure]

Figure I: Eddy Kinetic Energy (EKE) computed from velocity anomalies with respect to the monthly means in CREG12 and averaged over the 26 years of simulation (a) at 30 m, (b) at 69 m and (c) at 147 m. uper-imposed are mooring-based estimates of EKE fromvon Appen et al. [2022], computed with fourth-order Butterworth filter 2-day to 30-day cutoffs. The reader is referred to von Appen et al. (2022) for exact calculation method.

This indicate diminished baroclinic instabilities and thus likely explain the low eddy density. It now reads in the text :

*Lines 428-434 : We suggest that the inner part of this local reduction in eddy generation is linked to a stabilizing effect of the continental slope. The growth of instabilities is known to be hampered over regions where the ratio of the continental slope to the isopycnal slopes is greater than 1 [Manucharyan and Isachsen, 2019], as is the case for the slope of the CB in the model [not shown, see also Regan et al., 2020]. However, this reduction occurs up to 250 km away from the shelfbreak. There, we observed diminished background PV gradients [not shown, see Fig. 9 from Meneghello et al., 2021] associated to diminished baroclinic instabilities, which offer an alternative explanation for the extended area with diminished eddy density.*

**Minor comments**

**Reviewer:** Line 154. "from R=10 km to 2*pi*R=60 km". Is there a typo here?
**Authors:** There is no typo here, we adopted the definition of e.g. Tulloch et al. [2011]. However, no vortices are detected with larger radii anyway.

**Reviewer:** Line 270. Explain why there is a dominance of anticyclones at the very surface within the Canada basin and cyclones within the Chukchi shelf break area in winter?

[Figure]

Figure II: (a) Gradient of background potential vorticity computed with respect to the longitudinal direction $\partial_{lon}PV$ along a latitudinal section (see Fig. 1). Plain blue indicate values smaller that $-1 \cdot 10^{-11}$. (b) Number of eddies detected along that section, with 50 km wide longitudinal width.

**Authors:** We don't have a clear explanation of hypothesis for this feature. We have removed this sentence from the main manuscript.

**Reviewer:** Line 280. Explain what the theoretical mean turnaround time is and how it is calculated.

**Authors:** In the revised manuscript, we have added a definition of the turnaround time :

*Lines 282-283 : [...] their turnaround time scale, defined as the time it takes for a water parcel to do a full revolution, $\tau = 2\pi/|\Omega|$ [i.e. an approximation of the expression suggested by Smith and Vallis, 2001, that is $\tau = 2\pi/\zeta_{rms}$, where $\zeta_{rms}$ is the root-mean-square vorticity.].*

**Reviewer:** Line 298. How is "generation rate" defined?

**Authors:** Generation rate is used equivalently to eddy density, as it is computed at generation. To avoid any confusion, it does not appear anymore in the text.

**Reviewer:** Line 329. . . . in our domain. . .

**Authors:** Done.

**Reviewer:** Line 335-337. This sentence is not clear, as the expansion of the ice-free area also contributes to increased energy input to the ocean.

**Authors:** It now reads :

*Line 505-509: In the MIZ and open ocean, the increase in eddy generation is mainly a step increase in 2008 with reduced (shut down) interannual variability in the MIZ (Open Ocean) in the following years. This enhancement in the density of the eddy population presumably*

*results from the additional energy penetrating into the ocean in the recent state of the BG, in light with the current and future projections of Li et al. [2024]. In the open ocean, this accumulation of energy could be attributed to the acceleration of the BG from atmospheric forcings [Giles, 2012], and in the MIZ to a combination of less compacted ice [Martin et al., 2016] or to the thinner ice cover [Muilwijk et al., 2024]. A quantification of these different drivers is beyond the scope of this paper and left for future analysis.*

**References**

Wilken-Jon von Appen, Till M. Baumann, Markus A Janout, Nikolay V. Koldunov, Yueng-Djern Lenn, Robert S. Pickart, and Qiang Wang. Eddies and the Distribution of Eddy Kinetic Energy in the Arctic Ocean. 35(Special issue on the new arctic ocean):42–51, December 2022.

Heather C Regan, Camille Lique, Claude Talandier, and Gianluca Meneghello. Response of Total and Eddy Kinetic Energy to the Recent Spinup of the Beaufort Gyre. *JOURNAL OF PHYSICAL OCEANOGRAPHY*, 50, 2020.

Gianluca Meneghello, Edward Doddridge, John Marshall, Jeffery Scott, and Jean-Michel Campin. Exploring the Role of the "Ice–Ocean Governor" and Mesoscale Eddies in the Equilibration of the Beaufort Gyre: Lessons from Observations. *JOURNAL OF PHYSICAL OCEANOGRAPHY*, 50, 2020.

J. R. Carpenter and M.-L. Timmermans. Deep mesoscale eddies in the Canada Basin, Arctic Ocean. *Geophysical Research Letters*, 39(20):2012GL053025, October 2012. ISSN 0094-8276, 1944-8007. doi: 10.1029/2012GL053025. URL https://agupubs.onlinelibrary.wiley.com/doi/10.1029/2012GL053025.

A. J. G. Nurser and S. Bacon. Arctic Ocean Rossby radius Eddy length scales and the Rossby radius in the Arctic Ocean Arctic Ocean Rossby radius. *Ocean Sci. Discuss*, 10:1807–1831, 2013. ISSN 1807-1831. doi: 10.5194/osd-10-1807-2013. URL www.ocean-sci-discuss.net/10/1807/2013/.

Xinyue Li, Qiang Wang, Sergey Danilov, Nikolay Koldunov, Caili Liu, Vasco Müller, Dmitry Sidorenko, and Thomas Jung. Eddy activity in the Arctic Ocean projected to surge in a warming world. *Nature Climate Change*, 14(2):156–162, February 2024. ISSN 1758-6798. doi: 10.1038/s41558-023-01908-w. URL

https://www.nature.com/articles/s41558-023-01908-w. Publisher: Nature Publishing Group.

Caili Liu, Qiang Wang, Sergey Danilov, Nikolay Koldunov, Vasco Müller, Xinyue Li, Dmitry Sidorenko, and Shaoqing Zhang. Spatial Scales of Kinetic Energy in the Arctic Ocean. *Journal of Geophysical Research: Oceans*, 129(3):e2023JC020013, March 2024. ISSN 2169-9275, 2169-9291. doi: 10.1029/2023JC020013. URL https://agupubs.onlinelibrary.wiley.com/doi/10.1029/2023JC020013.

Francesco Nencioli, Changming Dong, Tommy Dickey, Libe Washburn, and James C. McWilliams. A Vector Geometry–Based Eddy Detection Algorithm and Its Application to a High-Resolution Numerical Model Product and High-Frequency Radar Surface Velocities in the Southern California Bight. *Journal of Atmospheric and Oceanic Technology*, 27 (3):564–579, March 2010. ISSN 1520-0426, 0739-0572. doi: 10.1175/2009JTECHO725.1. URL http://journals.ametsoc.org/doi/10.1175/2009JTECHO725.1.

Dudley B. Chelton, Michael G. Schlax, and Roger M. Samelson. Global observations of nonlinear mesoscale eddies. *Progress in Oceanography*, 91(2):167–216, October 2011. ISSN 00796611. doi: 10.1016/j.pocean.2011.01.002. URL https://linkinghub.elsevier.com/retrieve/pii/S0079661111000036.

Jan K. Rieck, Carolina O. Dufour, Louis-Philippe Nadeau, and Andrew F. Thompson. Heat Transport toward Sea Ice by Transient Processes and Coherent Mesoscale Eddies in an Idealized Southern Ocean. *Journal of Physical Oceanography*, 55(4):377–396, April 2025. ISSN 0022-3670, 1520-0485. doi: 10.1175/JPO-D-24-0073.1. URL https://journals.ametsoc.org/view/journals/phoc/55/4/JPO-D-24-0073.1.xml.

Gianluca Meneghello, John Marshall, Camille Lique, L Erik Isachsen, Edward Doddridge, Jean-Michel Campin, Heather Regan, and Claude Talandier. Genesis and Decay of Mesoscale Baroclinic Eddies in the Seasonally Ice-Covered Interior Arctic Ocean. *JOURNAL OF PHYSICAL OCEANOGRAPHY*, 51, 2021.

G. E. Manucharyan and P. E. Isachsen. Critical Role of Continental Slopes in Halocline and Eddy Dynamics of the Ekman-Driven Beaufort Gyre. *Journal of Geophysical Research: Oceans*, 124(4):2679–2696, April 2019. ISSN 21699291. doi: 10.1029/2018JC014624. Publisher: Blackwell Publishing Ltd.

Ross Tulloch, John Marshall, Chris Hill, and K. Shafer Smith. Scales, Growth Rates, and Spectral Fluxes of Baroclinic Instability in the Ocean. *Journal of Physical Oceanography*,

41(6):1057–1076, June 2011. ISSN 0022-3670, 1520-0485. doi: 10.1175/2011JPO4404.1. URL http://journals.ametsoc.org/doi/10.1175/2011JPO4404.1.

K. Shafer Smith and Geoffrey K. Vallis. The Scales and Equilibration of Midocean Eddies: Freely Evolving Flow. *Journal of Physical Oceanography*, 31(2):554–571, February 2001. ISSN 0022-3670, 1520-0485. doi: 10.1175/1520-0485(2001)031¡0554:TSAEOM¿2.0.CO;2. URL http://journals.ametsoc.org/doi/10.1175/1520-0485(2001)031<0554:TSAEOM>2.0.CO;2.

Katharine A Giles. Western Arctic Ocean freshwater storage increased by wind-driven spin-up of the Beaufort Gyre. *NATURE GEOSCIENCE*, 5, 2012.

Torge Martin, Michel Tsamados, David Schroeder, and Daniel L. Feltham. The impact of variable sea ice roughness on changes in Arctic Ocean surface stress: A model study. *Journal of Geophysical Research: Oceans*, 121 (3):1931–1952, 2016. ISSN 2169-9291. doi: 10.1002/2015JC011186. URL https://onlinelibrary.wiley.com/doi/abs/10.1002/2015JC011186. _eprint: https://agupubs.onlinelibrary.wiley.com/doi/pdf/10.1002/2015JC011186.

Morven Muilwijk, Tore Hattermann, Torge Martin, and Mats A. Granskog. Future sea ice weakening amplifies wind-driven trends in surface stress and Arctic Ocean spin-up. *Nature Communications*, 15(1):6889, August 2024. ISSN 2041-1723. doi: 10.1038/s41467-024-50874-0. URL https://www.nature.com/articles/s41467-024-50874-0.

---

## Author Comment (AC3)

**Authors responses to reviewer #1 comments**

**Characteristics of ocean mesoscale eddies in the Canadian Basin from a high resolution pan-Arctic model**

Noémie Planat, Carolina O. Dufour, Camille Lique, Jan K. Rieck, Claude Talandier, L. Bruno Tremblay

**Manuscript Reference Number: 2025-3527**

Date: November 25th, 2025

In the following, all page numbers refer to the revised manuscript.

**General comments**

**Reviewer:** This paper uses a 1/12 degree ocean general circulation model to studies the statistics of ocean mesoscale eddies in the Beaufort gyre of the Arctic Ocean. The authors focus on the period 1995-2020. The authors show strong regional and temporal variability. There is strong linkage with the sea ice cover, while the seasonality of the eddies is weaker during the pycnocline. The authors find, that except along the Chukchi shelf break, that most eddies have little to no temperature signal. The eddies in the upper layer increase with time over the simulation. The authors also suggest that the results from ITPs that suggest most Beaufort Gyre eddies are anti-cyclonic may be due to sampling bias.

This is an interesting paper, well worth publishing. And appropriate for Ocean Science. That said, there are ways the manuscript could be improved and I would suggest major revisions. Detailed justification is provided below.

**Authors:** We thank the reviewer for their detailed review of our manuscript and constructive suggestions which we believe have helped improve the manuscript. Below, we address all the concerns raised by the reviewer and indicate the changes made to the manuscript.
* * *
**Major comments**

**Reviewer:** Is a 1/12 degree model really high resolution? Many models of higher resolution now exist – for example there are papers that look at eddies at 1/60 degree in simulations of the Arctic Ocean, Nordic Seas and the Labrador Sea, for example. The advantage of a 1/12 degree simulation is the length of integration for the analysis. So, as long as the given resolution is able to resolve a significant part of the eddy spectrum at the given resolution, I can understand the authors using the configuration they did. But I'd like to see some more discussion of mesoscale eddies and understand how they are represented in CREG12.

Given the model's stratification, could the authors provide information on where the Rossby Radius is resolved and with how many grid cells, especially at the margins of the Beaufort Gyre.

**Authors:** We are well aware of the limitations of the 1/12 degree resolution. The choice of the model was motivated by the goal of the paper to investigate the evolution of the number of eddies over the last decades that have seen the most drastic changes in the sea ice state and circulation. The relatively long period of integration of the model not only allows for this investigation but also for a longer period of equilibration of the solution than that performed with fully eddy-resolving models. While the model only resolves the larger part of the mesoscale spectrum, this part has been shown to contain most of the kinetic energy [65%

of the EKE is contained in scales larger than 20km, Liu et al., 2024].

The revised manuscript contains the following additions and modifications to clearly state the limitations of the model in terms of resolution of mesoscale eddies and to motivate the choice of resolution :

- *Lines 147-153 (Method) : This relatively fine horizontal grid size allows for an explicit resolution of most of the mesoscale spectrum within the deep basins where the first Rossby radius of deformation $R_o$ is $\approx 10 - 15$ km, but not over the continental slope and shelf where $R_o < 7$ km [Nurser and Bacon, 2013, see also Fig. S1]. Higher resolution simulations of the Arctic Ocean ($\approx 1$ km) have shown that the EKE spectrum peaks around $50$ km [Li et al., 2024] and that more than $80\%$ (resp. $65\%$) of the EKE is contained in scales larger than $10$ km [resp. $20$ km; Liu et al., 2024]. Therefore, we argue that $1/12°$ is a resolution fine enough to represent most of the mesoscale features in the Beaufort Gyre and along its margins (but not over the shelves), while it runs at a cost that allows for decadal integration.*

- Supplementary materials : A figure (S1) showing the ratio of the first Rossby radius of deformation $Ro$ to the model grid spacing $dx$ is shown (as required by one of the reviewers). It shows that the BG area is resolved with $Ro/dx = 3$ and the Makarov Basin with $Ro/dx = 2.5$. It also shows that the shelves are resolved with $Ro/dx \leq 1$.

- Discussion : It is now clearly acknowledged that any direct and quantitative comparison with observations is limited by the model resolution that is not high enough to resolve the very small eddies detected by e.g. ITPs (down to 3 km).
  *Lines 608-611 : Yet, a proper and detailed comparison with observations would require using a model at higher-resolution given that most features identified with moorings, ITPs or satellite fall between the meso- and the submeso-scales. Such a comparison would also benefit from an adequate model subsampling, through an Observing System Simulation Experiments, to take into account the observations sampling biases.*

**Reviewer:** And if the motivation to use 1/12 degree is the longer timeseries, this needs to be discussed.

**Authors:** Our motivation is to detect eddies over the few decades that include the major changes in the sea ice and in the ocean circulation that have occurred in the Canada Basin. As mentioned above, this is feasible at the expense of the resolution. We have rewritten the introduction to emphasize this goal, and reorganised the result section to highlight the temporal changes with a dedicated subsection.

**Reviewer:** As well, some discussion of what is found at higher resolution is needed to provide the reader with confidence with respect to their results at the given resolution.

**Authors:** To our knowledge, analyses of energetics of the Beaufort Gyre in realistic higher resolution models have been only done at $\approx 1$ km in the FESOM model. We now refer to both Li et al. [2024] and Liu et al. [2024] in the method section, who show that most of the energy spectrum is at scales larger than 20 km in their model.

*Lines 149-151 : Higher resolution simulations of the Arctic Ocean ($\approx 1$ km) have shown that the EKE spectrum peaks around 50 km [Li et al., 2024] and that more than 80% (resp. 65%) of the EKE is contained in scales larger than 10 km [resp. 20 km; Liu et al., 2024].*

**Reviewer:** I'm trying to understand what the authors' number of eddies mean. The paper states about 6,250 eddies are detected per year and per depth level. That seems like a huge number. First off, I understand the authors point about the difficulty in determining the vertical coherence of the eddies. And the authors do show some changes with depth. But just saying per depth level makes it seem like there are significantly more eddies that there actually are – given most eddies are likely found through multiple levels. In might be better to average numbers over several levels and then state there are on average X eddies detected per year above pycnocline, Y in the pycnocline and Z below it (or using some other depth metric).

**Authors:** Indeed, 6250 appears like a large number. To obtain this number, we average over all 26 years of simulation and across all depth levels. While this is a gross estimate, this resulting number does account for the fact that eddies might span several depth levels. We have followed the recommendation of the reviewer and now report the number of eddies per layer in the abstract :

*Lines 4-6 : Over that period, about 6000 eddies per year are detected in the surface layer, while about 9000 eddies per year are detected in the pycnocline layer, and about 5500 eddies per year in the Atlantic Waters (AW) layer.*

As well as in the whole manuscript.

**Reviewer:** Additionally, how many eddy exist per day, on average (i.e. a timeseries by day through the mean seasonal cycle)? I wonder if the large yearly number if because of the short duration of the eddies being a function of them being lost by the tracking software, then re-found and thus recounted as a new eddy. A median duration of 4 days is short! Or if the eddies are being damped quickly given the setting at the given model resolution? If I look at figure 3c, I don't see that many eddies on the given day (with generally very short trajectories), so I am wanting more discussion of this, to help understand what that large

eddy number the paper provides really means.

**Authors:** The number of eddies detected every day across the Canadian Basin over the first 5 years of the simulation is shown at 70 m on Fig. I. On average, our algorithm detects 85000 eddies per year (including features that last only one day and that we filter in the analysis of the paper). We agree with the reviewer that the number of eddies is likely overestimated, although satellite-base detections have reported $\mathcal{O}(1000)$ eddies with surface signatures in the seasonal ice-free regions and MIZ [Kozlov et al., 2019, Kubryakov et al., 2021, Manucharyan et al., 2022]. At subsurface, however, eddy counts have relied on moorings and ITPs, and are thus probably underestimated. This comparison now appears at the beginning of the results section :

*Lines 311-313 : This large number opposes the very few vortices detected from in-situ observations below the ice [$\mathcal{O}(10)$ eddies per year, Cassianides et al., 2023, Zhao et al., 2014]. It is, however, closer to numbers reported from satellite observations in the Marginal Ice Zone or the Open Ocean [up to $\mathcal{O}(1000)$ eddies per year, Kubryakov et al., 2021, Kozlov et al., 2019, Manucharyan et al., 2022].*

This being said, we believe that 6000 eddies per year is overestimated, especially considering the dominance of short features. We attribute this overestimation to *(i)* the tracking procedure losing track of some eddies and *(ii)* dissipation of eddy energy in the model, with both impacting the eddy lifetime.

The algorithm may lose track of the weakest features, or of eddies splitting/merging. However, relaxing some parameters in the algorithm to better identify and thus track weak eddies may lead to the detection of spurious features. There is thus a trade-off between keeping track of weak eddies and including spurious features. This issue is faced by all eddy tracking algorithms, and sensitivity tests were performed (see Section 2.2). In the final configuration, Figure 5 shows that the algorithm can successfully keep track of individual eddies along their lifetime as expected, failing sometimes along their upper and lower boundaries, where the eddies are likely weaker. This figure have been included in the revised manuscript and discussed in section 3.1.1.

A quantification of the low bias due to the tracking algorithm is difficult and not attempted here. We still argue that part of the short eddies detected are likely real, short lasting, rotating vortices that are evanescent by nature in the model.

Among these short features, part can be attributed to the viscosity and energy level of the model that may differ from the reality and dissipate eddies too quickly. However, we also note that the short lifetime of eddies (few days) in the upper layer is consistent with the theoretical

[Figure]

Figure I: Seasonal cycle of the number of eddies detected per day for the five first years of the dataset at 70 m depth.

spindown time scale of eddies in layers affected by sea ice which was suggested by Meneghello et al. [2021] [based on results from Pedlosky, 1982]). At depth, eddies are expected to live longer (months to years), an increase in lifetime that is captured by our algorithm to a smaller extent (Figure 6e). The signature of subsurface and deep eddies is weaker than that of near-surface eddies which may be the reason why the algorithm may occasionally lose track of eddies especially at depth. Whether these weak vortices at depth are accurately reproducing real features [e.g. as suggested by Manucharyan and Stewart, 2022, from the stirring of PV gradients in the interior] or should be attributed to the weak energy level of the model (see Section 2.1.2) is difficult to assess. We have extended the discussion around the turbulent soup to include the above mentioned arguments (see below).

**Reviewer:** The author's do bring up the idea of turbulent soup, which I like – but I still feels this topic does need more to help the reader understand what the results mean.

**Authors:** We thank the reviewer for bringing up this point. Following the discussion above, we have extended the discussion on the turbulent soup and suggested a few hypotheses for the origin of this "soup":

*Lines 613-624 : When separating between short and long-lasting eddies based on their duration being respectively shorter or longer than their turnaround time, we find that the bulk of the eddy dataset consists of short-lasting eddies which we refer to as a "turbulent soup". Within this ephemeral eddy population, it is likely that some short features are artefact of the tracking algorithm, that may lose track of the weakest eddies, or of eddies splitting/merging. The OW detection algorithm is known to be biased towards weak eddies. Yet, most of the features are likely actual, short-lasting eddies that are evanescent by nature. Within the upper layer, the*

*very short lifetime of these eddies could be attributed to the presence of sea ice [spindown time scale of eddies due to ice friction is estimated around 4 days Meneghello et al., 2021]. At depth, weak vortices have been suggested to form from the stirring of interior PV gradients [Manucharyan and Stewart, 2022]. Alternatively, weak eddies may arise from the relatively low EKE in our model due to its resolution (Sect. 2). Overall, these relatively weak and short-lasting eddies that form a turbulent soup are not captured by observational dataset, which may explain some of the important differences found between our census and the observation-based literature. Nonetheless, these eddies may play an important role in the transfer of energy, and we leave for future analysis to quantify their integrated role in the penetration of heat, salt and nutrients into the deep basin.*

**Reviewer:** The authors talk about no temperature signal – what about salinity? The eddies like play an important role in freshwater exchange into and out of the gyre. This would be good to further explore.

**Authors:** The eddy population that we have detected shows little anomalies for both temperature and salinity as shown in Fig. 4e and f. This appears consistent with previous observational studies [Cassianides et al., 2023]. When looking at long-lasting eddies only though, we do see some individuals with larger temperature anomalies. The main reason why the salinity anomalies are not extensively discussed in the discussion of the long eddies (and associated heat transport) is because the salinity anomalies are much smaller than the temperature anomalies and do not display any spatial pattern (see supplementary S14, see also the scatter plot below with long eddy salinity anomalies at 4 different depths, Fig. II). As suggested in the discussion, it is possible that the cumulative effect of these eddies with little anomalies is important for the equilibrium of the gyre. However, to properly quantify their role in the freshwater exchange across the gyre, and asses their relative importance compared to transient but not circular features (e.g. filaments, fronts), one should quantify their FW transport, which we cannot properly do with this approach.

The abstract now also mention the absence of salinity anomalies on average :

*Lines 14-19 : The vast majority of eddies have no temperature nor salinity signature with respect to their environment, although a significant portion of long-lived eddies, located along the Chukchi shelf break, have a non-negligible temperature anomaly and penetrate into the Beaufort Gyre, thus suggesting a mechanism for the penetration of heat into the gyre.*

**Reviewer:** Strong increase in 2008 – BG spin-up and/or low ice year?

**Authors:** We think that it is likely a mix of the two, at least in the surface layer, and we have rephrased accordingly :

[Figure]

Figure II: Salinity anomalies of long lasting eddies at 30, 69, 147 and 508 m.

*Lines 525-527 : The increase in the number of eddies that peaks in 2008 in our model bears similarity to the EKE increase that was reported by Regan et al. [2020] to occur over one year following the gyre acceleration in 2007 and low sea ice record of that year.*

**Reviewer:** I have some technical questions about the model configuration and experiment. What does constraining the model to \*about\* 1.4 Sv at Bering Strait mean? A constant value with time? Constant annual mean? An annual mean of 1.4 Sv with interannual variability? Additionally, this value seems a bit large compared to the observations.

**Authors:** The 2 boundaries are forced using the output of the GLORYS12V1 reanalysis. From 1979 to 1992, a daily climatology of volume transport computed over years 1993 to 2020 of the reanalysis is applied (due to a lack of model output prior to 1993), then between 1993 and 2020, the daily volume transport of the reanalysis is used to force the boundary of our model.

We thank the reviewer for pointing out the error in the manuscript : the mean transport output from GLORYS12V1 is about 1.4 Sv and has been corrected to match observations estimates from Woodgate [2018] who give a transport of about 1.1 Sv. The wording in the text has been corrected as follows :

*Lines 167-168 : At Bering Strait, meridional velocities are adjusted to constrain the inflow to about 1.1 Sv, matching observation estimates [Woodgate, 2018].*

An updated DOI for the documentation of the run is also available : Talandier and Lique [2024]

**Reviewer:** As well, does the model consider the increase in recent years suggested with the observations? As well, what are the heat and freshwater transports, and how do they compare with the observations?

**Authors:** We show on Fig. III a comparison of volume (upper row), heat (middle row)

[Figure]

Figure III: Estimated transport (first line), fresh water transport (second line) and heat transport (last line) at Bering Strait from observations [green, Woodgate, 2018] and modified from Glorys 12 to correct the voluminc influx (black).

and salt fluxes (lower row) through Bering Strait between Woodgate et al. 2018 (green) and CREG12 (black). All three variables show changes of similar amplitude at seasonal and interannual time scales.

**Reviewer:** The river runoff I think needs to better explained. I thought the Stadnyk et al. paper mentioned uses the AHYPE model? And wasn't the output from that model used in Weiss-Gibbons et al., 2025? And in either case, I don't believe the AHYPE output provided Greenland discharge? Are you sure that didn't come from a different product?

**Authors:** The reviewer is right in mentioning this new reference : Weiss-Gibbons et al. [2024]. The runoff as well as the Greenland discharge data sets that the reviewer mentions are exactly the ones that have been used to force CREG12. It is now corrected in the manuscript and in the updated documentation of the run : Talandier and Lique [2024].

**Reviewer:** Given the importance of the sea-ice in setting the seasonal cycle of the eddies, more on the model representation of sea-ice would be useful. What does comparable to that

derived from satellite observations mean in practice? Especially given that the authors use ERA5 for forcing, which has a known warm bias in the Arctic. So I'm curious about what parameter set for SI3 was able to allow the simulation to get sea ice fields close to the satellite observations.

**Authors:** Some sea ice parameters have been adjusted to compensate the warm bias of ERA5, including the snow conductivity that is set to 0.5 W/m/K, the ice ocean drag coefficient that is set to 7e-3, and the atmosphere-ocean coefficient that is set to 1.2e-3. Ice strength is set to 2e-4 N/m2. These parameters are now mentioned in the Method section of the revised paper where the model is described (Section 2.1.1).

In addition, we have extended the validation of the model, with two additional figures that provide a comparison of sea ice extent and thickness between the model, satellite observations, and PIOMAS.

*Lines 176-185 : Over the period of analysis, the mean September sea ice concentration is comparable to that derived from satellite observations with small differences on the Eurasian shelf and a low bias in the western CB (Fig. 1a,b). On average across the Arctic and along the simulation, the sea ice extent deviates from that derived from satellite observations by −7% in September and −16% in March. When compared to the PIOMAS Arctic Sea Ice Volume Reanalysis [Zhang and Rothrock, 2003], the sea ice thickness is 35 cm thinner in September and 20 cm thinner in Mars (Fig. S2b, S3b). The interannual variability of sea ice extent is well captured by the model across the 26 years of simulation. A strong decline in sea ice starting around 2000 and persisting in time appears in the model in agreement with observations (see Fig. S2c, S3c in supplementary). The corresponding location of ice loss is generally well represented despite some biases in the ice concentration along the Eurasian shelf in summer and high biases along Yermack plateau and Greenland eastern shelf in winter (see Fig. S2a,b, S3a,b in supplementary).*

**Reviewer:** Given the discussion of the three main water masses, I would also be curious to see a time series of salinity (or freshwater content) in the model, compared to the Beaufort Gyre Experiment results. Especially since Rosenblum et al. has pointed out that many models under-estimate the freshwater content in the region.

**Authors:** We have added a figure comparing the fresh water content of the model to that computed from observations Proshutinsky et al. [2019] to the supplementary materials of the revised manuscript (Fig. S4). The associated text appears in the model validation section :

*Lines 194-196 : The overall fresh water content, referenced to 34.8 psu, shows a strong increase between 2003-2009 in the Canadian Basin as documented from the Beaufort Gyre Exploration Project [Proshutinsky et al., 2009] followed by a plateau (see Fig. S4 in supple-*

*mentary).*

**Reviewer:** Also, why use fixed depth layers for the vertical splitting, instead of isopyncals (or isohalines) to define the different water masses?

**Authors:** We used fixed depth layers as it provides a simple framework to use across the domain. In particular, where the isopycnals are tilted or even outcrop over the slope and shelf break, fixed depth layers guarantee that there is no averaging done between regions in contact with sea ice and those away from the sea ice influence; this would otherwise complicate the interpretation of the results. Additionally, a given isopycnal boundary may not allow us to delimit the same water mass across the whole domain. That being said, we agree with the reviewer that isopycnal layers are a natural and somewhat more physically grounded framework than depth layers. The two frameworks are likely equivalent within the BG where isopycnals are relatively flat but may provide significantly different pictures at the edge. We now acknowledge this difference in the method/discussion section:

*Lines 560-569 : It is important to note that the definition of the three layers relies on the statistical properties of the whole eddy field over 26 years and hence does not account for temporal variability of the mean circulation, the location of the gyre, and the mean isopycnal depths and slopes. Thus, the depths used as delimiters of the three layers do not necessarily correspond to the actual, instantaneous pycnoclines that they are assumed to represent at all times. This is particularly true where the isopycnal surfaces are strongly tilted or even outcrop over the slope and the shelf break. There, fixed depth layers ensure that there is no averaging done between regions in contact with sea ice and those away from the sea ice influence that would otherwise complicate the interpretation of the results. The results presented in this study, and in particular the spatial structure of the eddy properties and their temporal variability, remain consistent when slightly varying the layers' upper and lower boundaries indicating the robustness of the key features reported for the three layers especially in the centre of the BG where the isopycnals are relatively flat.*

**Reviewer:** Finally, given the authors look at changes with time, as the amount of sea ice is being reduced, can the authors speculate about what their results may mean for the future. And discuss the implications for those potential changes.

**Authors:** Following the reviewer's suggestion, we have added a discussion on future changes to Sect. 3.2 and in the concluding remarks of the paper :

- *Lines 513-517 (Sect. 3.2) : Overall, our results suggest that, in the upper layer, the number of eddies do not only increase because of an expansion of the open ocean area at the expense of sea ice, but also because of an energy surplus in the Canadian Basin in the*

*MIZ and open ocean, in line with conclusions from other modelling studies [e.g., Rieck et al., 2025]. The increase of energy below the pack ice, suggested in future projections of the Arctic [Rieck et al., 2025], is only seen in the upper mixed layer in the current climate.*

- *Lines 583-587 (Discussion): We argue that the higher density in the eddy population is the result of an increasing penetration of energy in the upper layers of the basin where the ice concentration is small enough. Therefore, in the MIZ and open ocean, these results confirm findings from Rieck et al. [2025] of an increased energy input associated to more energy conversion toward EKE, and from Li et al. [2024] who associate this increased EKE to higher baroclinic instabilities, despite increased eddy killing with more mobile ice, in simulations of the future Arctic.*

**Reviewer:** End the discussion with more discussion on the limitations of the model and the present study.

**Authors:** We have edited the discussion to include more on the limitation of the model and of the study as per the reviewer's suggestion:

- Extended discussion on limitations associated to the detection and tracking algorithm, see above, and lines 613-617.

- Discussion on the possible biases associated to the slightly weak EKE in the model, see above and lines 617-622.

- Limited comparison with observations due to the horizontal and vertical resolution, see above and lines 608-611.

- The absence of 3D reconstruction is now mentioned in Sect. 2 (method) and in Sect. 3.1 (Results): *Lines 243-246 : No 3D representation of eddies is attempted here as connecting the results between the vertical layers is not trivial, and would require a substantial development of the detection and tracking algorithms. A brief evaluation of the vertical structure of eddies is however proposed in Section 3.1.*
* * *
**Minor comments**

**Reviewer:** Line 19 – The increase in the AW layer is over what time period?

**Authors:** We now mention in the text: *"over the 26 years of the simulation"*

**Reviewer:** Figures 1, 3, 5, 6: Please use discrete color contour intervals to make the figures easier to read. **Authors:** Following the reviewer's recommendation, it has been done for Fig. 3, 6, 7, 8, and 9. Tests on Fig. 1 did not show a significant improvement of the figure.

**Reviewer:** Figure 2. Could you explain in more detail how the anomalies are calculated, rather than just saying similarly from monthly mean anomalies.
**Authors:** The caption now reads :
*Super-imposed are mooring-based estimates of EKE from von Appen et al. [2022], computed with fourth-order Butterworth filter with 2-day to 30-day cutoffs. The reader is referred to von Appen et al. [2022] for exact calculation method.*

**Reviewer:** Line 333. Is this small decrease significant?
**Authors:** Indeed, this exact number is not really meaningful. The sentence now reads :
*Line 502-503 : Over the 26 years, we find [...] and no change on average below the pack ice (Fig. 10).*

**Reviewer:** L335: Might some of the increase with time be a function of changes in the winds and energy input with time?
**Authors:** We have included more details to the interpretation of this result:
*Line 505-509: In the MIZ and open ocean, the increase in eddy generation is mainly a step increase in 2008 with reduced (shut down) interannual variability in the MIZ (Open Ocean) in the following years. This enhancement in the density of the eddy population presumably results from the additional energy penetrating into the ocean in the recent state of the BG, in light with the current and future projections of Li et al. [2024]. In the open ocean, this accumulation of energy could be attributed to the acceleration of the BG from atmospheric forcings [Giles, 2012], and in the MIZ to a combination of less compacted ice [Martin et al., 2016] or to the thinner ice cover [Muilwijk et al., 2024]. A quantification of these different drivers is beyond the scope of this paper and left for future analysis.*

**Reviewer:** L348. Half more eddies doesn't read well.
**Authors:** It is now written : 30% *more eddies*

**Reviewer:** Figure 7 caption. What does gradient of the SSH averaged over the CB mean?
**Authors:** We first compute the gradient of SSH (two dimension vector), then compute its norm pointwise and average it over the CB domain. We reformulate it as :
*[...] norm of the gradient of SSH calculated at every location of the domain and averaged over*

*the CB*

**Reviewer:** Table 2. Remind people of the layer definitions in the table or caption.
**Authors:** Done.

**References**

Caili Liu, Qiang Wang, Sergey Danilov, Nikolay Koldunov, Vasco Müller, Xinyue Li, Dmitry Sidorenko, and Shaoqing Zhang. Spatial Scales of Kinetic Energy in the Arctic Ocean. *Journal of Geophysical Research: Oceans*, 129(3):e2023JC020013, March 2024. ISSN 2169-9275, 2169-9291. doi: 10.1029/2023JC020013. URL `https://agupubs.onlinelibrary.wiley.com/doi/10.1029/2023JC020013`.

A. J. G. Nurser and S. Bacon. Arctic Ocean Rossby radius Eddy length scales and the Rossby radius in the Arctic Ocean Arctic Ocean Rossby radius. *Ocean Sci. Discuss*, 10:1807–1831, 2013. ISSN 1807-1831. doi: 10.5194/osd-10-1807-2013. URL `www.ocean-sci-discuss.net/10/1807/2013/`.

Xinyue Li, Qiang Wang, Sergey Danilov, Nikolay Koldunov, Caili Liu, Vasco Müller, Dmitry Sidorenko, and Thomas Jung. Eddy activity in the Arctic Ocean projected to surge in a warming world. *Nature Climate Change*, 14(2):156–162, February 2024. ISSN 1758-6798. doi: 10.1038/s41558-023-01908-w. URL `https://www.nature.com/articles/s41558-023-01908-w`. Publisher: Nature Publishing Group.

Igor E. Kozlov, Anastasia V. Artamonova, Georgy E. Manucharyan, and Arseny A. Kubryakov. Eddies in the Western Arctic Ocean From Spaceborne SAR Observations Over Open Ocean and Marginal Ice Zones. *Journal of Geophysical Research: Oceans*, 124 (9):6601–6616, September 2019. ISSN 2169-9275, 2169-9291. doi: 10.1029/2019JC015113. URL `https://agupubs.onlinelibrary.wiley.com/doi/10.1029/2019JC015113`.

A. A. Kubryakov, I. E. Kozlov, and G. E. Manucharyan. Large Mesoscale Eddies in the Western Arctic Ocean From Satellite Altimetry Measurements. *Journal of Geophysical Research: Oceans*, 126(5):e2020JC016670, May 2021. ISSN 2169-9275, 2169-9291. doi: 10.1029/2020JC016670. URL `https://agupubs.onlinelibrary.wiley.com/doi/10.1029/2020JC016670`.

Georgy E. Manucharyan, Rosalinda Lopez-Acosta, and Monica M. Wilhelmus. Spinning ice floes reveal intensification of mesoscale eddies in the western Arctic Ocean. *Scientific Reports*, 12(1):7070, April 2022. ISSN 2045-2322. doi: 10.1038/s41598-022-10712-z. URL `https://www.nature.com/articles/s41598-022-10712-z`.

Angélina Cassianides, Camille Lique, Anne-Marie Tréguier, Gianluca Meneghello, and Charly De Marez. Observed Spatio-Temporal Variability of the Eddy-Sea Ice Interactions in the Arctic Basin. *Journal of Geophysical Research: Oceans*, 128(6): e2022JC019469, June 2023. ISSN 2169-9275, 2169-9291. doi: 10.1029/2022JC019469. URL `https://agupubs.onlinelibrary.wiley.com/doi/10.1029/2022JC019469`.

Mengnan Zhao, Mary-Louise Timmermans, Sylvia Cole, Richard Krishfield, Andrey Proshutinsky, and John Toole. Characterizing the eddy field in the Arctic Ocean halocline. *Journal of Geophysical Research: Oceans*, 119(12):8800–8817, December 2014. ISSN 2169-9291. doi: 10.1002/2014JC010488. URL `http://dx.doi.org/10.1002/2014JC010488`.

Gianluca Meneghello, John Marshall, Camille Lique, L Erik Isachsen, Edward Doddridge, Jean-Michel Campin, Heather Regan, and Claude Talandier. Genesis and Decay of Mesoscale Baroclinic Eddies in the Seasonally Ice-Covered Interior Arctic Ocean. *JOURNAL OF PHYSICAL OCEANOGRAPHY*, 51, 2021.

J Pedlosky. *Geophysical Fluid Dynamics*. Springer edition, 1982.

Georgy E. Manucharyan and Andrew L. Stewart. Stirring of interior potential vorticity gradients as a formation mechanism for large subsurface-intensified eddies in the Beaufort Gyre. *Journal of Physical Oceanography*, August 2022. ISSN 0022-3670, 1520-0485. doi: 10.1175/jpo-d-21-0040.1. URL `https://journals.ametsoc.org/view/journals/phoc/aop/JPO-D-21-0040.1/JPO-D-21-0040.1.xml`. Publisher: American Meteorological Society.

Heather C Regan, Camille Lique, Claude Talandier, and Gianluca Meneghello. Response of Total and Eddy Kinetic Energy to the Recent Spinup of the Beaufort Gyre. *JOURNAL OF PHYSICAL OCEANOGRAPHY*, 50, 2020.

Rebecca A. Woodgate. Increases in the Pacific inflow to the Arctic from 1990 to 2015, and insights into seasonal trends and driving mechanisms from year-round Bering Strait mooring data. *Progress in Oceanography*, 160:124–154, January 2018. ISSN 00796611. doi: 10.1016/j.pocean.2017.12.007. Publisher: Elsevier Ltd.

Claude Talandier and Camille Lique. CREG12.L75-REF12, August 2024. URL `https://doi.org/10.5281/zenodo.17350785`.

Tahya Weiss-Gibbons, Andrew Tefs, Xianmin Hu, Tricia Stadnyk, and Paul G. Myers. Sensitivity of Simulated Arctic Ocean Salinity and Strait Transport to Interannually Variable Hydrologic Model Based Runoff. *Journal of Geophysical Research: Oceans*, 129(10): e2023JC020536, October 2024. ISSN 2169-9275, 2169-9291. doi: 10.1029/2023JC020536. URL `https://agupubs.onlinelibrary.wiley.com/doi/10.1029/2023JC020536`.

Jinlun Zhang and D. A. Rothrock. Modeling Global Sea Ice with a Thickness and Enthalpy Distribution Model in Generalized Curvilinear Coordinates. *Monthly Weather Review*, 131(5):845–861, May 2003. ISSN 0027-0644, 1520-0493. doi: 10.1175/1520-0493(2003)131¡0845:MGSIWA¿2.0.CO;2. URL `http://journals.ametsoc.org/doi/10.1175/1520-0493(2003)131<0845:MGSIWA>2.0.CO;2`.

A. Proshutinsky, R. Krishfield, J. M. Toole, M.-L. Timmermans, W. Williams, S. Zimmermann, M. Yamamoto-Kawai, T. W. K. Armitage, D. Dukhovskoy, E. Golubeva, G. E. Manucharyan, G. Platov, E. Watanabe, T. Kikuchi, S. Nishino, M. Itoh, S.-H. Kang, K.-H. Cho, K. Tateyama, and J. Zhao. Analysis of the Beaufort Gyre Freshwater Content in 2003–2018. *Journal of Geophysical Research: Oceans*, 124(12):9658–9689, 2019. ISSN 2169-9291. doi: 10.1029/2019JC015281. URL `https://onlinelibrary.wiley.com/doi/abs/10.1029/2019JC015281`. _eprint: https://agupubs.onlinelibrary.wiley.com/doi/pdf/10.1029/2019JC015281.

Andrey Proshutinsky, Richard Krishfield, Mary-Louise Timmermans, John Toole, Eddy Carmack, Fiona McLaughlin, William J Williams, Sarah Zimmermann, Motoyo Itoh, and Koji Shimada. Beaufort Gyre freshwater reservoir: State and variability from observations. 2009.

J. K. Rieck, J. Martínez Moreno, C. Lique, C. O. Dufour, and C. Talandier. Mean Kinetic Energy and Its Projected Changes Dominate Over Eddy Kinetic Energy in the Arctic Ocean. *Geophysical Research Letters*, 52(22):e2025GL117957, November 2025. ISSN 0094-8276, 1944-8007. doi: 10.1029/2025GL117957. URL `https://agupubs.onlinelibrary.wiley.com/doi/10.1029/2025GL117957`.

Wilken-Jon von Appen, Till M. Baumann, Markus A Janout, Nikolay V. Koldunov, Yueng-Djern Lenn, Robert S. Pickart, and Qiang Wang. Eddies and the Distribution of Eddy Kinetic Energy in the Arctic Ocean. 35(Special issue on the new arctic ocean):42–51, December 2022.

Katharine A Giles. Western Arctic Ocean freshwater storage increased by wind-driven spin-up of the Beaufort Gyre. *NATURE GEOSCIENCE*, 5, 2012.

Torge Martin, Michel Tsamados, David Schroeder, and Daniel L. Feltham. The impact of variable sea ice roughness on changes in Arctic Ocean surface stress: A model study. *Journal of Geophysical Research: Oceans*, 121 (3):1931–1952, 2016. ISSN 2169-9291. doi: 10.1002/2015JC011186. URL `https://onlinelibrary.wiley.com/doi/abs/10.1002/2015JC011186`. _eprint: https://agupubs.onlinelibrary.wiley.com/doi/pdf/10.1002/2015JC011186.

Morven Muilwijk, Tore Hattermann, Torge Martin, and Mats A. Granskog. Future sea ice weakening amplifies wind-driven trends in surface stress and Arctic Ocean spin-up. *Nature Communications*, 15(1):6889, August 2024. ISSN 2041-1723. doi: 10.1038/s41467-024-50874-0. URL `https://www.nature.com/articles/s41467-024-50874-0`.

---

## Author Response (AR2)

**Authors responses to reviewer #1 comments**

**Characteristics of ocean mesoscale eddies in the Canadian Basin from a high resolution pan-Arctic model**

Noémie Planat, Carolina O. Dufour, Camille Lique, Jan K. Rieck, Claude Talandier, L. Bruno Tremblay

**Manuscript Reference Number: 2025-3527**

Date: January 22nd, 2026

In the following, all page numbers refer to the revised manuscript.

**General comments**

**Reviewer:** The authors have done a good job revising their manuscript. At this point, I just have some minor revisions to suggest, mainly related to expanding the discussion of the study's limitations in the paper's summary.

**Authors:** We thank the reviewer for their positive appreciation of the revised manuscript and for their suggestions which we address below.
* * *
**Minor comments**

**Reviewer:** In the abstract when the authors mention the eddy numbers, it might be good to clearly indicate that the majority of them are short lived.

**Authors:** The recommendation has been followed (lines 10-11):

*On average, eddies travel* 11 *km, have a radius of* 12.1 *km, and last* 10 *days, although the majority of eddies are short-lived (*50% *of eddies last less than* 4 *days).*

**Reviewer:** Line 80: In the vertical. **Authors:** Done, thank you.

**Reviewer:** Line 113: On the other hand. **Authors:** Done, thank you.

**Reviewer:** Line 180: Mars sais en Francais :-) – March. **Authors:** Done, thank you !

**Reviewer:** Line 486 – Are the authors sure that there is a climate signal for the eddies, or might it also be related to the model being better able to resolve eddies as the Rossby radius increases.

**Authors:** We agree with the Reviewer that the change in the Rossby radius over the course of the simulation may impact the reported increase in the eddy population density. However, we believe that this impact is small. Indeed, in the Beaufort Box (previously referred to as the Canada Basin; see response to the Editor), and more generally around the Beaufort Gyre area, the increase in the Rossby radius between the first and last decades of the simulation remains relatively small ($< 1$km, Fig. I, II). Only along the Canadian Archipelago and in the Nansen Basin does the Rossby radius increase by $1 - 3$ km, but these regions correspond to areas where the density of eddies detected is small. To put these increases in perspective, an increase of radius for a circular vortex from 10 km to 11 km increases its area by 5 units of grid cell area approximately. In addition, the number of eddies increases at all levels despite contrasted changes in stratification across the water column (e.g, in the Beaufort Box, the stratification increases in the pycnocline layer but decreases in the surface layer ). Finally, if we look at eddies whose sizes fall above the detection threshold (i.e., above the Rossby

[Figure]

Figure I: Rossby radius for the first decade (left) and last decade (middle), and the difference between the two (right) computed following the simplified equation introduced by Chelton et al. (1998) and compared with the exact formulation in the Arctic Ocean by Nurser and Bacon (2013).

radius) at all times during the simulation (e.g., eddies with radii between 15 and 20 km or those between 20 and 30 km), their density increases throughout the simulation. We have added a comment to elaborate on this result that appears on lines 549-553:

*Along with the increase in the eddy population, between the first and last 5 years of the simulation, eddies become bigger (+0.7 km), travel further (+2.2 km) and carry relatively warmer waters (+0.0027°C; Table 2). These changes are in line with an increased stratification, which increases the Rossby radius. We estimate a change of +0.5 km for the Rossby radius in the BB between the first and last decade of the simulation. This increase in the Rossby radius enhances the effective resolution of the model, potentially leading to a higher number of eddies detected. Yet, the change of effective resolution, defined as $R_0/ds$ (where ds is the maximal grid spacing), is only significant in the northeastern side of the domain (not shown), a region where we detect overall very few vortices (for instance, see Fig. 7)*

**Reviewer:** Need to end with a more detailed discussion on model limitations and the likely need to look at question at higher resolution to confirm results.

**Authors:** We have added a more detailed discussion that appears in the second-to-last paragraph of the section (lines 721-740):

*This study is a first attempt to perform a systematic and quantitative characterization of mesoscale eddy properties in the Canadian Basin. Yet, an evaluation of the modelled eddy characteristics against observations is hampered by the incomplete resolution of the mesoscale spectrum in our 1/12° model at these high latitudes, given that most features identified with moorings, ITPs or satellite fall between the meso- and the submeso-scales [Cassianides et al., 2023, Kozlov et al., 2019]. Therefore, these results should be reproduced by a model with*

[Figure]

Figure II: Brunt-Väisälä frequency squared ($N^2$) averaged over the Beaufort Box (BB).

*higher resolution to be confirmed and compared with observations. The increase in horizontal resolution should be accompanied by an increase in vertical resolution that would not only improve the representation of the (sub)meso-scale features, but also improve the representation of some of the processes sourcing these eddies, especially in the ML, and therefore more accurately represent the variety of features found in the Canadian Basin [e.g., Manucharyan and Timmermans, 2013]. A comparison of eddy characteristics between model and observations should account for the typical sampling biases in observations, such as the spatial distribution of the ITPs and the seasonality of the satellite acquisition of ocean surface properties due to the sea ice cover [Kozlov et al., 2019]. This comparison could be undertaken with an Observing System Simulation Experiment and would help further our current understanding of eddies from observations and quantify the biases and uncertainties associated with the available observations. Finally, the other important limitation of our study lies in the approach used for the detection of eddies, which lacks a vertical dimension. We believe implementing a 3D reconstruction of eddies could form a substantial improvement of the present work, and would allow us to tackle additional questions regarding the formation mechanisms of the mesoscale eddies. In particular, one would be able to investigate the transport of the mesoscale eddies along isopycnals and their subduction at depth, as suggested in Fig. 11 and in the literature [Manucharyan and Timmermans, 2013], or the links between isopycnal displacements and eddy generation [Cassianides et al., 2023].*

**References**

Angélina Cassianides, Camille Lique, Anne-Marie Tréguier, Gianluca Meneghello, and Charly De Marez. *Observed Spatio-Temporal Variability of the Eddy-Sea Ice Interactions in the Arctic Basin.* Journal of Geophysical Research: Oceans, *128(6):* e2022JC019469, June 2023. ISSN 2169-9275, 2169-9291. doi: 10.1029/2022JC019469. URL `https://agupubs.onlinelibrary.wiley.com/doi/10.1029/2022JC019469`.

Igor E. Kozlov, Anastasia V. Artamonova, Georgy E. Manucharyan, and Arseny A. Kubryakov. *Eddies in the Western Arctic Ocean From Spaceborne SAR Observations Over Open Ocean and Marginal Ice Zones.* Journal of Geophysical Research: Oceans, *124(9):* 6601–6616, September 2019. ISSN 2169-9275, 2169-9291. doi: 10.1029/2019JC015113. URL `https://agupubs.onlinelibrary.wiley.com/doi/10.1029/2019JC015113`.

Georgy E. Manucharyan and Mary-Louise Timmermans. *Generation and Separation of Mesoscale Eddies from Surface Ocean Fronts.* Journal of Physical Oceanography, *43(12):* 2545–2562, December 2013. ISSN 0022-3670, 1520-0485. doi: 10.1175/JPO-D-13-094.1. URL `http://journals.ametsoc.org/doi/10.1175/JPO-D-13-094.1`.

**Authors responses to the editor's comments**

**Characteristics of ocean mesoscale eddies in the Canadian Basin from a high resolution pan-Arctic model**

Noémie Planat, Carolina O. Dufour, Camille Lique, Jan K. Rieck, Claude Talandier, L. Bruno Tremblay

**Manuscript Reference Number: 2025-3527**

Date: January 22nd, 2026

In the following, all page numbers refer to the revised manuscript.

**General comments**

**Reviewer:** The article has been substantially re-written, which makes the points a bit clearer (although I still don't think it goes far enough). The re-writing however introduces quite a few internal inconsistencies that I caught on a not entirely careful read, to be point where in my opinion it ends up being worse the initial submission... (so in that regard I disagree with the referee). Some of these are noted below; the authors probably do want to go through it a few more times, as I make no promises I have caught everything (since I didn't do a very careful read). The line numbers refer to the line numbers in the tracked manuscript version.
**Authors:** We thank the editor for the detailed review of the paper. We have addressed all comments below and have carefully proofread the manuscript.
* * *
**Major comments**

**Reviewer:** It's a minor decision so it will just come back to me. The science points that need addressing for me are the Lagrangian vs Eulerian usage, and baroclinic eddies having no/little buoyancy signature. The framing could be adjusted somewhat (see suggestion), but that is not the deal breaker for me (the above are though).
**Authors:** We have adjusted the framing following the editor's suggestions, and hope the clarification of Eulerian vs Lagrangian usage, as well as the discussion regarding the temperature and salinity anomalies, will address the remaining scientific questions. These modifications are detailed below.

**Reviewer:** line 190: "Lagrangian framework" is misleading, because Okubo-Weiss is an Eulerian measure. What is being done here is a detection by an Eulerian framework, but some quantities that are Lagrangian in nature such as implied eddy centre (of mass? area/volume?) are provided. It is also contradictory with what is written in line 298 later in the article.
**Authors:** To address this comment, we have removed the mention of "Lagrangian framework" in the introduction. It is now only mentioned in the Method section when opposing Lagrangian and Eulerian detection methods.

**Reviewer:** line 737: The lack of buoyancy signature seems suspicious/contradictory, since baroclinic eddies (by the definition of "baroclinic") requires them to have a buoyancy signature right (baroclinic torque would be $\nabla p \times \nabla \rho$, but no buoyancy anomaly means $\nabla \rho = 0$ so there is no baroclinic torque...)? This either implies they are not in fact baroclinic (e.g. horizontal shear instabilities), the detection is not really detecting geostrophic eddies that

this paper is supposed to be centred around, or some other thing. This one needs a serious attempt at commenting now that it has been mentioned...

**Authors:** We thank the editor for pointing out this issue. Indeed, potential density anomalies should be non-zero for all detected vortices that form through baroclinic instability. However, these anomalies are possibly very small, especially in the centre of the basin, where weak gradients of temperature and salinity are expected to generate the vortices. Quantifying the anomaly itself is not trivial, as it depends on what one considers to be the eddy "environment". Therefore, we have designed the metric used to quantify "significant anomalies" to robustly detect and report the strongest anomalies. This metric is therefore restrictive as it does not quantify the weakest anomalies. We believe the use of this metric is the reason why we do not find a density anomaly for each eddy.

We have added a few lines in the methodology (lines 333-339):

*The properties of the eddy environment are defined by spatially averaging over a box that we take to be $n = 3$ times larger than the eddy dimensions in $x$ and $y$ directions (thus not of identical size along both directions) and from which we remove the eddy area. We note $\Delta X = X_i^{eddy} - X_i^{env}$ the anomaly of property $X$ at the time of eddy generation $i$. If two eddies develop next to each other, they will become each other's environment as we do not use a 2D eddy mask. In the interior of the basin, the gradients of density that may generate eddies are generally small, and so is the density anomaly of each eddy. To increase the robustness of the quantification of these anomalies, we choose to report only on the strongest anomalies.*

The abstract has been reformulated as (lines 20-23) :

*The vast majority of eddies have a weak temperature and salinity signature with respect to their environment, although a significant portion of the long-lived eddies, located along the Chukchi shelf break, have a relatively large temperature anomaly and penetrate into the Beaufort Gyre, thus suggesting a mechanism for the penetration of heat into the gyre.*

We also have modified the wording of the discussion section to recall this limitation (lines 624-625):

*In addition, the majority of eddies do not have a significant temperature nor salinity anomaly relative to their environment, where significant only accounts for the strong anomalies. Nevertheless, some non-negligible anomalies are visible along the shelf in the surface layer (Fig. 6e,f)*

**Minor comments**

**Reviewer:** general formatting: Copernicus now only offer reduced copy-editing. There is still type-setting available, but I am not sure various formattings, spellings, bibliography entries etc. are checked for, and there are a whole load of things I spotted below that the authors should fix themselves.

**Authors:** We thank the editor for spotting the various editorial issues left in the manuscript. We have corrected them all, as well as read the paper carefully to remove remaining spelling and formatting errors.

**Reviewer:** general framing and the detection per level: As a dynamicist I still think the eddy detection per level is a misleading thing to do, but I note from this second reading that from the second paragraph in the introduction that there might not be an alternative choice as such with e.g. Ice-Tethered Profilers. Then I wonder if a better way of justifying the present choice taken in the article as something like

- ITPs detect and attribute eddy signals per depth

- since this work may want to compare with observational data, the present article takes the choice of comparing things per depth following established convention

- comment that it is known that geostrophic eddies can and do have a vertical structure/coherency, the present attribution probably over-estimates numbers, but that is just not an aspect considered in this work

I don't think it is a very good convention but it is a convention, which allows some deflection of the criticism. I just think rather than not saying anything and let other people pick at holes, it's probably better to highlight the issues, shut down that discussion and move on. This should/could be done at the introduction.

**Authors:** We agree that the detection per model level might be misleading. However, we believe it is the most robust way to detect eddies over the whole basin, and over the 26 years. Following the recommendations of the editor and reviewers during the first round of review, a comment on the vertical structure and coherency of the eddies was added to the introduction and at the beginning of the result section, as this vertical coherency is used to justify the analysis per layer. The choice of 2D vs 3D is clarified in the method section, where the detection is introduced. Finally, we removed from the paper most mentions of comparisons with observations, at the exception of an opening in the discussion. We thus believe that the approach of an eddy detection per level is clearly justified now. The 3D detection approach

is now mentioned at the end of the paper as a way forward to improve the characterization of eddies and the investigation of their generation and dissipation (lines 734-740) : *Finally, the other important limitation of our study lies in the approach used for the detection of eddies, which lacks a vertical dimension. We believe implementing a 3D reconstruction of eddies could form a substantial improvement of the present work, and would allow us to tackle additional questions regarding the formation mechanisms of the mesoscale eddies. In particular, one would be able to investigate the transport of the mesoscale eddies along isopycnals and their subduction at depth, as suggested in Fig. 11 and in the literature [Manucharyan and Timmermans, 2013], or the links between isopycnal displacements and eddy generation [Cassianides et al., 2023].*

**Reviewer:** sentence of line 145: Changing the sea-ice also changes the "generation" mechanism and instability characteristic at least from a linear point of view (because of changing the boundary condition), which is not discussed. This is also in one of Gianluca's papers (don't remember if it is the one cited here).

**Authors:** We agree with the reviewer and have modified the text to mention these changes in the second-to-last paragraph of the introduction (lines 136-137): *Likewise, changes in stratification and sea ice cover may have affected the eddy activity, characteristics, generation, and dissipation mechanisms. Notably, stronger baroclinic instabilities result from a less concentrated ice cover [Meneghello et al., 2021].*

**Reviewer:** sentence of line 147: Unnecessary leap-frogging sentence (and elsewhere). Why not just do something like "Eddies may persist beyond months at subsurface since they are shielded..." or similar?

**Authors:** We have modified the sentence following the recommendation of the editor.

**Reviewer:** line 166: "ALONG the same line..."

**Authors:** We have modified the sentence following the recommendation of the editor.

**Reviewer:** line 162 and 154: "ON THE one hand..." and "ON the other hand..."

**Authors:** We have modified the sentence following the recommendation of the editor.

**Reviewer:** line 177: comma after "Canadian Basin" to separate the two clauses.

**Authors:** We have modified the sentence following the recommendation of the editor.

**Reviewer:** line 179: Sentence beginning "In this paper..." defines what "eddy" is being referred to but should be much further up surely (although probably not right at the beginning of the introduction section). A ton of talk so far about "eddies" and only now are "eddies" being defined, which is out of order.

**Authors:** We now specify at the beginning of the introduction what we call eddy in the rest of the paper (lines 46-48):

*In the literature, the term eddy encompasses a broad range of definitions. Observations of eddies in the Arctic Ocean have, however, mostly reported on coherent structures identified as anomalies with respect to their environment. Thus, from now onwards, we will focus on these coherent structures.*

**Reviewer:** line 185: Remove "Besides" (don't need it) We have modified the sentence following the recommendation of the editor.

**Reviewer:** line 195 and elsewhere: The paper uses "Canada basin" and "Canadian Basin (CB)", which seems more confusing than necessary (and annoyed me to no end actually particularly later on in the article when it seems to be used inconsistently). See later...

**Authors:** We understand the confusion around these two terms. To clarify, we have changed throughout the whole manuscript "Canada Basin (abbreviated CB in the manuscript)", which corresponds to an area within the Beaufort Gyre that we use for our analyses, into "Beaufort Box (BB)". The area is defined at the beginning of Sect. 2.1.2, lines 195-197:

*In the rest of this paper, the Canadian Basin is defined as the region between $69 - 85°N$ and $108 - 180°W$, thus fully encompassing the BG and its surrounding area. For analysis purposes, we define the Beaufort Box (BB), a region in the BG between $73 - 77°N$ and $135 - 152°W$ (see Fig. 1).*

**Reviewer:** line 204: Want mathfont for z**Reviewer:** (as "$z^*$")
**Authors:** We have modified the font accordingly.

**Reviewer:** line 208: Floating comma after "Fig. S1"
**Authors:** The comma has been removed.

**Reviewer:** line 211: "..., WITH AN ASSOCIATED COMPUTATIONAL cost that allows for decadal integradtionS."
**Authors:** We have modified the sentence following the recommendation of the editor.

**Reviewer:** line 213: "...viscosity AND diffusivity..."
**Authors:** We have modified the sentence following the recommendation of the editor.

**Reviewer:** line 215: Reference should be "de Lavergne" (lower case "d"). In bibtex this would be done with author = de Lavergne, C. and ...., or author = "de Lavergne, C. and ..."
**Authors:** We have modified the bibtex file following the recommendations of the editor.

**Reviewer:** line 222: Inconsistent unit formatting even within the same sentence (change to W m-1 K-1 or N / m2, choose one and stick with it).
**Authors:** Units have been adjusted to "m-1" formatting through the whole paper.

**Reviewer:** line 227: "At THE Bering strait..."
**Authors:** We have modified the sentence following the recommendation of the editor.

**Reviewer:** line 237: comma after "observations"
**Authors:** We have added the comma following the recommendation of the editor.

**Reviewer:** line 238: "Across the Arctic... by AROUND -7% in September and -16% in March ON AVERAGE."
**Authors:** We have modified the sentence following the recommendation of the editor.

**Reviewer:** line 240: French spelling floating around, "Mars" → "March"
**Authors:** We thank the editor for spotting this spelling mistake, it has been corrected.

**Reviewer:** (Elsewhere in the article some of the sentences have a French syntax style to it. That is not a big issue as such, but it sticks out at the moment at least to me.)
**Authors:** We have carefully read the paper before resubmission and hope that we corrected all the French syntax that might have been remaining.

**Reviewer:** line 242: "sea ice" what? (cover? thickness? volume?)
**Authors:** This has been modified to sea ice concentration.

**Reviewer:** line 242: comma after "model"
**Authors:** We have added the comma following the recommendation of the editor.

**Reviewer:** line 245: "...in THE supplementary MATERIAL"
**Authors:** We have modified the sentence following the recommendation of the editor.

**Reviewer:** line 248: "...successfully REpresents..."
**Authors:** We have modified the sentence following the recommendation of the editor.

**Reviewer:** line 253: remove ", that allows and sustains this vertical temperature structure,", don't think that's needed.
**Authors:** The sentence has been removed following the recommendation of the editor.

**Reviewer:** line 254: comma after "BG"
**Authors:** We have added the comma following the recommendation of the editor.

**Reviewer:** line 255: I thought the preferred way nowadays is to not use "psu" as a unit (since concentration is dimensionless). Consider "g kg-1" or "g / kg", just be consistent with choices made elsewhere.
**Authors:** We thank the editor for this precision, which has been corrected to g kg-1 following the chosen unit format.

**Reviewer:** Eq. 1: Inconsistent forcing of "x" and "y" to e.g. line 289 and 290 (don't use , or whatever is being used here).
**Authors:** We thank the editor for spotting this formatting issue. We have modified the character style for $x$, $y$ in Eq. (1) and (2) (but not u, v that are kept as "roman").

**Reviewer:** line 292: "Eddies have to meet the following condition TO BE RETAINED IN THE CENSUS:..."
**Authors:** We have modified the sentence following the recommendation of the editor.

**Reviewer:** Eq. 2: Inconsistent mathfont as above.
**Authors:** Adressed, see above

**Reviewer:** below eq. 2: $\alpha$ used but not defined here any more. Move some relevant text from below to be closer to eq. 2 (it's just some empirical user-specified parameter right, so just say it).
**Authors:** We have added after Eq. (2) the following sentence (line 267):
*where $\alpha$ is a threshold value typically chosen between 0.2 and 0.5 [Isern-Fontanet et al., 2003, Chelton et al., 2007, Pasquero et al., 2001].*

**Reviewer:** line 298: "Eulerian over Lagrangian approach" is currently internally inconsistent

with what is written in the introduction with "Lagrangian framework".

**Authors:** To clarify this point, we have modified the introduction to remove any mention of "Lagrangian framework". It is now only used when discussing the Eulerian and Lagrangian eddy detection methods.

**Reviewer:** line 308-309: If framing as "we take current approach to be consistent with what ITPs do" mentioned above, sentence here should probably be moved or repeated in the introduction also.

**Authors:** The paper is not an attempt to compare the eddy population of the model with that detected by ITPs. We are well aware of fundamental differences in the two populations, due to the model resolution, the lack of representation of mixed-layer instabilities by the model, etc. However, the 2D detection and tracking of eddies across coherent water layers is consistent with the model itself and bears similarities to the observations.

**Reviewer:** line 314: "have an elliptic shape" → "be elliptical in shape"
**Authors:** We have modified the sentence following the recommendation of the editor.

**Reviewer:** line 316: "...an eddy of ABOUT 7.5 to 10km..."
**Authors:** We have modified the sentence following the recommendation of the editor.

**Reviewer:** line 321: suggest "...metrics investigated (THE mean eddy radius...) are robust to..." and remove an internal bracket, because the bracketed content is the subject of the sentence ("metrics") being expanded, and sentence makes sense without the bracketed content in principle.

**Authors:** We have modified the sentence which now reads (lines 292-294): *Sensitivity tests for $\alpha$ show that the vertical distribution of the metrics investigated (the mean eddy radius, duration, polarity $r_{C/T}$ i.e. ratio of cyclones to total number of eddies, and a proxy for the vorticity $|\Omega|$, see Sect. 2.2.3), are robust to changes of $\alpha$ from 0.1 to 0.5.*

**Reviewer:** line 328: "Similarly" → "Similar"
**Authors:** We have modified the sentence following the recommendation of the editor.

**Reviewer:** line 331: Comma after "eddies"
**Authors:** We have added the comma following the recommendation of the editor.

**Reviewer:** line 373: Either "...is said TO BE significant" or "...is significant" (remove "said")

**Authors:** The sentence now reads (line 345):

*Then, the anomaly $\Delta X$ is said to be significant if ..*

**Reviewer:** line 380 and later: Since the authors took the trouble to define "CB" why isn't it used here? Easy fix would be to have "Across the Canadian Basin (CB)" as a reminder, and then use CB throughout.

**Authors:** The confusion between Canada and Canadian has been solved as we now call Beaufort Box (BB) the area used for the analysis of the centre of the Beaufort Gyre (and previously named Canada Basin).

**Reviewer:** Table 1: Above point about "psu" as a unit (also Fig. 4 and elsewhere)

**Authors:** Addressed, see above.

**Reviewer:** Table 1: Do use "th" here probably, and elsewhere (e.g. line 396, Fig. 4)

**Authors:** We have modified the character style following the recommendation of the editor in Table 1 and the rest of the paper.

**Reviewer:** line 408: Probably "The vertical structure is CLOSE to the vertical structure..."

**Authors:** We have modified the sentence, it now reads (line 376):

*This vertical structure is similar to the vertical structure obtained from observations*

**Reviewer:** line 417: "...ARE FOUND, forming together the pycnocline layer."

**Authors:** We have modified the sentence following the recommendation of the editor.

**Reviewer:** line 428: "follows" → "follow" (because "properties" is the relevant subject)

**Authors:** We have modified the sentence which now reads (lines 693-694):

*Within the top $85\ m$, about $6,000$ eddies are detected every year. The properties of these eddies show a marked seasonal cycle (Fig. 6), mainly following the seasonal cycle of the sea ice cover.*

**Reviewer:** line 437: "barely changes" makes "comparatively" redundant

**Authors:** We have removed "comparatively" following the recommendation of the editor.

**Reviewer:** Fig. 6 caption: Missing a comma between "5,000" and "10,000"

**Authors:** We have added the comma following the recommendation of the editor.

**Reviewer:** line 570: Pretty sure the acronym "PV" is no longer defined, so acronym used before definition.

**Authors:** We thank the editor for pointing out this inconsistency, which we have corrected by removing the acronym PV.

**Reviewer:** line 594: Comma in "9000" (e.g. inconsistent with "5,500" in the same sentence)
**Authors:** We thank the editor for pointing out this inconsistency, we have added the comma.

**Reviewer:** line 596: Confusing use of sign, is it a "decrease of 20%" (so no need for "reduced"), "decreased to 20%", "reduced to -20%", or some theme and variations thereof? Offending article is probably that minus sign.
**Authors:** We thank the editor for pointing out the redundancy, which we have now removed throughout the whole manuscript. +/- sign are only kept when no information on increase/decrease in previously given by the sentence.

**Reviewer:** line 599: inconsistent formatting on "s" (probably do want it roman)
**Authors:** We thank the editor for pointing out this inconsistency, we have modified the character style.

**Reviewer:** line 633: "...associated WITH the generation..."
**Authors:** We have modified the sentence following the recommendation of the editor.

**Reviewer:** line 638 and later: Similarly for CB here.
**Authors:** As mentioned above, we have modified CB in BB throughout the whole manuscript and clarified the wording when needed.

**Reviewer:** line 643, 645 and elsewhere: "decreases by -55%" means an increase of 55%, which I assume is not what is meant. Similarly for later and in the next paragraph.
**Authors:** This has been corrected throughout the whole manuscript, see above.

**Reviewer:** line 666: "fastened" means to "lock in" (e.g. "please fasten your seat belt", like "fast ice" is "locked in/anchored ice"), presumably "faster" is meant here?
**Authors:** We have modified the sentence following the recommendation of the editor.

**Reviewer:** line 674: the "in particular" at the end is floating, either something like "...and IN PARTICULAR to changes in atmospheric forcing" and elsewhere, or just remove it (don't need it).
**Authors:** "In particular" has been moved following the recommendation of the editor.

**Reviewer:** line 678: "...the EDDY POPULATION density..." to mirror that in line 680

**Authors:** We have modified the sentence following the recommendation of the editor.

**Reviewer:** line 685: "...and in the MIZ DUE to..."?

**Authors:** This sentence has been removed for clarity.

**Reviewer:** line 691: Another confusing bit where "Canadian Basin" was defined to be "CB" (but not used in the first case)...

**Authors:** Addressed, see above.

**Reviewer:** line 696: ...but also we have "the CB" and "THE whole Canadian Basin" in the same sentence! What is going on? (I am going to guess the latter one is "THE whole Canada basin", noting the missing "the" and lower case "basin" consistent with somewhere around section 2 when it was defined).

**Authors:** Addressed, see above.

**Reviewer:** line 704: Previously was "Northwind" but now it's "NorthWind", why?

**Authors:** We thank the reviewer for pointing out this inconsistency that has now been corrected throughout the manuscript.

**Reviewer:** line 708: "...a WEAKER increase in the number..."

**Authors:** We have modified the sentence following the recommendation of the editor.

**Reviewer:** line 714: "...interpret THE OBSERVED changes."

**Authors:** We have modified the sentence following the recommendation of the editor.

**Reviewer:** Fig. 10: Again, we have "Canadian Basin", "CB" and "Canada Basin" (with the capitalisation when it was introduced without). Please do something about this, because it's either confusing or annoying at present, and neither is a good option when it can be avoided with some internal consistency check...

**Authors:** Addressed, see above.

**Reviewer:** section 4: So now I no longer have faith in whether "Canadian Basin" here really means "Canada basin" or "CB". Please go through this again and make things internally consistent. I am going to guess it's actually "Canada basin" that is being referred to? Because

"Canada basin" as used here encompasses the "BG" and "CB", but honestly who knows any-more...

**Authors:** We hope the chosen new convention and the adapted wording are clearer and prevent possible confusion.

**Reviewer:** line 778: Same point above regarding sea-ice and changing generation mechanism.

**Authors:** We have added a sentence in the discussion to point to the increased baroclinic instability with reduced ice cover, as is demonstrated in Meneghello et al., 2021 (lines 652-654):

*The* $1995-2020$ *period is marked by an overall increase in eddy density at all depths ($35-45\%$), in line with Meneghello et al. [2021] findings of enhanced baroclinic instabilities with reduced ice cover, leading to enhanced eddy generation.*

**Reviewer:** line 891: Please consider acknowledging the referees/editor if the authors think the comments have actually helped (even if the authors may not have enjoyed reading the comments...)

**Authors:** It now appears at the end of the acknowledgment paragraph (lines 774-775):

*We finally thank two anonymous reviewers and the editor J. Mak for their constructive and insightful input that we believe greatly improved the paper.*

**References**

Georgy E. Manucharyan and Mary-Louise Timmermans. Generation and Separation of Mesoscale Eddies from Surface Ocean Fronts. *Journal of Physical Oceanography*, 43(12): 2545–2562, December 2013. ISSN 0022-3670, 1520-0485. doi: 10.1175/JPO-D-13-094.1. URL http://journals.ametsoc.org/doi/10.1175/JPO-D-13-094.1.

Angélina Cassianides, Camille Lique, Anne-Marie Tréguier, Gianluca Meneghello, and Charly De Marez. Observed Spatio-Temporal Variability of the Eddy-Sea Ice Interactions in the Arctic Basin. *Journal of Geophysical Research: Oceans*, 128(6): e2022JC019469, June 2023. ISSN 2169-9275, 2169-9291. doi: 10.1029/2022JC019469. URL https://agupubs.onlinelibrary.wiley.com/doi/10.1029/2022JC019469.

Gianluca Meneghello, John Marshall, Camille Lique, L Erik Isachsen, Edward Doddridge,

Jean-Michel Campin, Heather Regan, and Claude Talandier. Genesis and Decay of Mesoscale Baroclinic Eddies in the Seasonally Ice-Covered Interior Arctic Ocean. *JOURNAL OF PHYSICAL OCEANOGRAPHY*, 51, 2021.

Jordi Isern-Fontanet, Emilio Garcia-Ladona, and Jordi Font. Identification of Marine Eddies from Altimetric Maps. *JOURNAL OF ATMOSPHERIC AND OCEANIC TECHNOLOGY*, 20, 2003.

Dudley B. Chelton, Michael G. Schlax, Roger M. Samelson, and Roland A. De Szoeke. Global observations of large oceanic eddies. *Geophysical Research Letters*, 34(15): 2007GL030812, August 2007. ISSN 0094-8276, 1944-8007. doi: 10.1029/2007GL030812. URL `https://agupubs.onlinelibrary.wiley.com/doi/10.1029/2007GL030812`.

C. Pasquero, A. Provenzale, and A. Babiano. Parameterization of dispersion in two-dimensional turbulence. *Journal of Fluid Mechanics*, 439:279–303, July 2001. ISSN 0022-1120, 1469-7645. doi: 10.1017/S0022112001004499. URL `https://www.cambridge.org/core/product/identifier/S0022112001004499/type/journal`$_a rticle$.

---

## Author Response (AR3)

**Authors responses to the editor's comments**

**Characteristics of ocean mesoscale eddies in the Canadian Basin from a high resolution pan-Arctic model**

Noémie Planat, Carolina O. Dufour, Camille Lique, Jan K. Rieck, Claude Talandier, L. Bruno Tremblay

**Manuscript Reference Number: 2025-3527**

Date: February 4th, 2026

**Minor comments**

**Reviewer:** line 128: lowercase "g" for "gyre" (to be consistent with usage in line 130)
**Authors:** This has been adjusted following the recommendations of the editor.

**Reviewer:** line 184: for consistent should have a space between "W" and "m-1" to be consistent with usage everywhere else (e.g. line 192)
**Authors:** This has been adjusted following the recommendations of the editor.

**Reviewer:** line 185 and many times elsewhere (e.g. table 1, line 360, line 458, Fig 10 caption): the usage of · was fine before, not sure why it was removed, please restore or use × to have "a.b $\times 10^c$"
**Authors:** This has been adjusted following the recommendations of the editor.

**Reviewer:** (optional) line 260: would normally have the u and v italic rather than roman (so $u$ instead of u, u or u), but this is used consistently in the manuscript so could just ignore this comment.
**Authors:** The typesetting of $u$, $v$ has been modified following the recommendations of the editor.

**Reviewer:** fig 3 caption: the $y$ in $\partial_y u$ should be italic (i.e. don't add anything to it or typeset it like I did)
**Authors:** This has been adjusted following the recommendations of the editor.

**Reviewer:** fig 4 caption: remove the dash before the "th" and straighten it like in the text (e.g. line 368). I normally do something like probably "$10^{\text{th}}$
**Authors:** This has been adjusted following the recommendations of the editor.

**Reviewer:** fig 5 top row colobar units: inconsistent with usage elsewhere, want "$m^2 s^{-2}$"
**Authors:** This has been corrected.

**Reviewer:** table 2: are the semicolons under $\Delta S$ and $\Delta T$ meant to be there?
**Authors:** Thanks for spotting that mistake that has been adjusted.